# A Guide Through the Zoo of Biased SGD

**Yury Demidovich**
AI Initiative, KAUST
`yury.demidovich@kaust.edu.sa`

**Grigory Malinovsky**
AI Initiative, KAUST
`grigorii.malinovskii@kaust.edu.sa`

**Igor Sokolov**
AI Initiative, KAUST
`igor.sokolov.1@kaust.edu.sa`

**Peter Richtárik**
AI Initiative, KAUST
`peter.richtarik@kaust.edu.sa`

## Abstract

Stochastic Gradient Descent (SGD) is arguably the most important single algorithm in modern machine learning. Although SGD with unbiased gradient estimators has been studied extensively over at least half a century, SGD variants relying on biased estimators are rare. Nevertheless, there has been an increased interest in this topic in recent years. However, existing literature on SGD with biased estimators (BiasedSGD) lacks coherence since each new paper relies on a different set of assumptions, without any clear understanding of how they are connected, which may lead to confusion. We address this gap by establishing connections among the existing assumptions, and presenting a comprehensive map of the underlying relationships. Additionally, we introduce a new set of assumptions that is provably weaker than all previous assumptions, and use it to present a thorough analysis of BiasedSGD in both convex and non-convex settings, offering advantages over previous results. We also provide examples where biased estimators outperform their unbiased counterparts or where unbiased versions are simply not available. Finally, we demonstrate the effectiveness of our framework through experimental results that validate our theoretical findings.

## 1 Introduction

Stochastic Gradient Descent (SGD) [Robbins and Monro, 1951] is a widely used and effective algorithm for training various models in machine learning. The current state-of-the-art methods for training deep learning models are all variants of SGD [Goodfellow et al., 2016; Sun, 2020]. The algorithm has been extensively studied in recent theoretical works [Bottou et al., 2018; Gower et al., 2019; Khaled and Richtárik, 2023]. In practice and theory, SGD with *unbiased* gradient oracles is mostly used. However, there has been a recent surge of interest in SGD with *biased* gradient oracles, which has been studied in several papers and applied in different domains.

In distributed parallel optimization where data is partitioned across multiple nodes, communication can be a bottleneck, and techniques such as structured sparsity [Alistarh et al., 2018; Wangni et al., 2018] or asynchronous updates [Niu et al., 2011] are involved to reduce communication costs. Nonetheless, sparsified or delayed SGD-updates are not unbiased anymore and require additional analysis [Stich and Karimireddy, 2020; Beznosikov et al., 2020].

Zeroth-order methods are often utilized when there is no access to unbiased gradients, e.g., for optimization of black-box functions [Nesterov and Spokoiny, 2017], or for finding adversarial examples in deep learning [Moosavi-Dezfooli et al., 2017; Chen et al., 2017]. Many zeroth-order training methods exploit biased gradient oracles [Nesterov and Spokoiny, 2017; Liu et al., 2018; Bergou et al., 2020; Boucherouite et al., 2022]. Various other techniques as smoothing, proximate

updates and preconditioning operate with inexact gradient estimators [d'Aspremont, 2008; Schmidt et al., 2011; Devolder et al., 2014; Tappenden et al., 2016; Karimireddy et al., 2018].

The aforementioned applications illustrate that SGD can converge even if it performs *biased* gradient updates, provided that certain "regularity" conditions are satisfied by the corresponding gradient estimators [Bottou et al., 2018; Ajalloeian and Stich, 2020; Beznosikov et al., 2020; Condat et al., 2022]. Moreover, biased estimators may show better performance over their unbiased equivalents in certain settings [Beznosikov et al., 2020].

In this work we study convergence properties and worst-case complexity bounds of stochastic gradient descent (SGD) with a *biased* gradient estimator (BiasedSGD; see Algorithm 1) for solving general optimization problems of the form

$$\min_{x \in \mathbb{R}^d} f(x),$$

where the function $f : \mathbb{R}^d \to \mathbb{R}$ is possibly nonconvex, satisfies several smoothness and regularity conditions.

**Assumption 0** *Function $f$ is differentiable, $L$-smooth (i.e., $\|\nabla f(x) - \nabla f(y)\| \leq L \|x - y\|$ for all $x, y \in \mathbb{R}^d$), and bounded from below by $f^* \in \mathbb{R}$.*

We write $g(x)$ for the gradient estimator, which is biased (i.e., $\mathbb{E}[g(x)]$ is not equal to $\nabla f(x)$, $\mathbb{E}[\cdot]$ stands for the expectation with respect to the randomness of the algorithm), in general. By a gradient estimator we mean a (possibly random) mapping $g : \mathbb{R}^d \to \mathbb{R}^d$ with some constraints. We denote by $\gamma$ an appropriately chosen learning rate, and $x^0 \in \mathbb{R}^d$ is a starting point of the algorithm.

---

**Algorithm 1** Biased Stochastic Gradient Descent (BiasedSGD)

---

**Input:** initial point $x^0 \in \mathbb{R}^d$; learning rate $\gamma > 0$
1: **for** $t = 0, 1, 2, \ldots$ **do**
2:     Construct a (possibly biased) estimator $g^t \stackrel{\text{def}}{=} g(x^t)$ of the gradient $\nabla f(x^t)$
3:     Compute $x^{t+1} = x^t - \gamma g^t$
4: **end for**

---

In the strongly convex case, $f$ has a unique global minimizer which we denote by $x^*$, and $f(x^*) = f^*$. In the nonconvex case, $f$ can have many local minima and/or saddle points. It is theoretically intractable to solve this problem to global optimality [Nemirovsky and Yudin, 1983]. Depending on the assumptions on $f$, and given some error tolerance $\varepsilon > 0$, will seek to find a random vector $x \in \mathbb{R}^d$ such that one of the following inequalities holds: i) $\mathbb{E}[f(x) - f^*] \leq \varepsilon$ (convergence in function values); ii) $\mathbb{E}\|x - x^*\|^2 \leq \varepsilon \|x^0 - x^*\|^2$ (iterate convergence); iii) $\mathbb{E}\|\nabla f(x)\|^2 \leq \varepsilon^2$ (gradient norm convergence).

## 2 Sources of bias

Practical applications of SGD typically involve the training of supervised machine learning models via empirical risk minimization [Shalev-Shwartz and Ben-David, 2014], which leads to optimization problems of a finite-sum structure:

$$f(x) = \frac{1}{n} \sum_{i=1}^n f_i(x). \tag{1}$$

In the single-machine setup, $n$ is the number of data points, $f_i(x)$ represents the loss of a model $x$ on a data point $i$. In this setting, data access is expensive, $g(x)$ is usually constructed with *subsampling* techniques such as minibatching and importance sampling. Generally, a subset $S \subseteq [n]$ of examples is chosen, and subsequently $g(x)$ is assembled from the information stored in the gradients of $\nabla f_i(x)$ for $i \in S$ only. This leads to estimators of the form $g(x) = \sum_{i \in S} v_i \nabla f_i(x)$, where $v_i$ are random variables typically designed to ensure the unbiasedness [Gower et al., 2019]. In practice, points might be sampled with unknown probabilities. In this scenario, a reasonable strategy to estimate the gradient is to take an average of all sampled $\nabla f_i$. In general, the estimator obtained is biased, and

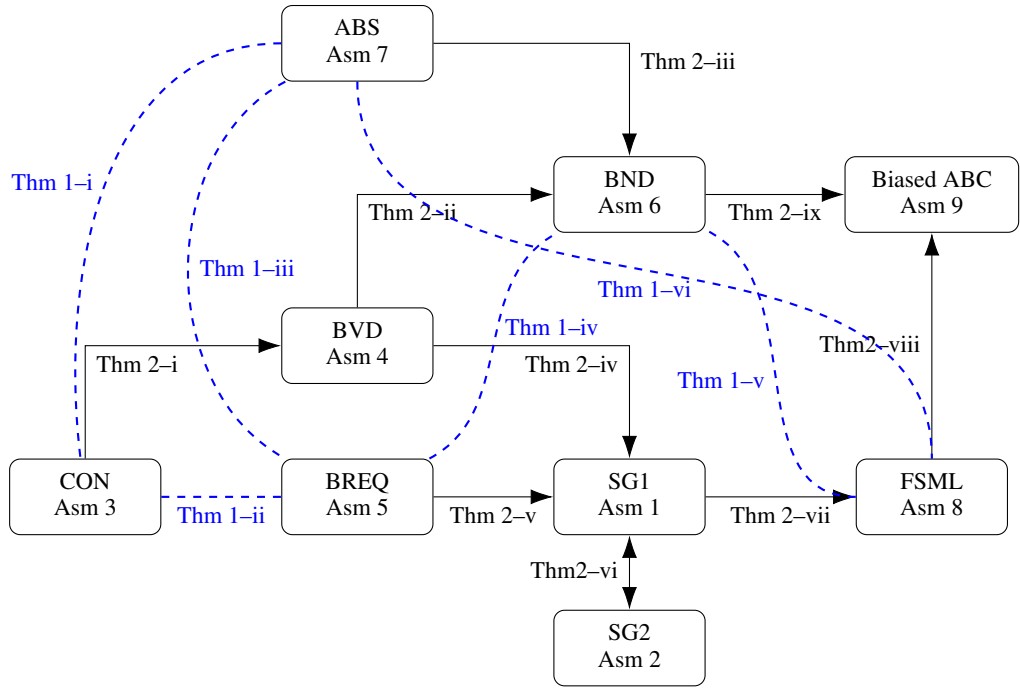

Figure 1: Assumption hierarchy. A single arrow indicates an implication and an absence of a reverse implication. The implications are transitive. A dashed line indicates a mutual abscence of implications. Our newly proposed assumption Biased ABC is the most general one.

such sources of bias can be characterized as arising from a lack of information about the subsampling strategy.

In the distributed setting, $n$ represemts the number of machines, and each $f_i$ represents the loss of model $x$ on all the training data stored on machine $i$. Since communication is typically very expensive, modern gradient-type methods rely on various gradient compression mechanisms that are usually randomized. Given an appropriately chosen compression map $\mathcal{C} : \mathbb{R}^d \to \mathbb{R}^d$, the local gradients $\nabla f_i(x)$ are first compressed to $\mathcal{C}_i(\nabla f_i(x))$, where $\mathcal{C}_i$ is an independent realization of $\mathcal{C}$ sampled by machine $i$ in each iteration, and subsequently communicated to the master node, which performs aggregation (typically averaging). This gives rise to SGD with the gradient estimator of the form

$$g(x) = \frac{1}{n} \sum_{i=1}^{n} \mathcal{C}_i\left(\nabla f_i(x)\right). \tag{2}$$

Many important compressors performing well in practice are of biased nature (e.g., Top-$k$, see Def. 3), which, in general, makes $g(x)$ biased as well.

Biased estimators are capable of absorbing useful information in certain settings, e.g., in the heterogeneous data regime. Unbiased estimators have to be random, otherwise they are equal to the identity mapping. However, greedy deterministic gradient estimators such as Top-$k$ often lead to better practical performance. In [Beznosikov et al., 2020, Section 4] the authors show an advantage of the Top-$k$ compressor over its randomized counterpart Rand-$k$ when the coordinates of the vector that we wish to compress are distributed uniformly or exponentially. In practice, deterministic biased compressors are widely used for low precision training, and exhibit great performance [Alistarh et al., 2018; Beznosikov et al., 2020].

## 3 Contributions

The most commonly used assumptions for analyzing SGD with biased estimators take the form of various structured bounds on the first and the second moments of $g(x)$. We argue that assumptions proposed in the literature are often too strong, and may be unrealistic as they do not fully capture how

bias and randomness in $g(x)$ arise in practice. In order to retrieve meaningful theoretical insights into the operation of BiasedSGD, it is important to model the bias and randomness both correctly, so that the assumptions we impart are provably satisfied, and accurately, so as to obtain as tight bounds as possible. Our work is motivated by the need of a more accurate and informative analysis of BiasedSGD in the strongly convex and nonconvex settings, which are problems of key importance in optimization research and deep learning. Our results are generic and cover both subsampling and compression-based estimators, among others.

The key contributions of our work are:

• Inspired by recent developments in the analysis of SGD in the nonconvex setting [Khaled and Richtárik, 2023], the analysis of BiasedSGD [Bottou et al., 2018; Ajalloeian and Stich, 2020], the analysis of biased compressors [Beznosikov et al., 2020], we propose a new assumption, which we call Biased ABC, for modeling the first and the second moments of the stochastic gradient.

• We show in Section 5.2 that Biased ABC is the weakest, and hence the most general, among all assumptions in the existing literature on BiasedSGD we are aware of (see Figure 1), including concepts such as Contractive (CON) [Cordonnier, 2018; Stich et al., 2018; Beznosikov et al., 2020], Absolute (ABS) [Sahu et al., 2021], Bias-Variance Decomposition (BVD) [Condat et al., 2022], Bounded Relative Error Quantization (BREQ) [Khirirat et al., 2018b], Bias-Noise Decomposition (BND) [Ajalloeian and Stich, 2020], Strong Growth 1 (SG1) and Strong Growth 2 (SG2) [Beznosikov et al., 2020], and First and Second Moment Limits (FSML) [Bottou et al., 2018] estimators.

• We prove that unlike the existing assumptions, which implicitly assume that the bias comes from either perturbation or compression, Biased ABC also holds in settings such as subsampling.

• We recover the optimal rates for general smooth nonconvex problems and for problems under the PŁ condition in the unbiased case and prove that these rates are also optimal in the biased case.

• In the strongly convex case, we establish a similar convergence result in terms of iterate norms as in [Hu et al., 2021a], however, under milder assumptions and not only for the classical version of SGD. Our proof strategy is very different and much simpler.

## 4   Existing models of biased gradient estimators

Since application of a gradient compressor to the gradient constitutes a gradient estimator, below we often reformulate known assumptions and results obtained for biased compressors in the more general form of biased gradient estimators. Beznosikov et al. [2020] analyze SGD under the assumption that $f$ is $\mu$-strongly convex, and propose three different assumptions for compressors.

**Assumption 1 (Strong Growth 1, SG1 – Beznosikov et al. [2020])** *Let us say that $g(x)$ belongs to a set $\mathbb{B}^1(\alpha, \beta)$ of biased gradient estimators, if, for some $\alpha, \beta > 0$, for every $x \in \mathbb{R}^d$, $g(x)$ satisfies*

$$\alpha \|\nabla f(x)\|^2 \le \mathbb{E}\left[\|g(x)\|^2\right] \le \beta \langle \mathbb{E}\left[g(x)\right], \nabla f(x) \rangle. \tag{3}$$

**Assumption 2 (Strong Growth 2, SG2 – Beznosikov et al. [2020])** *Let us say that $g(x)$ belongs to a set $\mathbb{B}^2(\tau, \beta)$ of biased gradient estimators, if, for some $\tau, \beta > 0$, for every $x \in \mathbb{R}^d$, $g(x)$ satisfies*

$$\max\left\{\tau\|\nabla f(x)\|^2, \frac{1}{\beta}\mathbb{E}\left[\|g(x)\|^2\right]\right\} \le \langle \mathbb{E}[g(x)], \nabla f(x) \rangle. \tag{4}$$

Note that each of Assumptions 1 and 2 imply

$$\mathbb{E}\left[\|g(x)\|^2\right] \le \beta^2 \|\nabla f(x)\|^2. \tag{5}$$

**Assumption 3 (Contractive, CON – Beznosikov et al. [2020])** *Let us say that $g(x)$ belongs to a set $\mathbb{B}^3(\delta)$ of biased gradient estimators, if, for some $\delta > 0$, for every $x \in \mathbb{R}^d$, $g(x)$ satisfies*

$$\mathbb{E}\left[\|g(x) - \nabla f(x)\|^2\right] \le \left(1 - \frac{1}{\delta}\right) \|\nabla f(x)\|^2. \tag{6}$$

The last condition is an abstraction of the contractive compression property (see Appendix L). Condat et al. [2022] introduce another assumption for biased compressors, influenced by a bias-variance decomposition equation for the second moment:

$$\mathbb{E}\left[\|g(x) - \nabla f(x)\|^2\right] = \|\mathbb{E}[g(x)] - \nabla f(x)\|^2 + \mathbb{E}\left[\|g(x) - \mathbb{E}[g(x)]\|^2\right]. \tag{7}$$

Let us write the assumption itself.

**Assumption 4 (Bias-Variance Decomposition, BVD – Condat et al. [2022])** *Let $0 \leq \eta \leq 1, \xi \geq 0$, for all $x \in \mathbb{R}^d$, the gradient estimator $g(x)$ satisfies*

$$\|\mathbb{E}[g(x)] - \nabla f(x)\|^2 \leq \eta \|\nabla f(x)\|^2, \tag{8}$$

$$\mathbb{E}\left[\|g(x) - \mathbb{E}[g(x)]\|^2\right] \leq \xi \|\nabla f(x)\|^2. \tag{9}$$

Khirirat et al. [2018b] proposed another assumption on deterministic compressors.

**Assumption 5 (Bounded Relative Error Quantization, BREQ – Khirirat et al. [2018b])** *For all $x \in \mathbb{R}^d$, for any $\rho, \zeta \geq 0$,*

$$\langle g(x), \nabla f(x) \rangle \geq \rho \|\nabla f(x)\|^2, \tag{10}$$

$$\|g(x)\|^2 \leq \zeta \|\nabla f(x)\|^2. \tag{11}$$

The restriction below was imposed on the gradient estimator $g(x)$ by Ajalloeian and Stich [2020]. For the purpose of clarity, we rewrote it in the notation adopted in our paper. We refer the reader to Appendix O for the proof of equivalence of these two definitioins.

**Assumption 6 (Bias-Noise Decomposition, BND – Ajalloeian and Stich [2020])** *Let $M, \sigma^2, \varphi^2$ be nonnegative constants, and let $0 \leq m < 1$. For all $x \in \mathbb{R}^d$, $g(x)$ satisfies*

$$\mathbb{E}\left[\|g(x) - \mathbb{E}[g(x)]\|^2\right] \leq M \|\mathbb{E}[g(x)]\|^2 + \sigma^2, \tag{12}$$

$$\|\mathbb{E}[g(x)] - \nabla f(x)\|^2 \leq m \|\nabla f(x)\|^2 + \varphi^2. \tag{13}$$

The following assumption was introduced by Sahu et al. [2021] (see also the work of Danilova and Gorbunov [2022]).

**Assumption 7 (Absolute Estimator, ABS – Sahu et al. [2021])** *For all $x \in \mathbb{R}^d$, there exists $\Delta \geq 0$ such that*

$$\mathbb{E}\left[\|g(x) - \nabla f(x)\|^2\right] \leq \Delta^2. \tag{14}$$

This condition is tightly related to the contractive compression property (see Appendix M). Further, Bottou et al. [2018] proposed the following restriction on a stochastic gradient estimator.

**Assumption 8 (First and Second Moment Limits, FSML – Bottou et al. [2018])** *There exist constants $0 < q \leq u, U \geq 0, Q \geq 0$, such that, for all $x \in \mathbb{R}^d$,*

$$\langle \nabla f(x), \mathbb{E}[g(x)] \rangle \geq q \|\nabla f(x)\|^2, \tag{15}$$

$$\|\mathbb{E}[g(x)]\| \leq u \|\nabla f(x)\|, \tag{16}$$

$$\mathbb{E}\left[\|g(x) - \mathbb{E}[g(x)]\|^2\right] \leq U \|\nabla f(x)\|^2 + Q. \tag{17}$$

Our first theorem, described informally below and stated and proved formally in the appendix, provides required counterexamples of problems and estimators for the diagram in Figure 1.

**Theorem 1** (Informal) *The assumptions connected by dashed lines in Figure 1 are mutually non-implicative.*

The result says that some pairs of assumptions are in a certain sense unrelated: none implies the other, and vice versa. In the next section, we introduce a new assumption, and provide deeper connections between all assumptions.

# 5 New approach: biased ABC assumption

## 5.1 Brief history

Several existing restrictions on the first moment of the estimator were very briefly outlined in the previous section (see (3), (8), (10), (13), (15)). Khaled and Richtárik [2023] recently introduced a very general and accurate Expected Smoothness assumption (we will call it the ABC-assumption in this paper) on the second moment of the unbiased estimator. Let us note that Polyak and Tsypkin [1973] explored a related assumption during their analysis of pseudogradient algorithms. They succeeded in establishing an asymptotic convergence bound for a variant of gradient descent in the unbiased scenario. In contrast, our study focuses on non-asymptotic convergence rates in the biased setting. We generalize the restrictions (3), (10), (15) on the first moment and combine them with the ABC-assumption to develop our Biased ABC framework.

**Assumption 9 (Biased ABC)** *There exist constants $A, B, C, b, c \geq 0$ such that the gradient estimator $g(x)$ for every $x \in \mathbb{R}^d$ satisfies[1]*

$$\langle \nabla f(x), \mathbb{E}[g(x)] \rangle \geq b \|\nabla f(x)\|^2 - c, \tag{18}$$

$$\mathbb{E}\left[\|g(x)\|^2\right] \leq 2A\left(f(x) - f^*\right) + B\|\nabla f(x)\|^2 + C. \tag{19}$$

The term $A\left(f(x) - f^*\right)$ in (19) naturally emerges when we bound the expression of the form $\sum_{i=1}^n q_i \|\nabla f_i(x)\|^2$, $q_i \geq 0$, $i \in [n]$ : while it can not be confined solely by the norm of the overall gradient $B\|\nabla f(x)\|^2$, nor by a constant $C$, smoothness suffices to bound this by $A\left(f(x) - f^*\right)$. Further, there exist quadratic stochastic optimization problems where the second moment of a stochastic gradient is precisely equal to $2(f(x) - f^*)$ (see Richtárik and Takáč [2020]).

Concerning the challenges in verifying the Biased ABC assumption, it is worth mentioning that in Machine Learning, loss functions are commonly bounded from below by $f^* = 0$. In Tables 2 and 8, we provide the constants that validate the fulfillment of our assumption by a wide range of practical estimators. Furthermore, Claims 2–4 can aid in determining these constants for various sampling schemes.

## 5.2 Biased ABC as the weakest assumption

As discussed in Section 4, there exists a Zoo of assumptions on the stochastic gradients in literature on BiasedSGD. Our second theorem, described informally below and stated and proved formally in the appendix, says that our new Biased ABC assumption is the least restrictive of all the assumptions reviewed in Section 4.

**Theorem 2** (Informal) *Assumption 9 (Biased ABC) is the weakest among Assumptions 1 − 9.*

Inequality (8) of BVD or inequality (13) of BND show that one can impose the restriction on the first moment by bounding the norm of the bias. We choose inequality (18) that restrains the scalar product between the estimator and the gradient on purpose: this approach turns out to be more general on its own. In the proof of Theorem 2-ix (see (46) and (47)) we show that (13) implies (18). Below we show the existence of a counterexample that the reverse implication does not hold.

**Claim 1** *There exists a finite-sum minimization problem for which a gradient estimator that satisfies inequality (18) of Assumption 9 does not satisfy inequality (13) of Assumption 6.*

The relationships among Assumptions 1–9 are depicted in Figure 1 based on the results of Theorem 1 and Theorem 2. It is evident from Figure 1 that Assumption 6 (BND) and Assumption 8 (FSML) are mutually non-implicative and represent the most general assumptions among those proposed in Assumptions 1–8.

The most significant difference between our Assumption 9 (Biased ABC) and Assumptions 6 and 8 is the inclusion of the term $A\left(f(x) - f^*\right)$ in the bound on the second moment of the estimator.

---

[1]In [Khaled and Richtárik, 2023], the "ABC assumption" was introduced in the unbiased case. However, we aim to establish theory for biased estimators. If we simply remove (18), then $g(x) = -\nabla f(x)$ satisfies (19) with $A = 0$, $B = 1$, $C = 0$, yet BiasedSGD clearly diverges in general.

| Assumption | $A$ | $B$ | $C$ | $b$ | $c$ |
|---|---|---|---|---|---|
| Asm 1 (**SG1**) [Beznosikov et al., 2020] | 0 | $\beta^2$ | 0 | $\frac{\alpha}{\beta}$ | 0 |
| Asm 2 (**SG2**) [Beznosikov et al., 2020] | 0 | $\beta^2$ | 0 | $\tau$ | 0 |
| Asm 3 (**CON**) [Beznosikov et al., 2020] | 0 | $2\left(2 - \frac{1}{\delta}\right)$ | 0 | $\frac{1}{2\delta}$ | 0 |
| Asm 4 (**BVD**) [Condat et al., 2022] | 0 | $2(1 + \xi + \eta)$ | 0 | $\frac{1-\eta}{2}$ | 0 |
| Asm 5 (**BREQ**) [Khirirat et al., 2018b] | 0 | $\zeta$ | 0 | $\rho$ | 0 |
| Asm 6 (**BND**) [Ajalloeian and Stich, 2020] | 0 | $2(M+1)(m+1)$ | $2(M+1)\varphi^2 + \sigma^2$ | $\frac{1-m}{2}$ | $\frac{\varphi^2}{2}$ |
| Asm 7 (**ABS**) [Sahu et al., 2021] | 0 | $2$ | $2\Delta^2$ | $\frac{1}{2}$ | $\frac{\Delta^2}{2}$ |
| Asm 8 (**FSML**) [Bottou et al., 2018] | 0 | $U + u^2$ | $Q$ | $q$ | 0 |

Table 1: Summary of known assumptions on biased stochastic gradients. Estimators satisfying any of them, belong to our general Biased ABC framework with parameters $A$, $B$, $C$, $b$ and $c$ provided in this table. For proofs, we refer the reader to Theorem 13.

The rationale behind this inclusion was detailed in Section 5.1. In general, estimators of the form $\sum_{i=1}^{n} q_i \left|\nabla f_i(x)\right|^2$, where $q_i \geq 0$, for $i \in [n]$, often arise in sampling schemes. We present two practical settings with sampling schemes (see Definitions 1 and 2) that can be described within the Biased ABC framework. These settings, in general, fall outside of the BND and FSML frameworks.

In Section D.2 (see Proof of Theorem 2, parts viii and ix) we present an example of a setting with a minimization problem and a gradient estimator that justifies the introduction of this term: BND and FSML frameworks do not capture the proposed setting, while Biased ABC does capture it.

In Table 1 we provide a representation of each of Assumptions 1 – 8 in our Biased ABC framework (based on the results of Theorem 13). Note that the constants in Table 1 are too pessimistic: given the estimator satisfying one of these assumptions, direct computation of constants in Biased ABC scope for it might lead to much more accurate results. In Table 2 we give a description of popular gradient estimators in terms of the Biased ABC framework. Finally, in Table 3 we list several popular estimators and indicate which of Assumptions 1–9 they satisfy.

| Estimator | Def | $A$ | $B$ | $C$ | $b$ | $c$ |
|---|---|---|---|---|---|---|
| **Biased independent sampling** [This paper] | Def. 1 | $\frac{\max_i\{L_i\}}{\min_i p_i}$ | 0 | $2A\Delta^* + s^2$ | $\min_i \{p_i\}$ | 0 |
| **Top-$k$** [Aji and Heafield, 2017] | Def. 3 | 0 | 1 | 0 | $\frac{k}{d}$ | 0 |
| **Rand-$k$** Stich et al. [2018] | Def. 4 | 0 | $\frac{d}{k}$ | 0 | 1 | 0 |
| **Biased Rand-$k$** [Beznosikov et al., 2020] | Def. 5 | 0 | $\frac{k}{d}$ | 0 | $\frac{k}{d}$ | 0 |
| **Adaptive random sparsification** [Beznosikov et al., 2020] | Def. 6 | 0 | 1 | 0 | $\frac{1}{d}$ | 0 |
| **General unbiased rounding** [Beznosikov et al., 2020] | Def. 7 | 0 | $\sup_{k \in \mathbb{Z}} \frac{a_k^2 + a_{k+1}^2}{4 a_k a_{k+1}} + \frac{1}{2}$ | 0 | 1 | 0 |
| **Natural compression** [Horváth et al., 2022] | Def. 9 | 0 | $\frac{9}{8}$ | 0 | 1 | 0 |
| **Scaled integer rounding** [Sapio et al., 2021] | Def. 15 | 0 | 2 | $\frac{2d}{\chi^2}$ | $\frac{1}{2}$ | $\frac{d}{2\chi^2}$ |

Table 2: Summary of popular estimators with respective parameters $A$, $B$, $C$, $b$ and $c$, satisfying our general Biased ABC framework. Constants $L_i$ are from Assumption 13, $\Delta^*$ is defined in (26). For more estimators, see Table 8.

## 6 Convergence of biased SGD under the biased ABC assumption

Convergence rates of theorems below are summarized in Table 4 and compared to their counterparts.

| Estimator \ Assumption | A1 | A2 | A3 | A4 | A5 | A6 | A7 | A8 | A9 |
|---|---|---|---|---|---|---|---|---|---|
| **Biased independent sampling** [This paper] | ✗ | ✗ | ✗ | ✗ | ✗ | ✗ | ✗ | ✗ | ✓ |
| **Top-$k$ sparsification** [Aji and Heafield, 2017] | ✓ | ✓ | ✓ | ✓ | ✓ | ✓ | ✗ | ✓ | ✓ |
| **Rand-$k$** [Stich et al., 2018] | ✓ | ✓ | ✗ | ✓ | ✗ | ✓ | ✗ | ✓ | ✓ |
| **Biased Rand-$k$** [Beznosikov et al., 2020] | ✓ | ✓ | ✓ | ✓ | ✗ | ✓ | ✗ | ✓ | ✓ |
| **Adaptive random sparsification** [Beznosikov et al., 2020] | ✓ | ✓ | ✓ | ✓ | ✗ | ✓ | ✗ | ✓ | ✓ |
| **General unbiased rounding** [Beznosikov et al., 2020] | ✓ | ✓ | ✗ | ✓ | ✗ | ✓ | ✗ | ✓ | ✓ |
| **Natural compression** [Horváth et al., 2022] | ✓ | ✓ | ✓ | ✓ | ✗ | ✓ | ✗ | ✓ | ✓ |
| **Scaled integer rounding** [Sapio et al., 2021] | ✓ | ✓ | ✗ | ✓ | ✓ | ✓ | ✓ | ✓ | ✓ |

Table 3: Coverage of popular estimators by known frameworks. For more estimators, see Table 9.

## 6.1 General nonconvex case

**Theorem 3** *Let Assumptions 0 and 9 hold. Let $\delta^0 \overset{def}{=} f(x^0) - f^*$, and choose the stepsize such that $0 < \gamma \leq \frac{b}{LB}$. Then the iterates $\{x^t\}_{t \geq 0}$ of* BiasedSGD *(Algorithm* (1)*) satisfy*

$$\min_{0 \leq t \leq T-1} \mathbb{E}\left[\left\|\nabla f(x^t)\right\|^2\right] \leq \frac{2\left(1 + LA\gamma^2\right)^T}{b\gamma T}\delta^0 + \frac{LC\gamma}{b} + \frac{c}{b}. \tag{20}$$

While one can notice the possibility of an exponential blow-up in (20), by carefully controlling the stepsize we still can guarantee the convergence of BiasedSGD. In Corollaries 5 and 6 (see the appendix) we retrieve the results of Theorem 2 and Corollary 1 from [Khaled and Richtárik, 2023] for the unbiased case. In Corollary 7 (see the appendix) we retrieve the result that is worse than that in [Ajalloeian and Stich, 2020, Theorem 4] by a multiplicative factor and an extra additive term, but under milder conditions (cf. Biased ABC and BND in Figure 1; see also Claim 1). If we set $A = c = 0$, we recover the result of [Bottou et al., 2018, Theorem 4.8] (see Corollary 8 in the appendix).

## 6.2 Convergence under PŁ-condition

One of the popular generalizations of strong convexity in the literature is the Polyak–Łojasiewicz assumption [Polyak, 1963; Karimi et al., 2016; Lei et al., 2019]. First, we define this condition.

**Assumption 10 (Polyak–Łojasiewicz)** *There exists $\mu > 0$ such that $\|\nabla f(x)\|^2 \geq 2\mu\left(f(x) - f^*\right)$, for all $x \in \mathbb{R}^d$.*

We now formulate a theorem that establishes the convergence of BiasedSGD for functions satisfying this assumption and Assumption 9.

**Theorem 4** *Let Assumptions 0, 9 and 10 hold. Choose a stepsize such that*

$$0 < \gamma < \min\left\{\frac{\mu b}{L(A + \mu B)}, \frac{1}{\mu b}\right\}. \tag{21}$$

*Letting $\delta^0 \overset{def}{=} f(x^0) - f^*$, for every $T \geq 1$, we have*

$$\mathbb{E}\left[f(x^T) - f^*\right] \leq (1 - \gamma\mu b)^T \delta^0 + \frac{LC\gamma}{2\mu b} + \frac{c}{\mu b}. \tag{22}$$

When $c = 0$, the last term in (22) disappears, and we recover the best known rates under the Polyak–Łojasiewicz condition [Karimi et al., 2016], but under milder conditions (see Corollary 10 in the appendix). Further, if we set $A = 0$, we obtain a result that is slightly weaker than the one obtained by Ajalloeian and Stich [2020, Theorem 6], but under milder assumptions (cf. Biased ABC and BND in Figure 1; see also Claim 1).

## 6.3 Strongly convex case

**Assumption 11** *Let $f$ be $\mu$-strongly-convex and continuously differentiable.*

| Theorem | Convergence rate | Compared to | Rate we compare to | Match? |
|---|---|---|---|---|
| Thm 3 | $\mathcal{O}\left(\frac{\delta^0 L}{\varepsilon^2}\max\left\{B, \frac{12\delta^0 A}{\varepsilon^2}, \frac{2C}{\varepsilon^2}\right\}\right)$ | 34-Thm 2 | $\mathcal{O}\left(\frac{\delta^0 L}{\varepsilon^2}\max\left\{B, \frac{12\delta^0 A}{\varepsilon^2}, \frac{2C}{\varepsilon^2}\right\}\right)$ | ✓ |
| Thm 3 | $\mathcal{O}\left(\max\left\{\frac{8(M+1)(m+1)}{(1-m)^2\varepsilon}, \frac{16(M+1)\varphi^2+2\sigma^2}{(1-m)^2\varepsilon^2}\right\}L\delta^0\right)$ | 1-Thm 4 | $\mathcal{O}\left(\max\left\{\frac{M+1}{(1-m)\varepsilon}, \frac{2\sigma^2}{(1-m)^2\varepsilon^2}\right\}L\delta^0\right)$ | ✗ |
| Thm 3 | $\mathcal{O}\left(\max\left\{\frac{8Q}{\varepsilon^2 q^2}, \frac{4(U+u^2)}{\varepsilon q^2}\right\}L\delta^0\right)$ | 6-Thm 4.8 | $\mathcal{O}\left(\max\left\{\frac{8Q}{\varepsilon^2 q^2}, \frac{4(U+u^2)}{\varepsilon q^2}\right\}L\delta^0\right)$ | ✓ |
| Thm 4 | $\tilde{\mathcal{O}}\left(\max\left\{\frac{2(M+1)(m+1)}{1-m}, \frac{2(M+1)\varphi^2+\sigma^2}{\epsilon\mu(1-m)+2\varphi^2}\right\}\frac{\kappa}{1-m}\right)$ | 1-Thm 6 | $\tilde{\mathcal{O}}\left(\max\left\{(M+1), \frac{\sigma^2}{\varepsilon\mu(1-m)+\varphi^2}\right\}\frac{\kappa}{1-m}\right)$ | ✗ |
| Thm 12 | $\tilde{\mathcal{O}}\left(\max\left\{2, \frac{L(U+u^2)}{q^2\mu}, \frac{LQ}{\varepsilon\mu^2 q^2}\right\}\right)$ | 6-Thm 4.6 | $\tilde{\mathcal{O}}\left(\max\left\{2, \frac{L\left(U+u^2\right)}{q^2\mu}, \frac{LQ}{\varepsilon\mu^2 q^2}\right\}\right)$ | ✓ |
| Thm 12 | $\tilde{\mathcal{O}}\left(\left(\frac{\beta^2}{\alpha}\right)^2\frac{L}{\mu}\right)$ | 5-Thm 12 | $\tilde{\mathcal{O}}\left(\frac{\beta^2}{\alpha}\frac{L}{\mu}\right)$ | ✗ |
| Thm 12 | $\tilde{\mathcal{O}}\left(\left(\frac{\beta}{\tau}\right)^2\frac{L}{\mu}\right)$ | 5-Thm 13 | $\tilde{\mathcal{O}}\left(\frac{\beta}{\tau}\frac{L}{\mu}\right)$ | ✗ |
| Thm 12 | $\tilde{\mathcal{O}}\left(\delta^2\frac{L}{\mu}\right)$ | 5-Thm 14 | $\tilde{\mathcal{O}}\left(\delta\frac{L}{\mu}\right)$ | ✗ |

Table 4: Complexity comparison. We examine whether we can achieve the same convergence rate as obtained under stronger assumptions. In most cases, we ensure the same rate, albeit with inferior multiplicative factors due to the broader scope of the analysis. The notation $\tilde{\mathcal{O}}\left(\cdot\right)$ hides a logarithmic factor of $\log\frac{2\delta^0}{\varepsilon}$.

Since Assumption 10 is more general than Assumption 11, Theorem 4 can be applied to functions that satisfy Assumption 11. If we set $A = c = 0$, we recover [Bottou et al., 2018, Theorem 4.6] (see Corollary 13 in the appendix). If $A = C = c = 0$, we retrieve results comparable to those in [Beznosikov et al., 2020, Theorems 12–14], up to a multiplicative factor (see Corollary 14 in the appendix). Due to $\mu$-strong convexity, our result (22) also implies an iterate convergence, since we have $\left\|x^T - x^*\right\|^2 \leq \frac{2}{\mu}\mathbb{E}\left[f(x^T) - f(x^*)\right]$. However, in this case an additional factor of $\frac{2}{\mu}$ arises. Below we present a stronger result, yet, at a cost of imposing a stricter condition on the control variables from Assumption 9.

**Assumption 12** *Let $A, B, C$ and $b$ be parameters from Assumption 9. Let $\mu$ be a strong convexity constant. Let $L$ be a smoothness constant. Suppose $A + L(B + 1 - 2b) < \mu$ holds.*

Under Assumptions 9 and 12 we establish a similar result as the one obtained by Hu et al. [2021a, Theorem 1]. The authors impose a restriction of $\frac{1}{\kappa}$ from above on a constant with an analogous role as $B + 1 - 2b$ in Assumptions 9 and 12 with $A = 0$. However, unlike us, the authors consider only a finite sum case which makes our result more general. Moreover, only a biased version of SGD with a simple sampling strategy is analyzed by Hu et al. [2021a]. Our results are applicable to a larger variety of gradient estimators and obtained under milder assumptions. Also, our proof strategy is different, and much simpler.

**Theorem 5** *Let Assumptions 0, 9, 11 and 12 hold. For every positive $s$, satisfying $A+L(B+1-2b) < s < \mu$, choose a stepsize $\gamma$ such that*

$$0 < \gamma \leq \min\left\{\frac{1 - \frac{1}{s}\left(A + L\left(B + 1 - 2b\right)\right)}{A + LB}, \frac{1}{\mu - s}\right\}. \tag{23}$$

*Then the iterates of* BiasedSGD *(Algorithm 1) for every $T \geq 1$ satisfy*

$$\mathbb{E}\left[\left\|x^T - x^*\right\|^2\right] \leq \left(1 - \gamma\left(\mu - s\right)\right)^T\left\|x^0 - x^*\right\|^2 + \frac{\gamma C + \frac{C+2c}{s}}{\mu - s}. \tag{24}$$

In the standard result for (unbiased) SGD, the convergence neighborhood term has the form of $\frac{\gamma C}{\mu}$, and it can be controlled by adjusting the stepsize. However, due to the generality of our analysis in the biased case, in (24) we obtain an extra uncontrollable neighborhood term of the form $\frac{C+2c}{s(\mu-s)}$.

When $A = C = c = 0$, $B = 1$, $b = 1$, $s \to 0$, we recover exactly the classical result for GD.

## 7 Experiments

**Datasets, Hardware, and Code Implementation.** The experiments utilized publicly available LibSVM datasets Chang and Lin [2011], specifically the splice, a9a, and w8a. These algorithms were developed using Python 3.8 and executed on a machine equipped with 48 cores of Intel(R) Xeon(R) Gold 6246 CPU @ 3.30GHz.

**Experiment: Problem Setting.** To validate our theoretical findings, we conducted a series of numerical experiments on a binary classification problem. Specifically, we employed logistic regression with a non-convex regularizer:

$$\min_{x \in \mathbb{R}^d} \left[ f(x) \stackrel{\text{def}}{=} \frac{1}{n} \sum_{i=1}^n f_i(x) \right], \text{ where } f_i(x) \stackrel{\text{def}}{=} \log\left(1 + \exp\left(-y_i a_i^\top x\right)\right) + \lambda \sum_{j=1}^d \frac{x_j^2}{1 + x_j^2},$$

and $(a_i, y_i) \in \mathbb{R}^d \times \{-1, 1\}, i = 1, \ldots, n$ represent the training data samples. In all experiments, we set the regularization parameter $\lambda$ to a fixed value of $\lambda = 1$. We use datasets from the open LibSVM library [Chang and Lin, 2011]. We examine the performance of the proposed BiasedSGD method with biased independent sampling without replacement (we call it BiasedSGD-ind) in various settings (see Definition 1). The primary goal of these numerical experiments is to demonstrate the alignment of our theoretical findings with the observed experimental results. To assess the performance of the methods throughout the optimization process, we monitor the metric $\|\nabla f(x^t)\|^2$, recomputed after every 10 iterations. The algorithms are terminated after completing 5000 iterations. For each method, we use the largest theoretical stepsize. Specifically, for BiasedSGD-ind, the stepsize is determined according to Corollary 4 and Claim 2 with $\gamma = \min\left\{ \frac{1}{\sqrt{LAK}}, \frac{b}{LB}, \frac{c}{LC} \right\}$, where $c = 0$, $A = \frac{\max_i L_i}{\min_i p_i}$, $B = 0$, $C = 2A\Delta^* + s^2$, $b = \min_i p_i$ and $s = 0$.

More experimental details are provided in Appendix A.

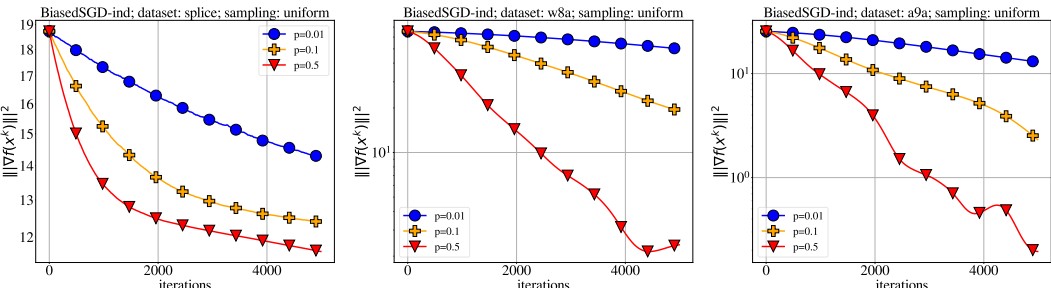

Figure 2: The performance of BiasedSGD-ind with different choices of probabilities.

**Experiment: The impact of the parameter $p$ on the convergence behavior.** In the first experiment, we investigate how the convergence of BiasedSGD-ind is affected as we increase the probabilities $p_i$, while keeping them equal for all data samples. According to the Corollary 4, larger $p_i$ values (resulting in an increase of the expected batch size) allow for a larger stepsize, which, in turn, improves the overall convergence. This behavior is evident in Figure 2. The experiment visualized in Figure 2 involves varying the probability parameter $p$ within the set $\{0.01, 0.1, 0.5\}$. This manipulation directly influences the value of $A$, consequently affecting the theoretical stepsize $\gamma$. In the context of BiasedSGD-ind, the stepsize $\gamma$ is defined as $\frac{1}{\sqrt{LAK}}$. A comprehensive compilation of these parameters is represented in Table 7.

## 8 Conclusion

In this work, we consolidate various recent assumptions regarding the convergence of biasedSGD and elucidate their implication relationships. Moreover, we introduce a weaker assumption, referred to as Biased ABC. We also demonstrate that Biased ABC encompasses stochastic gradient oracles that previous assumptions excluded. With this assumption, we provide a proof of biasedSGD convergence across multiple scenarios, including strongly convex, non-convex, and under the PŁ-condition. Convergence rates that we obtain are the same up to a constant factor due to the broader setting and in some cases they coincide with the rates obtained under stricter assumptions. Furthermore, we examine the most widely used estimators in the literature related to SGD, represent them within the context of our Biased ABC framework, and analyze their compatibility with all previous frameworks.

## Acknowledgements

This work of all authors was supported by the KAUST Baseline Research Scheme (KAUST BRF). The work of Y. Demidovich and P. Richtárik was supported by the KAUST Extreme Computing Research Center (KAUST ECRC), and the work of P. Richtárik was supported by the SDAIA-KAUST Center of Excellence in Data Science and Artificial Intelligence (SDAIA-KAUST AI).

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
