# Contents

## A Experiments: missing details

This section completes the experimental details mentioned in Section 7. The corresponding code can be found in the provided repository: `https://github.com/IgorSokoloff/guide-[]biased-[]sgd-[]experiments`. A summarized description of the datasets is available in Table 5.

Table 5: Summary of the datasets

| Dataset | $n$ (dataset size) | $d$ (# of features) |
|---------|--------------------|--------------------|
| splice  | 1000               | 60                 |
| a9a     | 32560              | 123                |
| w8a     | 49749              | 300                |

**Hyperparameters.** For the selected logistic regression problem, the smoothness constants $L$ and $L_i$ of the functions $f$ and $f_i$ were explicitly calculated as shown below:

$$L = \lambda_{max}\left(\frac{1}{4m}\mathbf{A}^\top \mathbf{A} + 2\lambda \mathbf{I}\right)$$

$$L_i = \lambda_{\max}\left(\frac{1}{4}a_i a_i^\top + 2\lambda \mathbf{I}\right).$$

In the above equations, $\mathbf{A}$ represents the dataset (data matrix), and $a_i$ signifies its $i$-th row. Smoothness constants for the logistic regression objective on the selected datasets are presented in Table 6.

Table 6: Smoothness Constants for Logistic Regression with $\lambda = 1$

| Dataset | $L$   | $L_{\max}$ |
|---------|-------|-----------|
| w8a     | 1.66  | 29.5      |
| a9a     | 2.57  | 4.5       |
| splice  | 97.83 | 163.25    |

Each method utilized the largest possible theoretical stepsize. For the BiasedSGD-ind method, the stepsize is determined based on Corollary 4 and Claim 2 with $\gamma = \min\left\{\frac{1}{\sqrt{LAK}}, \frac{b}{LB}, \frac{c}{LC}\right\}$, where $c = 0$, $A = \frac{\max_i L_i}{\min_i p_i}$, $B = 0$, $C = 2A\Delta^* + s^2$, $b = \min_i p_i$ and $s = 0$.

**Experiment: The impact of the parameter $p$ on the convergence behavior (extra details).** The experiment visualized in Figure 2 involves varying the probability parameter $p$ within the set $\{0.01, 0.1, 0.5\}$. This manipulation directly influences the value of $A$, consequently affecting the theoretical stepsize $\gamma$. In the context of BiasedSGD-ind, the stepsize $\gamma$ is defined as $\frac{1}{\sqrt{LAK}}$. A comprehensive compilation of these parameters is represented in Table 7.

## B Sources of bias: further discussion and new estimators

In Section 2 of the main part of the paper we describe different sources of bias and provide general forms of estimators that arise in each scenario. However, we do not present any concrete practical examples of stochastic gradients. In this section we define several important realistic estimators and characterize them in terms of Biased ABC framework. For proofs of results in this section, see Section I.

For a finite-sum problem 1, consider a setting when the bias is induced by a subsampling strategy of which we lack the information. Let us introduce (without aiming to be exhaustive) a specific (and practical) sampling distribution and an estimator, which satisfies Assumption 9.

Table 7: Parameters $A$ and theoretical stepsizes, determined by the choice of parameter $p$ and dataset

| Dataset | $p$ | $A$ | Theoretical stepsize for BiasedSGD-ind $\gamma = \min \frac{1}{\sqrt{LAK}}$ |
|---------|-----|-----|---------------------------------------|
| splice | 0.01 | 16325.0 | $3.54 \cdot 10^{-4}$ |
|        | 0.1 | 1632.5 | $1.12 \cdot 10^{-3}$ |
|        | 0.5 | 326.5 | $2.50 \cdot 10^{-3}$ |
| a9a | 0.01 | 550.0 | $1.01 \cdot 10^{-2}$ |
|     | 0.1 | 55.0 | $3.19 \cdot 10^{-2}$ |
|     | 0.5 | 11.0 | $7.13 \cdot 10^{-2}$ |
| w8a | 0.01 | 3050.0 | $4.96 \cdot 10^{-3}$ |
|     | 0.1 | 305.0 | $1.57 \cdot 10^{-2}$ |
|     | 0.5 | 61.0 | $3.51 \cdot 10^{-2}$ |

**Definition 1 (Biased independent sampling without replacement)** *Let $p_1, p_2, \ldots, p_n$ be probabilities, $0 < p_i \leq 1$ for all $i \in [n]$, $\sum_{i=1}^{n} p_i \in (0, n]$. For every $i \in [n]$, define a random set as follows:*

$$S_i = \begin{cases} \{i\} & \text{with probability } p_i, \\ \varnothing & \text{with probability } 1 - p_i. \end{cases}$$

*Define a random subset $S \subseteq [n]$ by taking the union of these random sets: $S \stackrel{def}{=} \bigcup_{i=1}^{n} S_i$. Put*

$$\mathbb{I}_{i \in S} = \begin{cases} 1, & i \in S, \\ 0, & \text{otherwise}. \end{cases} \tag{25}$$

*For every $i \in [n]$, define $v_i = \frac{\mathbb{I}_{i \in S}}{|S|}$. Let $g(x) = \tilde{g}(x) + X$, where*

$$\tilde{g}(x) = \frac{1}{|S|} \sum_{i=1}^{n} \mathbb{I}_i \nabla f_i(x),$$

*and $X$ is a random variable independent of $S$, such that $\mathbb{E}[X] = 0$, $\mathbb{V}[X] = s^2$.*

The practical setting where this stochastic gradient might be useful can have the following structure. There is an oracle that, for every $i \in [n]$, decides with an unknown probability $p_i$ whether to provide the information of $\nabla f_i$ at the iteration $k$ or not. Since the probabilities $p_i$ are unknown, they may be substituted for their estimators $\mathbb{I}_i$. The stochastic gradient is then calculated as a simple average of all gradients with these estimators as weights. Note that a setting with $\sum_{i=1}^{n} p_i = 1$ corresponds to the single-machine setup.

The subsampling strategy from Definition 1 can be used in another practical scenario. Consider a situation where access to the entire dataset is not available. In such cases, a *fixed batch strategy* can be employed. This strategy involves sampling a single batch $S$ at step $0$ and subsequently using it throughout the entire optimization process.

In the proof of Theorem 2 (parts viii and ix), we demonstrate that in a very simple setting the stochastic gradient from Definition 1 does not satisfy Assumptions 6 and 8 (and, therefore, to any other assumption from Section 4). We want to show that under very mild restrictions on functions $f_i$, $g(x)$ satisfies Biased ABC assumption.

**Assumption 13** *Each $f_i$ is bounded from below by $f_i^*$ and $L_i$-smooth. That is, for all $x, y \in \mathbb{R}^d$, we have*

$$f_i(y) \leq f_i(x) + \langle \nabla f_i(x), y - x \rangle + \frac{L_i}{2} \|y - x\|^2.$$

Here and many times below in the paper we rely on the following important lemma.

**Lemma 1** *Let $f$ be a function for which Assumption 0 is satisfied. Then, for all $x \in \mathbb{R}^d$, we have*

$$\|\nabla f(x)\|^2 \leq 2L D_f(x, x^*).$$

In the nonconvex case the expression takes the following form:

$$\|\nabla f(x)\|^2 \le 2L\left(f(x) - f^*\right), \quad \forall x \in \mathbb{R}^d.$$

This lemma appears in [Khaled and Richtárik, 2023] and in several recent works on the convergence of SGD. We give its proof in Sectioin P. Equipped with Lemma 1, we can prove the following claim that motivates the inclusion of a Bregman Divergence term in (19). The reason why biased sampling gradient estimator does not satisfy Assumptions 1, 6 and 8 is because its variance contains a sum of squared client gradient norms, which, in general, can not be bounded in terms of the squared norm of the full gradient. In fact, for a variety of biased sampling estimators this obstacle may occur, and this additionally motivates establishing new theory under the general assumption proposed in the present paper.

**Claim 2** *Suppose Assumptions 0 and 13 hold. Let*

$$\Delta^* \stackrel{def}{=} \frac{1}{n} \sum_{i=1}^{n} \left(f^* - f_i^*\right). \tag{26}$$

*Then, gradient estimator from Definitioin 1 satisfies Assumption 9 with* $b = \min_i \{p_i\}$, $c = 0$,

$$A = \frac{\max_i\{L_i\}}{\min_i p_i}, \quad B = 0, \quad C = 2A\Delta^* + s^2.$$

In [Khaled and Richtárik, 2023], for a finite-sum problem (1), in the unbiased case the following general stochastic gradient is considered. Given a sampling vector $v \in \mathbb{R}^d$ drawn from some distribution $\mathcal{D}$ (where a sampling vector is one such that $\mathbb{E}_{\mathcal{D}}[v_i] = c_i$, $c_i \ge 0$, for all $i \in [n]$), for $x \in \mathbb{R}^d$, define the stochastic gradient $g(x) \stackrel{def}{=} \frac{1}{n} \sum_{i=1}^{n} v_i \nabla f_i(x)$. We do not require $v_i$ to cause unbiasedness. Under mild assumptions on functions $f_i$ and the sampling vectors $v_i$, we prove that $g(x)$ satisfies Biased ABC assumption, for all non-degenerate distributions $\mathcal{D}$.

**Claim 3** *Suppose Assumption 13 holds and, for all $i \in [n]$, we have $\mathbb{E}\left[v_i^2\right] < \infty$. Then Assumption 9 holds for $g(x)$ with $A = \max_i \left\{L_i \mathbb{E}\left[v_i^2\right]\right\}$, $B = 0$, $C = 2A\Delta^*$, $b = \min_i \{c_i\}$, $c = 0$.*

Note, that in [Khaled and Richtárik, 2023, Proposition 2] it is proven that $\Delta^* \ge 0$. The requirement of $\mathbb{E}\left[v_i^2\right] < \infty$ is very weak and satisfied for almost all practical subsampling schemes in the literature. However, the generality of Claim 3 comes at a cost since it leads to very pessimistic choices of constants in Assumption 9.

Our framework is general enough to establish the convergence of biased stochastic gradient quantization or compression schemes. Consider the finite-sum problem (1) and let us propose the following new practical biased gradient estimator.

**Definition 2 (Distributed general biased rounding)** *Let $\{a_k\}_{k \in \mathbb{Z}}$ be an arbitrary increasing sequence of positive numbers such that $\inf_k\{a_k\} = 0$, and $\sup_k\{a_k\} = \infty$. Then, for all $j \in [n]$, $i \in [d]$, define*

$$\tilde{g}_j(x)_i \stackrel{def}{=} \mathrm{sign}\left(\nabla f(x)_i\right) \arg \min_{y \in \{a_k\}} |y - |\nabla f(x)_i||, \quad i \in [d].$$

*For every $j \in [n]$, define mutually independent random variables*

$$\mathbb{I}_j = \begin{cases} 1, & \text{with probability } 0 < p_j < 1, \\ 0, & \text{with probability } 1 - p_j. \end{cases}$$

*For every $x \in \mathbb{R}^d$, define a gradient estimator*

$$g(x) = \frac{1}{n} \sum_{j=1}^{n} \left(\mathbb{I}_j \tilde{g}_j(x) + (1 - \mathbb{I}_j) \nabla f_j(x)\right).$$

The practical setting where $g(x)$ might be used is a distributed problem where client node $j \in [n]$ decides with probability $p_j$ whether to send the compressed gradient or not. Master nodes which

does not know $p_j$ simply averages the received stochastic gradients. In this case we preserve more information in comparison to the setting when we use compression at every step. On the other hand, gradients are compressed with positive probability, and we diminish the communication complexity versus the setting without any compression. That is, we have a flexible setting which is useful in practice.

As before, we prove that $g(x)$ satisfies Biased ABC assumptioin under very mild conditions.

**Claim 4** *Suppose Assumption 13 holds and, for all $i \in [n]$, we have $\mathbb{E}\left[v_i^2\right] < \infty$. Then the distributed general biased rounding estimator $g(x)$ satisfies Assumption 9 with*

$$A = A_r \stackrel{def}{=} \frac{2}{n} \max_j \{L_j\} \max_j \{p_j(1-p_j)\} \left( \left( \sup_{k \in \mathbb{Z}} \frac{2a_{k+1}}{a_k + a_{k+1}} \right)^2 + 1 \right), \qquad (27)$$

$$B = B_r \stackrel{def}{=} 2 \max_j \{p_j^2\} \left( \left( \sup_{k \in \mathbb{Z}} \frac{2a_{k+1}}{a_k + a_{k+1}} \right)^2 + 1 \right), \qquad (28)$$

$$C = C_r \stackrel{def}{=} \frac{4}{n} \max_j \{L_j\} \max_j \{p_j(1-p_j)\} \left( \left( \sup_{k \in \mathbb{Z}} \frac{2a_{k+1}}{a_k + a_{k+1}} \right)^2 + 1 \right) \Delta^*, \qquad (29)$$

$$b = b_r \stackrel{def}{=} \max_j \{p_j\} \cdot \inf_{k \in \mathbb{Z}} \frac{2a_k}{a_k + a_{k+1}} + \max_j \{1 - p_j\} \qquad (30)$$

$$c = c_r = 0. \qquad (31)$$

From Claims 2, 3 and 4 we see that, in fact, Biased ABC is not an additional assumption, but an inequality that is automatically satisfied under such settings.

One of the simplest models of bias is the case of additive noise, that is

$$g(x) = \nabla f(x) + \mathcal{Z},$$

where $\mathcal{Z}$ is a random variable satisfying $\mathbb{E}\left[\mathcal{Z}\right] = a$, $a \in \mathbb{R}^d$, $\mathbb{E}\left[\|\mathcal{Z}\|^2\right] = \sigma^2$, $\sigma \in \mathbb{R}$. It may happen in practise that, e.g., during the communication process in the distributed setting of the finite-sum problem (1) transmitted gradients become noisy, and this simple model captures such a scenario. Models of this type were previously analyzed in [Ajalloeian and Stich, 2020]. Clearly, BND assumption is satisfied. It means (see Figure 1), that they are covered by Biased ABC framework as well. However, models of this type impose rather strong restrictions on the stochastic gradient: they fail to capture a multiplicative biased noise that arises in the case of gradient compression operators and are not suitable for simulating subsampling schemes.

## C   Known gradient estimators in biased ABC framework

In this section we define several known biased gradient estimators and for each of them, we present values of control variables $A, B, C, b, c$ within our Biased ABC framework. Also, these values are shown in Table 8 for convenience of the reader. Formal proofs can be found in Section J. In Table 9 we demonstrate a summary on inclusioin of each estimator from this section into every framework from Section 4.

**Definition 3 (Top-$k$ sparsifier – Aji and Heafield [2017]; Alistarh et al. [2018])** *Let gradient estimator $g(x)$ be defined as*

$$g(x) \stackrel{def}{=} \sum_{i=d-k+1}^{d} (\nabla f(x))_{(i)} e_{(i)}, \quad \forall x \in \mathbb{R}^d,$$

*where coordinates are ordered with respect to their absolute values:*

$$|(\nabla f(x))_{(1)}| \leq |(\nabla f(x))_{(2)}| \leq \ldots \leq |(\nabla f(x))_{(d)}|.$$

**Claim 5** *Top-k sparsifier $g(x)$ satisfies Assumption 9 with $b = \frac{k}{d}$, $c = 0$, $A = 0$, $B = 1$, $C = 0$.*

**Definition 4 (Rand-$k$ – Stich et al. [2018])** *For every $x \in \mathbb{R}^d$, let*

$$g(x) \overset{def}{=} \frac{d}{k} \sum_{i \in S} (\nabla f(x))_i \, e_i,$$

*where $S$ is a random subset of $[d]$ chosen uniformly.*

**Claim 6** *Rand-k estimator $g(x)$ satisfies Assumption 9 with $A = 0$, $B = \frac{d}{k}$, $C = 0$, $b = 1$, $c = 0$.*

**Definition 5 (Biased Rand-$k$ sparsifier – Beznosikov et al. [2020])** *For every $x \in \mathbb{R}^d$, let*

$$g(x) \overset{def}{=} \sum_{i \in S} (\nabla f(x))_i \, e_i,$$

*where $S$ is a random subset of $[d]$ chosen uniformly.*

**Claim 7** *Biased Rand-k sparsifier $g(x)$ satisfies Assumption 9 with $b = \frac{k^2}{d^2}$, $c = 0$, $A = C = 0$, $B = \frac{k}{d}$.*

**Definition 6 (Adaptive random sparsification – Beznosikov et al. [2020])** *Adaptive random sparsification estimator is defined via*

$$g(x) \overset{def}{=} (\nabla f(x))_i \, e_i \quad \text{with probability} \quad \frac{|(\nabla f(x))_i|}{\|\nabla f(x)\|_1}$$

**Claim 8** *Adaptive random sparsifier $g(x)$ satisfies Assumption 9 with $A = C = c = 0$, $B = 1$, $b = \frac{1}{d}$.*

**Definition 7 (General unbiased rounding estimator – Beznosikov et al. [2020])** *Let $\{a_k\}_{k \in \mathbb{Z}}$ be an arbitrary increasing sequence of positive numbers such that $\inf_k a_k = 0$, $\sup_k a_k = \infty$. Define the rounding estimator $g(x)$ in the following way: if $a_k \leq |\nabla f(x)_i| \leq a_{k+1}$, for a coordinate $i \in [d]$, then*

$$g(x)_i = \begin{cases} \text{sign}(\nabla f(x)_i) a_k, & \text{with probability } \frac{a_{k+1} - |\nabla f(x)_i|}{a_{k+1} - a_k}, \\ \text{sign}(\nabla f(x)_i) a_{k+1}, & \text{with probability } \frac{|\nabla f(x)_i| - a_k}{a_{k+1} - a_k}. \end{cases}$$

Put

$$Z \overset{def}{=} \sup_{k \in \mathbb{Z}} \left( \frac{a_k}{a_{k+1}} + \frac{a_{k+1}}{a_k} + 2 \right). \tag{32}$$

**Claim 9** *General unbiased rounding estimator $g(x)$ satisfies Assumption 9 with $A = C = c = 0$, $B = \frac{Z}{4}$, $b = 1$.*

**Definition 8 (General biased rounding – Beznosikov et al. [2020])** *Let $(a_k)_{k \in \mathbb{Z}}$ be an arbitrary increasing sequence of positive numbers such that $\inf a_k = 0$ and $\sup a_k = \infty$. Then general biased rounding is defined via*

$$g(x)_i \overset{def}{=} \text{sign}((\nabla f(x))_i) \arg \min_{t \in (a_k)} |t - |(\nabla f(x))_i||, \quad i \in [d].$$

Put

$$F = \sup_{k \in \mathbb{Z}} \frac{2a_{k+1}}{a_k + a_{k+1}}, \; G = \inf_{k \in \mathbb{Z}} \frac{2a_k}{a_k + a_{k+1}}. \tag{33}$$

**Claim 10** *General biased rounding estimator $g(x)$ satisfies Assumption 9 with $A = C = c = 0$, $B = F^2$, $b = \frac{G^2}{F}$.*

**Definition 9 (Natural compression – Horváth et al. [2022])** *Natural compression estimator $g_{nat}(x)$ is the special case of general unbiased rounding operator (see Definition 7) when $a_k = 2^k$, $k \in \mathbb{N}$.*

**Claim 11** *Natural compression estimator $g(x)$ satisfies Assumption 9 with $A = C = c = 0$, $B = \frac{9}{8}$, $b = 1$.*

**Definition 10 (General exponential dithering – Beznosikov et al. [2020])** *For $a > 1$, define general exponential dithering estimator with respect to $\ell_p$-norm and with $s$ exponential levels $0 < a^{1-s} < a^{2-s} < \cdots < a^{-1} < 1$ via*

$$(g(x))_i \stackrel{def}{=} \|\nabla f(x)\|_p \times \mathrm{sign}\left((\nabla f(x))_i\right) \times \xi\left(\frac{|(\nabla f(x))_i|}{\|\nabla f(x)\|_p}\right),$$

*where the random variable $\xi(t)$ for $t \in \left[a^{-u-1}, a^{-u}\right]$ is set to either $a^{-u-1}$ or $a^{-u}$ with probabilities proportional to $a^{-u} - t$ and $t - a^{-u-1}$, respectively.*

Put $r = \min(p, 2)$ and

$$H_a = \frac{1}{4}\left(a + \frac{1}{a} + 2\right) + d^{\frac{1}{r}} a^{1-s} \min\left(1, d^{\frac{1}{r}} a^{1-s}\right) \tag{34}$$

**Claim 12** *General exponential dithering estimator $g(x)$ satisfies Assumption 9 with $A = C = c = 0$, $B = H_a$, $b = 1$, where $H_a$ is defined in (34).*

**Definition 11 (Natural dithering – Horváth et al. [2022])** *Natural dithering without norm compression is the special case of general exponential dithering when $a = 2$ (see Definition 10).*

**Claim 13** *Natural dithering estimator satisfies Assumption 9 with $A = C = c = 0$, $B = H_2$, $b = 1$.*

**Definition 12 (Composition of Top-$k$ with exponential dithering – Beznosikov et al. [2020])**
*Let $g_{top}(x)$ be the Top-$k$ sparsification operator (see Definition 3) and $g_{dith}(x)$ be general exponential dithering operator with some base $a > 1$ and parameter $H_a$ from (34). Define a new compression operator as the composition of these two:*

$$g(x) \stackrel{def}{=} g_{dith}\left(g_{top}(x)\right).$$

In this definition we imply that the dithering operator is applied to the vector yielded after Top-$k$ sparsification, not to the gradient as it was defined.

**Claim 14** *Composition of Top-$k$ with exponential dithering estimator $g(x)$ satisfies Assumption 9 with $A = C = c = 0$, $B = H_a^2$, $b = \frac{k}{dH_a}$.*

**Definition 13 (Gaussian smoothing – Polyak [1987])** *The following zero-order stochastic gradient, which we call Gaussian smoothing as in [Ajalloeian and Stich, 2020], is defined as*

$$g_{GS}(x) = \frac{f(x + \tau z) - f(x)}{\tau} \cdot z,$$

*where $\tau > 0$ is a smoothing parameter, and $z \sim \mathcal{N}\left(0, I\right)$ is a random Gaussian vector.*

**Claim 15** *Gaussian smoothing estimator $g(x)$ satisfies Assumption 9 with*

$$A = A_{GS} \stackrel{def}{=} 0, \ B = B_{GS} \stackrel{def}{=} 2(d + 4), \ C = C_{GS} \stackrel{def}{=} \frac{\tau^2}{2} L^2 (d + 6)^3,$$

$$b = b_{GS} = \frac{1}{2}, \ c = c_{GS} \stackrel{def}{=} \frac{\tau^2}{8} L^2 (d + 3)^3. \tag{35}$$

**Definition 14 (Hard-threshold sparsifier – Sahu et al. [2021] )** *For some $w \geq 0$, define the estimator $g_{HT}^w(x)$ as*

$$(g_{HT}^w(x))_i = \begin{cases} (\nabla f(x))_i, & |(\nabla f(x))_i| \geq w, \\ 0, & \text{otherwise}, \end{cases}$$

*for every $i \in [d]$.*

**Claim 16** *Hard-threshold estimator sastisfies Assumption 9 with $A = C = 0$, $B = 1$, $b = 1$, $c = w^2 d$.*

**Definition 15 (Scaled integer rounding – Sapio et al. [2021])** *In a distributed setting (2), for every $i \in [n]$, let $\mathcal{C}_i : \nabla f_i(x) \to \frac{1}{\chi} R\left(\chi \nabla f_i(x)\right)$, where $\chi > 0$ is a scaling factor, $R$ is a rounding to the nearest integer operator. That is, a scaling integer rounding estimator is defined as*

$$g(x) = \frac{1}{n} \sum_{i=1}^{n} \frac{1}{\chi} R\left(\chi \nabla f_i(x)\right).$$

**Claim 17** *Scaling integer estimator satisfies Assumption 9 with $A = 0$, $B = 2$, $C = \frac{2d}{\chi^2}$, $b = \frac{1}{2}$, $c = \frac{d}{2\chi^2}$.*

**Definition 16 (Biased dithering – Khirirat et al. [2018b])** *Biased dithering estimator $g(x)$ is defined as*

$$(g(x))_i = \|\nabla f(x)\| \operatorname{sign}\left((\nabla f(x))_i\right), \quad i \in [d], \quad \forall x \in \mathbb{R}^d.$$

**Claim 18** *Biased dithering operator satisfies Assumption 9 with $A = 0$, $B = d$, $C = 0$, $b = 1$, $c = 0$.*

**Definition 17 (Sign compression – [Karimireddy et al., 2019])** *Sign compression operator is defined as*

$$g(x) \stackrel{def}{=} \frac{\|\nabla f(x)\|_1}{d} \operatorname{sign}\left(\nabla f(x)\right), \quad \forall x \in \mathbb{R}^d.$$

**Claim 19** *Sign compression operator satisfies Assumption 9 with $A = C = c = 0$, $B = 2\left(2 - \frac{1}{d}\right)$, $b = \frac{1}{2d}$.*

**Definition 18 (Composition of sampling and $0$-order estimator – Leluc and Portier [2022])**
*Let $h > 0$ be a constant, and there exists $c > 0$, such that a gradient estimator $G_h(x)$ satisfies $\|\mathbb{E}\left[G_h(x)\right] - \nabla f(x)\| \leq ch$, for all $x \in \mathbb{R}^d$. Let $D$ be a random matrix independent of $G_h(x)$, which is equal to $e_j e_j^\top \in \mathbb{R}^d \times \mathbb{R}^d$ with probability $\lambda_j \geq 0$, $e_j \in \mathbb{R}^d$ is the $j$-th unit vector, $j \in [d]$, $\sum_{j=1}^{d} \lambda_j = 1$. For some constants $\widetilde{A}, \widetilde{C} \geq 0$, let $\mathbb{E}\left[\|G_h(x)\|^2\right] \leq 2\widetilde{A}(f(x) - f^*) + \widetilde{C}$. Let us define Composition of coordinate sampling and zeroth-order estimator $g(x) = D \cdot G_h(x)$.*

**Claim 20** *Composition of coordinate sampling and zeroth-order estimator satisfies Assumption 9 with $A = \widetilde{A} \max_j \{\lambda_j\}$, $B = 0$, $C = \widetilde{C} \max_j \{\lambda_j\}$, $b = \frac{1}{2} \min_j \{\lambda_j\}$, $c = \frac{1}{2} \max_j \{\lambda_j\} \cdot c^2 h^2$.*

In Table 8 we gather the results from the current section. In Table 9 we show whether the estimators in this section fit or not to mentioned in the present work frameworks.

We would like to note that biased gradient estimators are widely used outside classical stochastic optimization/finite-sum and distributed training settings. Works [Chen et al., 2021b,c,d] are devoted to stochastic compositional/minimax/bilevel optimization, works [Hu et al., 2020b, 2021b, 2020a] — to conditional stochastic optimization, works [Levy et al., 2020; Wang et al., 2021] — to distributionally robust optimization, work [Ji et al., 2022] — to meta-learning.

# D   Relations between assumptions 1–9

## D.1   Counterexamples to Figure 1

In Section 4 of the main part of the paper we outlined Theorem 1 in an informal way. Below we state it rigorously.

**Theorem 1** (Formal) *The following relations hold:*

   *i  There is a minimization problem for which Assumption 3 is satisfied, but Assumption 7 is not. That is, (CON) does not imply (ABS). The reverse implication also does not hold true.*

| Name of an estimator | Definition | $A$ | $B$ | $C$ | $b$ | $c$ |
|---|---|---|---|---|---|---|
| **Biased independent sampling** This paper | Def. 1 | $\frac{\max_i\{L_i\}}{\min_i p_i}$ | 0 | $2A\Delta^* + s^2$ | $\min_i\{p_i\}$ | 0 |
| **Distributed general biased rounding** This paper | Def. 2 | $A_r$ | $B_r$ | $C_r$ | $b_r$ | $c_r$ |
| **Top-$k$** [Aji and Heafield, 2017; Alistarh et al., 2018] | Def. 3 | 0 | 1 | 0 | $\frac{k}{d}$ | 0 |
| **Rand-$k$** [Stich et al., 2018] | Def. 4 | 0 | $\frac{d}{k}$ | 0 | 1 | 0 |
| **Biased Rand-$k$** [Beznosikov et al., 2020] | Def. 5 | 0 | $\frac{k}{d}$ | 0 | $\frac{k}{d}$ | 0 |
| **Adaptive random sparsification** [Beznosikov et al., 2020] | Def. 6 | 0 | 1 | 0 | $\frac{1}{d}$ | 0 |
| **General unbiased rounding** [Beznosikov et al., 2020] | Def. 7 | 0 | $\frac{Z}{4}$ | 0 | 1 | 0 |
| **General biased rounding** [Beznosikov et al., 2020] | Def. 8 | 0 | $F^2$ | 0 | $\frac{G^2}{F}$ | 0 |
| **Natural compression** [Horváth et al., 2022] | Def. 9 | 0 | $\frac{9}{8}$ | 0 | 1 | 0 |
| **General exponential dithering** [Beznosikov et al., 2020] | Def. 10 | 0 | $H_a$ | 0 | 1 | 0 |
| **Natural dithering** [Horváth et al., 2022] | Def. 11 | 0 | $H_2$ | 0 | 1 | 0 |
| **Composition of Top-$k$ and exp dithering** [Beznosikov et al., 2020] | Def. 12 | 0 | $H_a^2$ | 0 | $\frac{k}{dH_a}$ | 0 |
| **Gaussian smoothing** [Polyak, 1987] | Def. 13 | $A_{GS}$ | $B_{GS}$ | $C_{GS}$ | $b_{GS}$ | $c_{GS}$ |
| **Hard-threshold sparsifier** [Sahu et al., 2021] | Def. 14 | 0 | 1 | 0 | 1 | $w^2 d$ |
| **Scaled integer rounding** [Sapio et al., 2021] | Def. 15 | 0 | 2 | $\frac{2d}{\chi^2}$ | $\frac{1}{2}$ | $\frac{d}{2\chi^2}$ |
| **Biased dithering** [Khirirat et al., 2018a] | Def. 16 | 0 | $d$ | 0 | 1 | 0 |
| **Sign compression** [Karimireddy et al., 2019] | Def. 17 | 0 | $4 - \frac{2}{d}$ | 0 | $\frac{1}{2d}$ | 0 |

Table 8: Summary of the estimators with respective parameters $A$, $B$, $C$, $b$ and $c$, satisfying our general Biased ABC framework. Constants $L_i$ are from Assumption 13, $\Delta^*$ is defined in (26), $A_r, B_r, C_r, b_r, c_r$ are defined in (27)–(31), $Z$ is defined in (32), $F$ and $G$ are defined in (33), $H_a$ is defined in (34), $A_{GS}, B_{GS}, C_{GS}, b_{GS}, c_{GS}$ are defined in (35).

| Name of an estimator \ Assumption | A1 | A2 | A3 | A4 | A5 | A6 | A7 | A8 | A9 |
|---|---|---|---|---|---|---|---|---|---|
| **Biased independent sampling** [This paper] | ✗ | ✗ | ✗ | ✗ | ✗ | ✗ | ✗ | ✗ | ✓ |
| **Distributed general biased rounding** [This paper] | ✗ | ✗ | ✗ | ✗ | ✗ | ✗ | ✗ | ✗ | ✓ |
| **Top-$k$ sparsification** [Aji and Heafield, 2017; Alistarh et al., 2018] | ✓ | ✓ | ✓ | ✓ | ✓ | ✓ | ✗ | ✓ | ✓ |
| **Rand-$k$** [Stich et al., 2018] | ✓ | ✓ | ✗ | ✓ | ✗ | ✓ | ✗ | ✓ | ✓ |
| **Biased Random-$k$** [Beznosikov et al., 2020] | ✓ | ✓ | ✓ | ✓ | ✗ | ✓ | ✗ | ✓ | ✓ |
| **Adaptive random sparsification** [Beznosikov et al., 2020] | ✓ | ✓ | ✓ | ✓ | ✗ | ✓ | ✗ | ✓ | ✓ |
| **General unbiased rounding** [Beznosikov et al., 2020] | ✓ | ✓ | ✗ | ✓ | ✗ | ✓ | ✗ | ✓ | ✓ |
| **General biased rounding** [Beznosikov et al., 2020] | ✓ | ✓ | ✓ | ✓ | ✓ | ✓ | ✗ | ✓ | ✓ |
| **Natural compression** [Horváth et al., 2022] | ✓ | ✓ | ✓ | ✓ | ✗ | ✓ | ✗ | ✓ | ✓ |
| **General exponential dithering** [Beznosikov et al., 2020] | ✓ | ✓ | ✓ | ✓ | ✗ | ✓ | ✗ | ✓ | ✓ |
| **Natural dithering** [Horváth et al., 2022] | ✓ | ✓ | ✓ | ✓ | ✗ | ✓ | ✗ | ✓ | ✓ |
| **Composition of Top-$k$ and exp dithering** [Beznosikov et al., 2020] | ✓ | ✓ | ✓ | ✓ | ✗ | ✓ | ✗ | ✓ | ✓ |
| **Gaussian smoothing** [Polyak, 1987] | ✗ | ✗ | ✗ | ✗ | ✗ | ✓ | ✗ | ✗ | ✓ |
| **Hard-threshold sparsifier** [Sahu et al., 2021] | ✗ | ✓ | ✓ | ✓ | ✗ | ✓ | ✓ | ✓ | ✓ |
| **Scaled integer rounding** [Sapio et al., 2021] | ✓ | ✓ | ✗ | ✓ | ✓ | ✓ | ✓ | ✓ | ✓ |
| **Biased dithering** [Khirirat et al., 2018a] | ✓ | ✓ | ✗ | ✗ | ✓ | ✗ | ✗ | ✓ | ✓ |
| **Sign compression** [Karimireddy et al., 2019] | ✓ | ✓ | ✓ | ✓ | ✓ | ✓ | ✗ | ✓ | ✓ |

Table 9: Summary on an inclusion of popular estimators into every known framework.

*ii There is a minimization problem for which Assumption 3 is satisfied, but Assumption 5 is not. That is, (CON) does not imply (BREQ). The reverse implication also does not hold true.*

*iii There is a minimization problem for which Assumption 5 is satisfied, but Assumption 7 is not. That is, (BREQ) does not imply (ABS). The reverse implication also does not hold true.*

*iv There is a minimization problem for which Assumption 5 is satisfied, but Assumption 6 is not. That is, (BREQ) does not imply (BND). The reverse implication also does not hold true.*

*v There is a minimization problem for which Assumption 1 is satisfied, but Assumption 6 is not. That is, (SG1) does not imply (BND). The reverse implication also does not hold true.*

*vi There is a minimization problem for which Assumption 7 is satisfied, but Assumption 8 is not. That is, (ABS) does not imply (FSML). The reverse implication also does not hold true.*

Clearly, this theorem implies that there is a mutual abscence of implications between Assumption 7 (ABS) and Assumption 4 (BVD), Assumption 7 (ABS) and Assumption 1 (SG1), Assumption 7 (ABS) and Assumption 2 (SG2), Assumption 4 (BVD) and Assumption 5 (BREQ).

**Proof of Theorem 1** Let us prove all of the assertions stated above in Theorem 1 one by one.

i Consider $f(x) = x^2$, $g(x) = \frac{3}{2}\nabla f(x) = 3x$. We have

$$\mathbb{E}\left[\|g(x) - \nabla f(x)\|^2\right] = \left\|\frac{1}{2}\nabla f(x)\right\|^2$$
$$= x^2, \tag{36}$$

which implies due to (7) that

$$\|\mathbb{E}\left[g(x)\right] - \nabla f(x)\|^2 \le x^2, \tag{37}$$
$$\mathbb{E}\left[\|g(x) - \mathbb{E}\left[g(x)\right]\|^2\right] \le x^2. \tag{38}$$

Clearly, the estimator satisfies Assumption 3 with $\delta = \frac{4}{3}$.

Clearly, the right-hand side of (36) can not be bounded by any constant $\Delta^2$, for all $x \in \mathbb{R}$. Therefore, $g(x)$ does not satisfy Assumption 7.

Let us show that the reverse implication does not hold as well.

Let $f(x) = x^2$, $x \in \mathbb{R}$. Let $g(x) = 2x + 1$. Then $g(x)$ satisfies Assumptions 7. Indeed,

$$\mathbb{E}\left[\|g(x) - \mathbb{E}\left[g(x)\right]\|^2\right] = 0, \tag{39}$$

$$\|\mathbb{E}\left[g(x)\right] - \nabla f(x)\|^2 = 1, \tag{40}$$

which means that, due to (7), we have $\mathbb{E}\left[\|g(x) - \mathbb{E}\left[g(x)\right]\|^2\right] = 1$, and we can choose $\Delta^2 = 1$.

However, there is no $\delta > 0$, such that $\mathbb{E}\left[\|g(x) - \mathbb{E}\left[g(x)\right]\|^2\right] = 1$ can be bounded from above by $\left(1 - \frac{1}{\delta}\right)\|\nabla f(x)\|^2 = 4\left(1 - \frac{1}{\delta}\right)x^2$, for all $x \in \mathbb{R}$. Therefore, $g(x)$ does not satisfy Assumption 3.

ii The implication does not hold trivially, since Assumption 5 is formulated for deterministic estimators only.

Let us show that the reverse implication does not hold as well.

Suppose $g(x) = 3\nabla f(x)$ is a deterministic gradient estimator of $f(x)$ with $\|\nabla f(x)\|^2$ unbounded from above by a constant. Then $g(x)$ satisfies Assumption 5. Indeed, we have

$$\langle g(x), \nabla f(x)\rangle = 3\|\nabla f(x)\|^2,$$
$$\|g(x)\|^2 = 9\|\nabla f(x)\|^2.$$

It means that we can choose $\rho = 3$, $\zeta = 9$. However, since we have

$$\|\mathbb{E}\left[g(x)\right] - \nabla f(x)\|^2 = 4\|\nabla f(x)\|^2,$$

and the variance is 0 ($g(x)$ is deterministice), there is no $\delta > 0$, such that

$$\mathbb{E}\left[\|g(x) - \mathbb{E}\left[g(x)\right]\|^2\right] = 4\|\nabla f(x)\|^2$$

can be bounded from above by $\left(1 - \frac{1}{\delta}\right)\|\nabla f(x)\|^2$, for all $x \in \mathbb{R}$. Therefore, $g(x)$ does not satisfy Assumption 3.

iii Consider the example of the problem and the estimator from the proof of Theorem 1–i. Let $f(x) = x^2$, $g(x) = \frac{3}{2}\nabla f(x) = 3x$. We have

$$\langle g(x), \nabla f(x)\rangle = 6x^2, \quad \|g(x)\|^2 = 9x^2,$$

which means that this estimator satisfies Assumption 5 with $\rho = \frac{3}{2}$, $\zeta = \frac{9}{4}$.

Clearly, the right-hand side of (36) can not be bounded by any constant $\Delta^2$, for all $x \in \mathbb{R}$. Therefore, $g(x)$ does not satisfy Assumption 7.

The reverse implication does not hold trivially, since Assumption 5 is formulated for deterministic estimators only.

iv Suppose $g(x) = 3\nabla f(x)$ is a deterministic gradient estimator of $f(x)$ with $\|\nabla f(x)\|^2$ unbounded from above by a constant. In the proof of Theorem 1–ii we showed that $g(x)$ satisfies Assumption 5 with $\rho = 3$, $\zeta = 9$. However, since we have

$$\|\mathbb{E}\left[g(x)\right] - \nabla f(x)\|^2 = 4\|\nabla f(x)\|^2,$$

we are not able to find $0 \le m \le 1$ and $\varphi^2 \ge 0$, such that

$$\|\mathbb{E}\left[g(x)\right] - \nabla f(x)\|^2 \le \eta\|\nabla f(x)\|^2 + \varphi^2,$$

for all $x \in \mathbb{R}^d$. Therefore, $g(x)$ does not satisfy Assumption 4.

The reverse implication does not hold trivially, since Assumption 5 is formulated for deterministic estimators only.

v Recall the stochastic estimator from Definition 7.

Suppose $g(x)$ is a general unbiased rounding estimator multiplied by a factor of 3. Suppose that $\|\nabla f(x)\|^2$ is not bounded from above. The estimator $g(x)$ is biased:

$$\mathbb{E}\left[g(x)\right] = 3\nabla f(x).$$

Therefore,

$$\|\mathbb{E}\left[g(x)\right] - \nabla f(x)\|^2 = 4\|\nabla f(x)\|^2. \tag{41}$$

This biased estimator does not satisfy Assumption 6 since there is no $0 \le m < 1$, such that $\|\mathbb{E}\left[g(x)\right] - \nabla f(x)\|^2 \le m\|\nabla f(x)\|^2 + \varphi^2$.

Without loss of generality we assume that $x \ge 0$.

$$\begin{aligned}
\mathbb{E}\left[\|g(x) - \mathbb{E}\left[g(x)\right]\|^2\right] &= \mathbb{E}\left[\|g(x)\|^2\right] - 9\|\nabla f(x)\|^2 \\
&= \left(\frac{9}{4}\sup_{k \in \mathbb{N}}\left(\frac{a_k}{a_{k+1}} + \frac{a_{k+1}}{a_k} + 2\right) - 9\right)\|\nabla f(x)\|^2 \\
&\ge 0. \tag{42}
\end{aligned}$$

Observe that $\langle \mathbb{E}\left[g(x)\right], \nabla f(x)\rangle = 3\|\nabla f(x)\|^2$. It means that the gradient estimator satisfies Assumption 1 with $\alpha = \frac{9Z}{4}$, $\beta = \frac{3Z}{4}$, where $Z$ is defined in (32).

Let us show that the reverse implication does not hold as well.

As in the proof of Theorem 1–i, let $f(x) = x^2$, $x \in \mathbb{R}$, $g(x) = 2x + 1$. From (39) and (40), we conclude that $g(x)$ satisfies Assumptions 6 with $M = \sigma^2 = m = 0$, $\varphi^2 = 1$.

However, there is no constant $\frac{\alpha}{\beta} \ge 0$, such that a function

$$\langle \mathbb{E}\left[g(x)\right], \nabla f(x)\rangle = 2x(2x + 1)$$

can be bounded from below by

$$\frac{\alpha}{\beta}\|\nabla f(x)\|^2 = \frac{\alpha}{\beta}4x^2,$$

for all $x$. Therefore, $g(x)$ does not satisfy Assumption 1.

vi Let $f(x) = x^2$, $x \in \mathbb{R}$, $g(x) = 2x + 1$. In the proof of Theorem 1–i we showed that $g(x)$ satisfies Assumption 7. However, $g(x)$ does not satisfy Assumption 8. There is no constant $q \ge 0$, such that a function

$$\langle \mathbb{E}\left[g(x)\right], \nabla f(x)\rangle = 2x(2x + 1)$$

can be bounded from below by
$$q \left\| \nabla f(x) \right\|^2 = 4qx^2,$$
for all $x$. Therefore, $g(x)$ does not satisfy Assumption 8.

Let us show that the reverse implication does not hold as well.

Suppose $g(x)$ is a general unbiased rounding estimator (see Definition 7) multiplied by a factor of 3. Suppose that $\left\| \nabla f(x) \right\|^2$ is not bounded from above. This estimator satisfies Assumption 8. Indeed, observe that $\langle \mathbb{E}\left[ g(x) \right], \nabla f(x) \rangle = 3 \left\| \nabla f(x) \right\|^2$. Also, $\left\| \mathbb{E}\left[ g(x) \right] \right\|^2 = 9 \left\| \nabla f(x) \right\|^2$. Therefore, we can choose $q = u = 3, U = Z - 9, Q = 0$.

Due to (7), (41) and (42), we have

$$\mathbb{E}\left[ \left\| g(x) - \nabla f(x) \right\|^2 \right] = 4 \left\| \nabla f(x) \right\|^2 + \left( \frac{9}{4} \sup_{k \in \mathbb{N}} \left( \frac{a_k}{a_{k+1}} + \frac{a_{k+1}}{a_k} + 2 \right) - 9 \right) \left\| \nabla f(x) \right\|^2$$
$$\geq 4 \left\| \nabla f(x) \right\|^2.$$

Then $g(x)$ does not satisfy Assumption 7 since there is no $\Delta \geq 0$, such that $4 \left\| \nabla f(x) \right\|^2 \leq \Delta^2$ holds, for all $x \in \mathbb{R}^d$.

$\blacksquare$

## D.2 Implications in Figure 1

In Section 5.2 of the main part of the paper we outlined Theorem 2 in an informal way. Below we state it rigorously.

**Theorem 2** (Formal) *Let Assumption 0 hold for the function $f$. Then the following relations hold:*

*i Suppose a gradient estimator $g(x)$ satisfies Assumption 3. Then $g(x)$ satisfies Assumption 4 with $\eta = 1 - \frac{1}{\delta}, \xi = 1 - \frac{1}{\delta}$. That is, $(CON)$ implies $(BVD)$. The reverse implication does not hold.*

*ii Suppose a gradient estimator $g(x)$ satisfies Assumption 4. Then $g(x)$ satisfies Assumption 6 with $m = \eta, \varphi^2 = 0, M = \frac{2\xi(1+\eta)}{(1-\eta)^2}, \sigma^2 = 0$. That is, $(BVD)$ implies $(BND)$. The reverse implication does not hold.*

*iii Suppose a gradient estimator $g(x)$ satisfies Assumption 7. Then $g(x)$ satisfies Assumption 6 with $M = m = 0, \sigma^2 = \varphi^2 = \Delta^2$. That is, $(ABS)$ implies $(BND)$. The reverse implication does not hold.*

*iv Suppose a gradient estimator $g(x)$ satisfies Assumption 4. Then $g(x)$ satisfies Assumption 1 with $\alpha = \frac{(1-\eta)^2}{2(1+\eta)}, \beta = \frac{2}{1-\eta} \max\{\xi, 2\xi + \eta - 1\}$. That is, $(BVD)$ implies $(SG1)$. The reverse implication does not hold.*

*v Suppose a gradient estimator $g(x)$ satisfies Assumption 5. Then $g(x)$ satisfies Assumption 1. That is, $(BREQ)$ implies $(SG1)$. The reverse implication does not hold.*

*vi Assumption 1 $(SG1)$ is equivalent to Assumption 2 $(SG2)$.*

*vii Suppose a gradient estimator $g(x)$ satisfies Assumption 1. Then $g(x)$ satisfies Assumption 8 with $u = U = \beta^2, Q = 0, q = \frac{\alpha}{\beta}$. That is, (SG1) implies (FSML). The reverse implication does not hold.*

*viii Suppose a gradient estimator $g(x)$ satisfies Assumption 8. Then $g(x)$ satisfies Assumption 9 with $A = 0, B = U + u^2, C = Q, b = q, c = 0$. That is, (FSML) implies $(Biased\ ABC)$. The reverse implication does not hold.*

*ix Suppose a gradient estimator $g(x)$ satisfies Assumption 6. Then $g(x)$ satisfies Assumption 9 with $A = 0, B = 2(M+1)(m+1), C = 2(M+1)\varphi^2 + \sigma^2, b = \frac{1-m}{2}, c = \frac{\varphi^2}{2}$. That is, $(BND)$ implies $(Biased\ ABC)$. The reverse implication does not hold.*

**Proof of Theorem 2** Let us prove all of the assertions stated above in Theorem 2 one by one.
i. From (6) and from (7), we easily derive the following inequalities:

$$\left\| \mathbb{E}\left[ g(x) \right] - \nabla f(x) \right\|^2 \leq \left( 1 - \frac{1}{\delta} \right) \left\| \nabla f(x) \right\|^2,$$

and

$$\mathbb{E}\left[\|g(x) - \mathbb{E}\left[g(x)\right]\|^2\right] \le \left(1 - \frac{1}{\delta}\right) \|\nabla f(x)\|^2.$$

Therefore, we can choose $\eta = 1 - \frac{1}{\delta}$, $\xi = 1 - \frac{1}{\delta}$.

Next, let us show that the reverse implication does not hold. Suppose $g(x)$ is a gradient estimator of the following form:

$$g(x) = \nabla f(x) + X, \text{ where } X = \begin{cases} 4 \nabla f(x), & \text{with probability } \frac{1}{4} \\ 0, & \text{with probability } \frac{3}{4}. \end{cases}$$

For the estimator $g(x)$ we have

$$\|\mathbb{E}\left[g(x)\right] - \nabla f(x)\|^2 = \|\nabla f(x)\|^2,$$

and

$$\mathbb{E}\left[\|g(x) - \mathbb{E}\left[g(x)\right]\|^2\right] = \mathbb{E}\left[\|X\|^2\right] - \|\mathbb{E}\left[X\right]\|^2 = 3 \|\nabla f(x)\|^2.$$

We can choose $\eta = 1$, $\xi = 3$, so $g(x)$ satisfies Assumption 4. But there is no $\delta \ge 1$, such that, for all $x \in \mathbb{R}^d$,

$$\mathbb{E}\left[\|g(x) - \nabla f(x)\|^2\right] \overset{(7)}{=} \mathbb{E}\left[\|g(x) - \mathbb{E}\left[g(x)\right]\|^2\right] + \|\mathbb{E}\left[g(x)\right] - \nabla f(x)\|^2 = 4 \|\nabla f(x)\|^2$$

does not exceed $\left(1 - \frac{1}{\delta}\right) \|\nabla f(x)\|^2$. Then $g(x)$ does not satisfy Assumption 3.

ii. Since we know that
$$\|\mathbb{E}\left[g(x)\right] - \nabla f(x)\|^2 \le \eta \|\nabla f(x)\|^2, \tag{43}$$
we can choose $m = \eta$ and $\varphi^2 = 0$. By Young's Inequality (Lemma 3, (68)), from (43) we derive that

$$(1 - \eta) \|\nabla f(x)\|^2 \le 2 \langle \mathbb{E}[g(x)], \nabla f(x) \rangle - \|\mathbb{E}[g(x)]\|^2$$
$$\le \frac{(1 - \eta) \|\nabla f(x)\|^2}{2} + \frac{2 \|\mathbb{E}[g(x)]\|^2}{(1 - \eta)} - \|\mathbb{E}[g(x)]\|^2.$$

Hence,

$$\|\nabla f(x)\|^2 \le \frac{2(1 + \eta)}{(1 - \eta)^2} \|\mathbb{E}\left[g(x)\right]\|^2.$$

Also, we know that

$$\mathbb{E}\left[\|g(x) - \mathbb{E}\left[g(x)\right]\|^2\right] \le \xi \|\nabla f(x)\|^2.$$

Therefore, we arrive at

$$\mathbb{E}\left[\|g(x) - \mathbb{E}\left[g(x)\right]\|^2\right] \le \frac{2\xi(1 + \eta)}{(1 - \eta)^2} \|\mathbb{E}\left[g(x)\right]\|^2.$$

We can choose $M = \frac{2\xi(1+\eta)}{(1-\eta)^2}$, $\sigma^2 = 0$.

Next, let us show that the reverse implication does not hold. As in the proof of Theorem 1–i, let $f(x) = x^2$, $x \in \mathbb{R}$. Let $g(x) = 2x + 1$. From (39) and (40), we conclude that $g(x)$ satisfies Assumption 6 with $M = \sigma^2 = m = 0$, $\varphi^2 = 1$.

However, there is no $0 \le \eta \le 1$, such that $\|\mathbb{E}\left[g(x)\right] - \nabla f(x)\|^2 = 1$ is bounded from above by $\xi \|\nabla f(x)\|^2 = 4\eta x^2$, for all $x \in \mathbb{R}$. It means that Assumption 4 does not hold.

iii Indeed, (7) and (14) imply $\mathbb{E}\left[\|g(x) - \mathbb{E}\left[g(x)\right]\|^2\right] \le \Delta^2$ and $\|\mathbb{E}\left[g(x)\right] - \nabla f(x)\|^2 \le \Delta^2$. Therefore, Assumption 6 is satisfied with $M = m = 0$, $\sigma^2 = \varphi^2 = \Delta^2$.

Next, let us prove that the reverse implication does not hold. Consider the example of the problem and the estimator from the proof of Theorem 1–iii. From (37) and (38) we conclude that the estimator satisfies Assumption 6 with $M = \frac{1}{9}$, $m = \frac{1}{4}$, but Assumption 7 is not satisfied.

iv. Since we know that

$$\|\mathbb{E}[g(x)] - \nabla f(x)\|^2 \le \eta \|\nabla f(x)\|^2, \tag{44}$$

we obtain

$$(1 - \eta)\|\nabla f(x)\|^2 \le 2\langle \mathbb{E}[g(x)], \nabla f(x)\rangle - \|\mathbb{E}[g(x)]\|^2. \tag{45}$$

Then

$$\mathbb{E}\left[\|g(x) - \mathbb{E}[g(x)]\|^2\right] \le \xi \|\nabla f(x)\|^2$$
$$\le \frac{2\xi}{1-\eta}\langle \mathbb{E}[g(x)], \nabla f(x)\rangle - \frac{\xi}{1-\eta}\|\mathbb{E}[g(x)]\|^2.$$

If $\xi + \eta \le 1$, we obtain that

$$\mathbb{E}\left[\|g(x)\|^2\right] \le \frac{2\xi}{1-\eta}\langle \mathbb{E}[g(x)], \nabla f(x)\rangle.$$

Otherwise,

$$\mathbb{E}\left[\|g(x)\|^2\right] \le \frac{2\xi}{1-\eta}\langle \mathbb{E}[g(x)], \nabla f(x)\rangle + \left(\frac{\xi}{1-\eta} - 1\right)\|\mathbb{E}[g(x)]\|^2$$
$$\le \frac{2(2\xi + \eta - 1)}{1-\eta}\langle \mathbb{E}[g(x)], \nabla f(x)\rangle.$$

Hence, we can choose $\beta = \frac{2}{1-\eta}\max\{\xi, 2\xi + \eta - 1\}$. Further, by Young's Inequality (Lemma 3, (68)), from (43) we derive that

$$(1-\eta)\|\nabla f(x)\|^2 \le 2\langle \mathbb{E}[g(x)], \nabla f(x)\rangle - \|\mathbb{E}[g(x)]\|^2$$
$$\le \frac{(1-\eta)\|\nabla f(x)\|^2}{2} + \frac{2\|\mathbb{E}[g(x)]\|^2}{(1-\eta)} - \|\mathbb{E}[g(x)]\|^2.$$

Then we have

$$\|\mathbb{E}[g(x)]\|^2 \ge \frac{(1-\eta)^2}{2(1+\eta)}\|\nabla f(x)\|^2.$$

Therefore, we can choose $\alpha = \frac{(1-\eta)^2}{2(1+\eta)}$.

Let us show that the inverse implication does not hold.

Consider the problem and the estimator from the proof of Theorem 1–v. Since $\|\nabla f(x)\|^2$ is not bounded from above, this estimator does not satisfy Assumption 4: there is no $0 \le \eta \le 1$ such that

$$\|\mathbb{E}[g(x)] - \nabla f(x)\|^2 \le \eta \|\nabla f(x)\|^2.$$

However, recall that Assumption 1 is satisfied with $\alpha = Z$, $\beta = \frac{Z}{3}$, where $Z$ is defined in (32).

v. Observe that

$$\|\nabla f(x)\|^2 \le \frac{1}{\rho}\langle g(x), \nabla f(x)\rangle.$$

Therefore,

$$\|g(x)\|^2 \le \zeta \|\nabla f(x)\|^2 \le \frac{\zeta}{\rho}\langle g(x), \nabla f(x)\rangle,$$

and we can choose $\beta = \frac{\zeta}{\rho}$ in Assumption 1. By Young's Inequality (Lemma 3, (68)), we have

$$\rho \|\nabla f(x)\|^2 \le \langle g(x), \nabla f(x)\rangle$$
$$\le \frac{\|g(x)\|^2}{2\rho} + \frac{\rho \|\nabla f(x)\|^2}{2}.$$

This implies that $\|g(x)\|^2 \geq \rho^2 \|\nabla f(x)\|^2$, and we can choose $\alpha = \rho^2$ in Assumption 1.

The reverse implication does not hold. Since Assumption 5 is formulated for deterministic estimators only, any stochastic estimator that satisfies Assumption 1 does not satisfy Assumption 5.

vi. It follows from assertions 1 and 2 of Theorem 14.

vii Recall that Assumption 1 implies (5). Since $\|\mathbb{E}[g(x)]\|^2 \leq \mathbb{E}\left[\|g(x)\|^2\right]$, we can choose $u = \beta$. From $\langle \mathbb{E}[g(x)], \nabla f(x) \rangle \geq \alpha \|\nabla f(x)\|$, we conclude that $q$ can be set to $\frac{\alpha}{\beta}$. Furthermore, $\mathbb{E}\left[\|g(x) - \mathbb{E}[g(x)]\|^2\right] \leq \mathbb{E}\left[\|g(x)\|^2\right]$ and (5) imply that we can put $U$ equal to $\beta^2$, $Q = 0$. Note, that Theorem 14 states that $\beta^2 \geq \alpha$. Therefore, the requirement $q \leq u$ from Assumption 8 is also satisfied.

Let us prove that the reverse implication does not hold. For every $x \in \mathbb{R}$, consider $f(x) = x^3$, $g(x) = Y\nabla f(x) + Z$, where $Y$ is a random variable with $\text{Bern}\left(\frac{1}{2}\right)$ distribution, independent of a random variable $Z$ that attains values $\pm 1$ with equal probability. First, we establish relations (15), (16) and (17) in this setting:

$$\langle \nabla f(x), \mathbb{E}[g(x)] \rangle = \frac{1}{2}\|\nabla f(x)\|^2 = \frac{9}{2}x^4,$$

$$\|\mathbb{E}[g(x)]\|^2 = \frac{1}{4}\|\nabla f(x)\|^2 = \frac{9}{4}x^4,$$

$$\mathbb{E}\left[\|g(x)\|^2\right] - \|\mathbb{E}[g(x)]\|^2 = \mathbb{E}\left[Y^2\|\nabla f(x)\|^2 + 2YZ\nabla f(x) + Z^2\right] - \frac{1}{4}\|\nabla f(x)\|^2$$

$$= \frac{1}{2}\|\nabla f(x)\|^2 + 1 - \frac{1}{4}\|\nabla f(x)\|^2$$

$$= \frac{1}{4}\|\nabla f(x)\|^2 + 1$$

$$= \frac{9}{4}x^4 + 1.$$

This implies, that $g(x)$ satisfies Assumption 8 with $q = u = \frac{1}{2}$, $U = \frac{1}{4}$ and $Q = 1$.

Consider the implication (5) from Assumption 1. Notice, that

$$\mathbb{E}\left[\|g(x)\|^2\right] = \frac{1}{2}\|\nabla f(x)\|^2 + 1 = \frac{9}{2}x^4 + 1,$$

and it can not be bounded from above by $\beta^2\|\nabla f(x)\|^2 = 9\beta^2 x^4$, for all $x \in \mathbb{R}$. Therefore, (5) does not hold, which means that Assumption 1 also does not hold.

viii. Suppose $g(x)$ satisfies Assumption 8.

From (15), we conclude that $b$ can be chosen as $q$, $c$ can be chosen as $0$. Further, (17) implies that

$$\mathbb{E}\left[\|g(x)\|^2\right] \leq U\|\nabla f(x)\|^2 + \|\mathbb{E}[g(x)]\|^2 + Q.$$

From (16), we obtain that

$$\mathbb{E}\left[\|g(x)\|^2\right] \leq \left(U + u^2\right)\|\nabla f(x)\|^2 + Q.$$

Therefore, we can choose $A = 0$, $B = U + u^2$, $C = Q$.

Next, let us prove that the reverse implication does not hold. Consider function $f$ which is 1-smooth and lower bounded by $0$ :

$$f(x) = \begin{cases} \frac{x^2}{2}, & \text{if } |x| < 1, \\ |x| - \frac{1}{2}, & \text{otherwise.} \end{cases}$$

(Huber Loss). Consider a biased estimator

$$g(x) = \begin{cases} \nabla f(x) + \sqrt{|x|} + 1 & \text{with probability } 1/2, \\ \nabla f(x) - \sqrt{|x|} + 1 & \text{with probability } 1/2. \end{cases}$$

Observe that $\mathbb{E}g(x) = \nabla f(x) + 1$. Suppose condition (17) of Assumption 8 holds. Then there exist constants $U, Q \geq 0$ such that,

$$\mathbb{E}\left[\|g(x)\|^2\right] - \|\mathbb{E}\left[g(x)\right]\|^2 \leq U\|\nabla f(x)\|^2 + Q.$$

Consider the point $x = U + Q + 4$, then $|x| > 1$ and hence $\nabla f(x) = 1$ by the definition of $f$. Then we obtain

$$\mathbb{E}\left[\|g(x)\|^2\right] \leq U + Q + 4$$

On the other hand, we fall into contradiction since

$$\mathbb{E}\left[\|g(x)\|^2\right] = \frac{1}{2}\left((2 + \sqrt{x})^2 + (2 - \sqrt{x})^2\right) = x + 4 = U + Q + 8.$$

It follows that condition (17) of Assumption 8 does not hold. We now show that Assumption 9 holds: first, suppose that $x \geq 1$, then

$$\begin{aligned} \mathbb{E}\left[\|g(x)\|^2\right] &= \frac{1}{2}\left((2 + \sqrt{|x|})^2 + (2 - \sqrt{|x|})^2\right) \\ &= \frac{1}{2}(8 + 2|x|) = 4 + |x| \\ &= \frac{9}{2} + \left(f(x) - f^{\text{inf}}\right), \end{aligned}$$

since for $|x| \geq 1$ we have $f(x) - f^{\text{inf}} = |x| - 1/2$. In turn,

$$\langle \nabla f(x), \mathbb{E}\left[g(x)\right]\rangle = \langle \nabla f(x), \nabla f(x) + 1\rangle \geq \|\nabla f(x)\|^2.$$

Suppose that $x \leq -1$, then $\nabla f(x) = -1$, and

$$\begin{aligned} \mathbb{E}\left[\|g(x)\|^2\right] &= \frac{1}{2}\left((\sqrt{|x|})^2 + (-\sqrt{|x|})^2\right) \\ &= |x| \\ &= \frac{1}{2} + \left(f(x) - f^{\text{inf}}\right). \end{aligned}$$

In turn,

$$\langle \nabla f(x), \mathbb{E}\left[g(x)\right]\rangle = \langle \nabla f(x), \nabla f(x) + 1\rangle = \|\nabla f(x)\|^2 - 1.$$

Now suppose that $|x| \leq 1$, then

$$\begin{aligned} \mathbb{E}\left[\|g(x)\|^2\right] &= \frac{1}{2}\left(\left(x + \sqrt{|x|}\right)^2 + \left(x - \sqrt{|x|}\right)^2\right) \\ &= x^2 + |x| \\ &\leq 1 + 1 \\ &= 2. \end{aligned}$$

In turn,

$$\langle \nabla f(x), \mathbb{E}\left[g(x)\right]\rangle = \langle \nabla f(x), \nabla f(x) + 1\rangle = \|\nabla f(x)\|^2 + x \geq \|\nabla f(x)\|^2 - 1.$$

It means that, for all $x \in \mathbb{R}$,

$$\mathbb{E}\left[\|g(x)\|^2\right] \leq f(x) - f^{inf} + \frac{9}{2}$$

and

$$\langle \nabla f(x), \mathbb{E}\left[g(x)\right]\rangle \geq \|\nabla f(x)\|^2 - 1.$$

It follows that Assumption 9 is satisfied with $A = \frac{1}{2}, B = 0, C = \frac{9}{2}, b = c = 1$.

ix. First, we bound the second moment of $g(x)$ :

$$\mathbb{E}\left[\|g(x)\|^2\right] = \mathbb{E}\left[\|g(x) - \mathbb{E}\left[g(x)\right]\|^2\right] + \|\mathbb{E}\left[g(x)\right]\|^2$$

$$= \mathbb{E}\|\mathcal{N}(x,Y)\|^2 + \|\nabla f(x) + b(x)\|^2$$

$$\leq (M+1)\|\nabla f(x) + b(x)\|^2 + \sigma^2$$

$$\leq 2(M+1)\|\nabla f(x)\|^2 + 2(M+1)\|b(x)\|^2 + \sigma^2$$

$$\leq 2(M+1)(m+1)\|\nabla f(x)\|^2 + 2(M+1)\varphi^2 + \sigma^2.$$

We can choose $A = 0$, $B = 2(M+1)(m+1)$, $C = 2(M+1)\varphi^2 + \sigma^2$ in Assumption 9. Further, note that (13) can be rewritten in an equivalent way in terms of the lower bound on the scalar product:

$$\langle \nabla f(x), \mathbb{E}\left[g(x)\right]\rangle = \frac{\|\nabla f(x)\|^2}{2} + \frac{\|\mathbb{E}\left[g(x)\right]\|^2}{2} - \frac{\|\mathbb{E}\left[g(x)\right] - \nabla f(x)\|^2}{2}$$

$$\geq \frac{1-m}{2}\|\nabla f(x)\|^2 + \frac{\|\mathbb{E}\left[g(x)\right]\|^2}{2} - \frac{\varphi^2}{2}. \tag{46}$$

Therefore,

$$\langle \nabla f(x), \mathbb{E}\left[g(x)\right]\rangle \geq \frac{1-m}{2}\|\nabla f(x)\|^2 - \frac{\varphi^2}{2}. \tag{47}$$

Observe that in (47) we used only a trivial lower bound of 0 on $\mathbb{E}\left[g(x)\right]$, which signifies that our assumption on scalar product (18) is less restrictive than the Assumption 13 on the bias term.

Let us prove that the reverse implication does not hold. Consider the problem and the estimator from the proof of Theorem 1–viii. Suppose that condition (12) of Assumption 6 holds. Then there exist $M, \sigma^2 \geq 0$ such that

$$\mathbb{E}\left[\|g(x)\|^2\right] - \|\mathbb{E}\left[g(x)\right]\|^2 \leq M\|\mathbb{E}\left[g(x)\right]\|^2 + \sigma^2.$$

Consider the point $x = 4(M+1) + \sigma^2$, then $|x| > 1$ and hence $\nabla f(x) = 1$ by the definition of $f$. Then we obtain that

$$\mathbb{E}\left[\|g(x)\|^2\right] \leq 4(M+1) + \sigma^2.$$

On the other hand, we fall into contradiction since

$$\mathbb{E}\left[\|g(x)\|^2\right] = \frac{1}{2}\left(\left(2 + \sqrt{|x|}\right)^2 + \left(2 - \sqrt{|x|}\right)^2\right) = x + 4 = 4(M+1) + \sigma^2 + 4.$$

It is shown in the proof of Theorem 1–viii that Assumption 9 is satisfied with $A = \frac{1}{2}$, $B = 0$, $C = \frac{9}{2}$, $b = c = 1$.

$\blacksquare$

### D.2.1 Proof of Claim 1

Let $p_1 = p_2 = \frac{1}{3}$ be probabilities. For every $i \in \{1,2\}$, define a random set as follows:

$$S_i = \begin{cases} \{i\} & \text{with probability } p_i, \\ \varnothing & \text{with probability } 1 - p_i. \end{cases}$$

Define a random subset $S \subseteq \{1,2\}$ by taking the union of these random sets:

$$S \stackrel{\text{def}}{=} S_1 \cup S_2.$$

For every $i \in \{1,2\}$, define $v_i = \frac{\mathbb{1}_{i \in S}}{p_i^2}$. Let

$$g(x) = \frac{1}{2}\sum_{i=1}^{n} v_i \nabla f_i(x).$$

Consider $f(x) = \frac{1}{2}\left(f_1(x) + f_2(x)\right)$, where $f_1(x) = x_1^2$, $f_2(x) = x_2^2$. For the introduced stochastic gradient, we have

$$\langle \mathbb{E}\left[g(x)\right], \nabla f(x) \rangle = 3\left(x_1^2 + x_2^2\right). \tag{48}$$

Therefore, $g(x)$ satisfies (18) of Assumption 9 with $b = 3$, $c = 0$. Observe that

$$\mathbb{E}\left[\|g(x)\|^2\right] = 27\left(x_1^2 + x_2^2\right). \tag{49}$$

Therefore, $g(x)$ also satisfies (19) with $A = 0, B = 27, C = 0$.

Recall that inequality (13) of Assumption 6 is equivalent to (46).

Since $\|\mathbb{E}\left[g(x)\right]\|^2 = 9\left(x_1^2 + x_2^2\right)$, the right-hand side of (46) is equal to

$$\frac{10 - m}{2}\left(x_1^2 + x_2^2\right) - \frac{\varphi^2}{2},$$

$0 \le m < 1$, $\varphi^2 \ge 0$. This expression can not bound (48) from below, for all $x = (x_1, x_2) \in \mathbb{R}^2$. Hence, this gradient estimator does not satisfy (13) of Assumption 6.

■

# E General nonconvex case: history and corollaries from Theorem 3

In Section 6.1 we have formulated Theorem 3 on convergence of BiasedSGD under Biased ABC assumption and compared the rate obtained to the known convergence results in nonconvex case. Below we present recent results, derive several corollaries from Theorem 3 and make a formal comparison of our results to the known results.

## E.1 Known results

Convergence of BiasedSGD in general smooth case has been studied in several papers. The next two results are Lemma 3 and Theorem 4 from [Ajalloeian and Stich, 2020]. We formulate them as a theorem and its corollary respectively.

**Theorem 6** *Under Assumptions 0 and 6, and for any stepsize $\gamma \le \frac{1}{(M+1)L}$, it holds after $T$ steps of* BiasedSGD *that*

$$\frac{1}{T}\sum_{t=0}^{T-1}\mathbb{E}\left[\|\nabla f(x^t)\|^2\right] \le \frac{2\delta^0}{T\gamma(1-m)} + \frac{\gamma L \sigma^2}{1-m} + \frac{\varphi^2}{1-m}.$$

**Corollary 1** *Under Assumptions 0 and 6, and by choosing the stepsize $\gamma = \min\left\{\frac{1}{(M+1)L}, \frac{\varepsilon(1-m)}{2L\sigma^2}\right\}$, for $\varepsilon > 0$, we have that*

$$T = \mathcal{O}\left(\max\left\{\frac{4(M+1)}{\varepsilon(1-m)}, \frac{8\sigma^2}{\varepsilon^2(1-m)^2}\right\}L\delta^0\right)$$

*iterations suffice to obtain*

$$\frac{1}{T}\sum_{t=0}^{T-1}\mathbb{E}\left[\|\nabla f(x^t)\|^2\right] = \mathcal{O}\left(\varepsilon + \frac{\varphi^2}{1-m}\right).$$

The convergence result that we get in Theorem 3 is formulated in terms of minimum of expected squared gradient norms. However, in Corollary 1 the convergence established not for the minimum, but for the mean of expected squared gradient norms. Since the minimum is not greater than the mean, we can immediately restate Corollary 1 in a slightly weaker form:

**Corollary 2** *Under Assumptions 0 and 6, and by choosing the stepsize* $\gamma = \min\left\{\frac{1}{(M+1)L}, \frac{\varepsilon(1-m)}{2L\sigma^2}\right\}$, *for* $\varepsilon > 0$, *we have that*

$$T = \mathcal{O}\left(\max\left\{\frac{4(M+1)}{\varepsilon(1-m)}, \frac{8\sigma^2}{\varepsilon^2(1-m)^2}\right\}L\delta^0\right)$$

*iterations suffice to obtain*

$$\min_{0 \leq t \leq T-1}\mathbb{E}\left[\left\|\nabla f(x^t)\right\|^2\right] = \mathcal{O}\left(\varepsilon + \frac{\varphi^2}{1-m}\right).$$

The result below is Theorem 4.8 from [Bottou et al., 2018].

**Theorem 7** *Under Assumptions 0 and 8, and for any stepsize* $0 < \gamma \leq \frac{q}{L(U+u^2)}$, *for all* $T \in \mathbb{N}$, *the following inequality holds:*

$$\frac{1}{T}\sum_{t=0}^{T-1}\mathbb{E}\left[\left\|\nabla f(x^t)\right\|^2\right] \leq \frac{\gamma LQ}{q} + \frac{2\delta^0}{Tq\gamma}.$$

To be able to make a further comparison of convergence rates, we need to establish the rate the above theorem yields. Once again, the convergence result that we get in Theorem 3 is formulated in terms of minimum of expected squared gradient norms. However, in Corollary 7 the convergence established not for the minimum, but for the mean of expected squared gradient norms. Since minimum is smaller than the mean, we can immediately write the corollary in a slightly weaker form:

**Corollary 3** *For* $\varepsilon > 0$, *choose stepsize* $\gamma > 0$ *as* $\gamma = \min\left\{\frac{\varepsilon q}{2LQ}, \frac{q}{L(U+u^2)}\right\}$. *Then, if*

$$T \geq \max\left\{\frac{8Q}{\varepsilon^2 q^2}, \frac{4(U+u^2)}{\varepsilon q^2}\right\}L\delta^0,$$

*we have that*

$$\min_{0 \leq t \leq T-1}\mathbb{E}\left[\left\|\nabla f(x^t)\right\|^2\right] \leq \varepsilon.$$

### E.2 Corollaries from Theorem 3

In general, Theorem 3 guarantees the convergence towards some neghborhood of the $\varepsilon$-stationary point, that can not be made less than $\frac{c}{b}$. Therefore, we have the following corollary.

**Corollary 4** *Choose the stepsize* $\gamma > 0$ *as* $\gamma = \min\left\{\frac{1}{\sqrt{LAT}}, \frac{b}{LB}, \frac{c}{LC}\right\}$. *Then if*

$$T \geq \frac{6\delta^0 L}{c}\max\left\{\frac{B}{b}, \frac{6\delta^0 A}{c}, \frac{C}{c}\right\},$$

*we have*

$$\min_{0 \leq t \leq T-1}\mathbb{E}\left[\left\|\nabla f(x^t)\right\|^2\right] \leq \frac{3c}{b}.$$

Next two corollaries are Theorem 2 and Corollary 1 from [Khaled and Richtárik, 2023]. However, in that work the authors obtain these results in the unbiased case, i.e. when $\mathbb{E}[g(x)] = \nabla f(x)$ holds, for all $x \in \mathbb{R}^d$. In our case we only require $\langle \mathbb{E}[g(x)], \nabla f(x)\rangle \geq \|\nabla f(x)\|^2$ to hold, for all $x \in \mathbb{R}^d$.

**Corollary 5** *Suppose* $c = 0$, $b = 1$. *Choose the stepsize such that* $0 < \gamma \leq \frac{1}{LB}$. *Then the iterates* $\{x^t\}_{t \geq 0}$ *of* BiasedSGD *(Algorithm* (1)*) satisfy*

$$\min_{0 \leq t \leq T-1}\mathbb{E}\left[\left\|\nabla f(x^t)\right\|^2\right] \leq \frac{2\left(1 + LA\gamma^2\right)^T}{\gamma T}\delta^0 + LC\gamma. \tag{50}$$

**Corollary 6** *Suppose $c = 0$ and $b = 1$. Fix $\varepsilon > 0$. Choose the stepsize $\gamma > 0$ as $\gamma = \min\left\{\frac{1}{\sqrt{LAT}}, \frac{1}{LB}, \frac{\varepsilon}{2LC}\right\}$. Then, if*

$$T \geq \frac{12\delta^0 L}{\varepsilon^2} \max\left\{B, \frac{12\delta^0 A}{\varepsilon^2}, \frac{2C}{\varepsilon^2}\right\},$$

*we have*

$$\min_{0 \leq t \leq T-1} \mathbb{E}\left[\|\nabla f(x^t)\|\right] \leq \varepsilon.$$

The next corollary contains the result similar to the one obtained in Theorem 4 from [Ajalloeian and Stich, 2020]. However, we impose weaker assumptions (compare Biased ABC and BND in Figure 1; see also Claim 1).

**Corollary 7** *Suppose $A = 0$, $b \leq 1$. Choose stepsize $\gamma > 0$ as $\gamma = \min\left\{\frac{b}{LB}, \frac{\varepsilon b}{2LC}\right\}$. Then, for $\varepsilon > 0$, we have that*

$$\mathcal{T} = \mathcal{O}\left(\max\left\{\frac{8C}{b^2\varepsilon^2}, \frac{4B}{b^2\varepsilon}\right\}L\delta^0\right)$$

*iterations suffice for*

$$\min_{0 \leq t \leq T-1} \mathbb{E}\left[\|\nabla f(x^t)\|^2\right] = \mathcal{O}\left(\varepsilon + \frac{c}{b}\right).$$

If we substitute $B$ for $2(M + 1)(m + 1)$, $C$ for $2(M + 1)\varphi^2 + \sigma^2$, $b$ for $\frac{1-m}{2}$, $c$ for $\frac{\varphi^2}{2}$ in accordance with Theorem 13 (see also Table 1), Corollary 7 yields the rate of $\mathcal{O}\left(\max\left\{\frac{8(M+1)(m+1)}{(1-m)^2\varepsilon}, \frac{16(M+1)\varphi^2+2\sigma^2}{(1-m)^2\varepsilon^2}\right\}L\delta^0\right)$ while Corollary 2 (see Theorem 4 from [Ajalloeian and Stich, 2020]) grants the rate of $\mathcal{T} = \mathcal{O}\left(\max\left\{\frac{2\sigma^2}{(1-m)^2\varepsilon^2}, \frac{M+1}{(1-m)\varepsilon}\right\}L\delta^0\right)$. Our result is worse by a factor of $\frac{1}{1-m}$ and by an additive term of $\mathcal{O}\left(\frac{(M+1)\varphi^2}{(1-m)^2\varepsilon^2}L\delta^0\right)$.

**Corollary 8** *Suppose $A = c = 0$. For $\varepsilon > 0$, choose stepsize $\gamma = \min\left\{\frac{b}{LB}, \frac{b\varepsilon}{LC}\right\}$. Then, if*

$$T \geq \max\left\{\frac{8C}{\varepsilon^2 b^2}, \frac{4B}{\varepsilon b^2}\right\}L\delta^0,$$

*we have that*

$$\min_{0 \leq t \leq T-1} \mathbb{E}\left[\|\nabla f(x^t)\|^2\right] \leq \varepsilon.$$

To recover the result from Corollary 3, one needs to substitute $B$ for $U + u^2$, $C$ for $Q$, $b$ for $q$ in accordance with the representation of Assumption 8 in Biased ABC framework (see Theorem 13 and Table 1).

### E.3 Proof of Corollary 3

If $\gamma = \frac{\varepsilon q}{2LQ}$, and $T \geq \frac{8LQ\delta^0}{\varepsilon^2 q^2}$, then we have that

$$\frac{\gamma LQ}{q} \leq \frac{\varepsilon}{2}, \qquad \frac{2\delta^0}{Tq\gamma} = \frac{4LQ\delta^0}{T\varepsilon q^2} \leq \frac{\varepsilon}{2}.$$

If $\gamma = \frac{q}{L(U+u^2)}$ and $T \geq \frac{4L(U+u^2)\delta^0}{\varepsilon q^2}$, then we obtain that

$$\frac{\gamma LQ}{q} \leq \frac{\varepsilon q}{2LQ} \cdot \frac{LQ}{q} \leq \frac{\varepsilon}{2}, \qquad \frac{2\delta^0}{Tq\gamma} = \frac{2\delta^0 L(U+u^2)}{Tq^2} \leq \frac{\varepsilon}{2}.$$

Therefore, we get that

$$\min_{0 \leq t \leq T-1} \mathbb{E}\left[\|\nabla f(x^t)\|^2\right] \leq \varepsilon.$$

■

## E.4 Key lemma

Our main convergence result in the nonconvex scenario relies on the following key lemma.

**Lemma 2** *Let Assumptions 0 and 9 hold. Choose stepsize $\gamma$ satisfying*

$$0 < \gamma \leq \frac{b}{LB}. \tag{51}$$

*Then, for any $T \geq 1$, the iterates $\{x^t\}$ of Algorithm 1 satisfy*

$$\frac{b}{2} \sum_{t=0}^{T-1} w_t r^t \leq \frac{w_{-1}}{\gamma} \delta^0 - \frac{w_{T-1}}{\gamma} \delta^T + \frac{LC\gamma + c}{2} \sum_{t=0}^{T-1} w_t.$$

**Proof of Lemma 2** From Assumption 0 we have

$$f(x^{t+1}) \leq f(x^t) + \langle \nabla f(x^t), x^{t+1} - x^t \rangle + \frac{L}{2} \left\| x^{t+1} - x^t \right\|^2$$

$$= f(x^t) - \gamma \langle \nabla f(x^t), g^t \rangle + \frac{L\gamma^2}{2} \left\| g^t \right\|^2. \tag{52}$$

Let us take expectation of both sides of (52) conditioned on $x^t$ and apply Assumption 9:

$$\mathbb{E}\left[f(x^{t+1})|x^t\right] \leq f(x^t) - \gamma b \left\| \nabla f(x^t) \right\|^2 + c\gamma$$

$$+ \frac{L\gamma^2}{2} \left( 2A(f(x^t) - f^*) + B \left\| \nabla f(x^t) \right\|^2 + C \right)$$

$$= f(x^t) - \gamma \left( b - \frac{LB\gamma}{2} \right) \left\| \nabla f(x^t) \right\|^2$$

$$+ LA\gamma^2 \left( f(x^t) - f^* \right) + \frac{LC\gamma^2}{2} + c\gamma. \tag{53}$$

Subtract $f^*$ from both sides. Take expectation on both sides and use the tower property. For every $t \geq 0$, put $\delta^t \stackrel{\text{def}}{=} \mathbb{E}\left[f(x^t) - f^*\right]$ and $r^t \stackrel{\text{def}}{=} \mathbb{E}\left[\left\| \nabla f(x^t) \right\|^2\right]$. We obtain that

$$\gamma \left( b - \frac{LB\gamma}{2} \right) r^t \leq \left( 1 + LA\gamma^2 \right) \delta^t - \delta^{t+1} + \frac{LC\gamma^2}{2} + c\gamma.$$

Due to our choice of stepsize (51), we obtain that

$$\frac{\gamma b}{2} r^t \leq \left( 1 + LA\gamma^2 \right) \delta^t - \delta^{t+1} + \frac{LC\gamma^2}{2} + c\gamma. \tag{54}$$

Fix $w_{-1} > 0$ and, for all $t \geq 0$, define $w_t = \frac{w_{t-1}}{1+LA\gamma^2}$. Multiplying both sides of (54) by $\frac{w_t}{\gamma}$, we obtain

$$\frac{b w_t r^t}{2} \leq \frac{w_{t-1}}{\gamma} \delta^t - \frac{w_t}{\gamma} \delta^{t+1} + \frac{LC\gamma w_t}{2} + \frac{c w_t}{2}.$$

For every $0 \leq t \leq T - 1$, sum these inequalities. We arrive at

$$\frac{b}{2} \sum_{t=0}^{T-1} w_t r^t \leq \frac{w_{-1}}{\gamma} \delta^0 - \frac{w_{T-1}}{\gamma} \delta^T + \frac{LC\gamma + c}{2} \sum_{t=0}^{T-1} w_t. \tag{55}$$

∎

## E.5 Proof of Theorem 3

From (55) we derive that

$$\frac{b}{2} \sum_{t=0}^{T-1} w_t r^t \leq \frac{w_{-1}}{\gamma} \delta^0 + \frac{LC\gamma + c}{2} \sum_{t=0}^{T-1} w_t. \tag{56}$$

Observe that we can obtain the following lower bound on a sum of weights:

$$\sum_{t=0}^{T-1} w_t \geq T w_{T-1} = \frac{T w_{-1}}{\left(1 + LA\gamma^2\right)^T}.$$

Dividing both parts of (56) by $\sum_{t=0}^{T-1} w_t$ and using the lower bound on it, we get the statement of Theorem 3:

$$\min_{0 \leq t \leq T-1} r^t \leq \frac{2\left(1 + LA\gamma^2\right)^T}{b\gamma T} \delta^0 + \frac{LC\gamma}{b} + \frac{c}{b}.$$

■

## E.6  Proof of Corollary 4

We bound each term in the right-hand side of (20) by $\frac{c}{b}$.

If $\gamma = \frac{1}{\sqrt{LAT}}$, and if $T \geq \frac{36\left(\delta^0\right)^2 LA}{c^2}$, then we have

$$\frac{2\left(1 + LA\gamma^2\right)^T}{b\gamma T} \delta^0 \leq \frac{6\delta^0 \sqrt{LA}}{b\sqrt{T}} \leq \frac{c}{b}.$$

If $\gamma = \frac{b}{LB}$, and if $T \geq \frac{6LB\delta^0}{bc}$, then we obtain

$$\frac{2\left(1 + LA\gamma^2\right)^T}{b\gamma T} \delta^0 \leq \frac{6LB\delta^0}{bT} \leq \frac{c}{b}.$$

If $\gamma = \frac{c}{LC}$, and if $T \geq \frac{6LC\delta^0}{c^2}$, then we obtain

$$\frac{2\left(1 + LA\gamma^2\right)^T}{\gamma T} \delta^0 \leq \frac{6LC\delta^0}{bcT} \leq \frac{c}{b}.$$

Due to the choice of $\gamma$, we have $\frac{LC\gamma}{b} \leq \frac{c}{b}$. The last term is $\frac{c}{b}$ itself.

Therefore, we obtain

$$\min_{0 \leq t \leq T-1} \mathbb{E}\left[\left\|\nabla f(x^t)\right\|^2\right] \leq \frac{3c}{b}.$$

■

## E.7  Proof of Corollary 5

The proof is easy: one needs to substitute $b$ for 1 and $c$ for 0 in (20).

■

## E.8  Proof of Corollary 6

We bound each term in the right-hand side of (50) by $\frac{\varepsilon^2}{2}$.

If $\gamma = \frac{1}{\sqrt{LAT}}$, and if $T \geq \frac{144\left(\delta^0\right)^2 LA}{\varepsilon^4}$, then we have

$$\frac{2\left(1 + LA\gamma^2\right)^T}{\gamma T} \delta^0 \leq \frac{6\delta^0 \sqrt{LA}}{\sqrt{T}} \leq \frac{\varepsilon^2}{2}.$$

If $\gamma = \frac{1}{LB}$, and if $T \geq \frac{12LB\delta^0}{\varepsilon^2}$, then we obtain

$$\frac{2\left(1 + LA\gamma^2\right)^T}{\gamma T} \delta^0 \leq \frac{6LB\delta^0}{T} \leq \frac{\varepsilon^2}{2}.$$

If $\gamma = \frac{\varepsilon}{2LC}$, and if $T \geq \frac{24LC\delta^0}{\varepsilon^4}$, then we obtain

$$\frac{2\left(1 + LA\gamma^2\right)^T}{\gamma T}\delta^0 \leq \frac{12LC\delta^0}{\varepsilon^2 T} \leq \frac{\varepsilon^2}{2}.$$

Due to the choice of $\gamma$, we have $LC\gamma \leq \frac{\varepsilon^2}{2}$.

Therefore, we obtain

$$\min_{0 \leq t \leq T-1} \mathbb{E}\left[\left\|\nabla f(x^t)\right\|\right] \leq \varepsilon.$$

■

## E.9 Proof of Corollary 7

When $A = 0$, from (20) we have that

$$\min_{0 \leq t \leq T-1} r^t \leq \frac{2}{b\gamma T}\delta^0 + \frac{LC\gamma}{b} + \frac{c}{b}.$$

If $\gamma = \frac{\varepsilon b}{2LC}$ and $T \geq \frac{8\delta^0 LC}{b^2\varepsilon^2}$, then we get that

$$\frac{2}{b\gamma T}\delta^0 = \frac{4LC\delta^0}{b^2 T\varepsilon} \leq \frac{\varepsilon}{2}, \qquad \frac{LC\gamma}{b} \leq \frac{\varepsilon}{2}.$$

If $\gamma = \frac{b}{LB}$ and $T \geq \frac{4\delta^0 LB}{b^2 T}$, then we obtain that

$$\frac{2}{b\gamma T}\delta^0 = \frac{2LB\delta^0}{b^2 T} \leq \frac{\varepsilon}{2}, \qquad \frac{LC\gamma}{b} = \frac{LC}{b} \cdot \frac{b}{LB} \leq \frac{LC}{b} \cdot \frac{\varepsilon b}{2LC} = \frac{\varepsilon}{2}.$$

It follows that $\min_{0 \leq t \leq T-1} r^t = \mathcal{O}\left(\varepsilon + \frac{c}{b}\right).$

■

## E.10 Proof of Corollary 8

It follows from (20), that when $A = c = 0$, holds

$$\min_{0 \leq t \leq T-1} \mathbb{E}\left[\left\|\nabla f(x^t)\right\|^2\right] \leq \frac{2\delta^0}{b\gamma T} + \frac{LC\gamma}{b}.$$

If $\gamma = \frac{b\varepsilon}{2LC}$, and $T \geq \frac{8L\delta^0 C}{b^2\varepsilon^2}$, then we have that

$$\frac{LC\gamma}{b} \leq \frac{\varepsilon}{2}, \qquad \frac{2\delta^0}{b\gamma T} = \frac{4\delta^0 LC}{b^2\varepsilon T} \leq \frac{\varepsilon}{2}.$$

if $\gamma = \frac{b}{LB}$, and $T \geq \frac{4\delta^0 LB}{b^2\varepsilon}$, then we obtain that

$$\frac{LC\gamma}{b} \leq \frac{b\varepsilon}{2LC} \cdot \frac{LC}{b} = \frac{\varepsilon}{2}, \qquad \frac{2\delta^0}{b\gamma T} = \frac{2\delta^0 LB}{b^2 T} \leq \frac{\varepsilon}{2}.$$

It follows that $\min_{0 \leq t \leq T-1} \mathbb{E}\left[\left\|\nabla f(x^t)\right\|^2\right] \leq \varepsilon.$

■

## F  Convergence under PŁ-condition (assumption 10)

In Section 6.2 we have formulated Theorem 4 on convergence of BiasedSGD under Biased ABC assumption and compared the rate obtained to the known convergence results subject to PŁ-condition. Below we present recent results, derive several corollaries from Theorem 4 and make a formal comparison of our results to the known results.

### F.1 Corollaries from Theorem 4

As before in the general nonconvex case, Theorem 4 guarantees the convergence towards some neghborhood of the $\varepsilon$-stationary point, that can not be made less than $\frac{c}{\mu b}$. Therefore, we have the following corollary.

**Corollary 9** *Choose stepsize $\gamma > 0$ as $\gamma = \min\left\{\frac{\mu b}{L(A+\mu B)}, \frac{1}{2\mu b}, \frac{2c}{LC}\right\}$. Then, if*

$$T \geq \max\left\{2, \frac{L(A+\mu B)}{\mu^2 b^2}, \frac{LC}{2c\mu b}\right\} \log \frac{\mu b \delta^0}{c},$$

*we have*

$$\mathbb{E}\left[f(x^T) - f^*\right] \leq \frac{3c}{\mu b}.$$

Without bias terms, we recover the best known rates under Polyak–Łojasiewicz condition (Karimi et al. [2016]) subject to milder conditions.

**Corollary 10** *Suppose $c = 0$. Choose the stepsize $\gamma > 0$ as $\gamma = \min\left\{\frac{\mu b}{L(A+\mu B)}, \frac{1}{2\mu b}, \frac{\varepsilon \mu b}{LC}\right\}$. Then, if*

$$T \geq \max\left\{2, \frac{L(A+\mu B)}{\mu^2 b^2}, \frac{LC}{\varepsilon \mu^2 b^2}\right\} \log \frac{2\delta^0}{\varepsilon},$$

*we have*

$$\mathbb{E}\left[f(x^T) - f^*\right] \leq \varepsilon.$$

Plugging in $A = 0$, we recover the result similar to the one obtained in Theorem 6 of [Ajalloeian and Stich, 2020]. However, we impose weaker assumptions (compare Biased ABC and BND in Figure 1; see also Claim 1).

**Corollary 11** *Suppose $A = 0, b \leq 1$. Choose stepsize $\gamma > 0$ as $\gamma = \min\left\{\frac{b}{BL}, \frac{\varepsilon \mu b + 2c}{LC}\right\}$. Then, for $\varepsilon > 0$, we have that*

$$\mathcal{T} = \mathcal{O}\left(\max\left\{\frac{B}{b}, \frac{C}{\varepsilon \mu b + 2c}\right\} \frac{\kappa}{b} \log \frac{2\delta^0}{\varepsilon}\right)$$

*iterations suffice for*

$$\mathbb{E}\left[f(x^T) - f^*\right] = \mathcal{O}\left(\varepsilon + \frac{2c}{\mu b}\right).$$

If we substitute $B$ for $2(M+1)(m+1)$, $C$ for $2(M+1)\varphi^2 + \sigma^2$, $b$ for $\frac{1-m}{2}$, $c$ for $\frac{\varphi^2}{2}$ in accordance with Theorem 13 (see also Table 1), Corollary 11 yields the rate of $\mathcal{O}\left(\max\left\{\frac{2(M+1)(m+1)}{1-m}, \frac{2(M+1)\varphi^2 + \sigma^2}{\epsilon \mu(1-m) + 2\varphi^2}\right\}\right) \frac{\kappa}{1-m} \log \frac{2\delta^0}{\varepsilon}$ which is worse by an additive term of $\mathcal{O}\left(\frac{(M+1)\varphi^2}{\epsilon \mu(1-m) + 2\varphi^2} \frac{\kappa}{1-m} \log \frac{2\delta^0}{\varepsilon}\right)$ than the rate granted by Theorem 6 of Ajalloeian and Stich [2020].

### F.2 Proof of Theorem 4.

Due to (53) and Assumption 10, we have

$$\mathbb{E}\left[f(x^{t+1})|x^t\right] \leq f(x^t) - 2\gamma\mu\left(b - \frac{LB\gamma}{2}\right)\left(f(x^t) - f^*\right)$$

$$+ 2\gamma^2 \frac{LA}{2}\left(f(x^t) - f^*\right) + \frac{LC\gamma^2}{2} + c\gamma$$

$$= f(x^t) - 2\gamma\left(f(x^t) - f^*\right)\left[\mu\left(b - \frac{LB\gamma}{2}\right) - \frac{LA\gamma}{2}\right] + \frac{LC\gamma^2}{2} + c\gamma.$$

Subtract $f^*$ from both sides. Take expectation of both sides and use the tower property. Applying inequality (21), we obtain

$$\mathbb{E}\left[f(x^{t+1}) - f^*\right] \leq (1 - \gamma\mu b)\mathbb{E}\left[f(x^t) - f^*\right] + \frac{LC\gamma^2}{2} + c\gamma.$$

Unrolling the recursion, we arrive at

$$\mathbb{E}\left[f(x^T) - f^*\right] \leq (1 - \gamma\mu b)^T \, \mathbb{E}\left[f(x^0) - f^*\right] + \frac{LC\gamma}{2\mu b} + \frac{c}{\mu b}.$$

∎

### F.3  Proof of Corollary 9

We bound every term of (22) by $\frac{c}{\mu b}$.

If $\gamma = \frac{\mu b}{L(A+\mu B)}$, and if $T \geq \frac{L(A+\mu B)}{\mu^2 b^2} \log \frac{\mu b \delta^0}{c}$, we have

$$(1 - \gamma\mu b)^T \, \delta^0 = \left(1 - \frac{1}{L(A+\mu B)}\right)^T \delta^0 \leq e^{-\frac{T}{L(A+\mu B)}} \delta^0 \leq \frac{c}{\mu b}.$$

If $\gamma = \frac{1}{2\mu b}$, and if $T \geq 2 \log \frac{\mu b \delta^0}{c}$, we have

$$(1 - \gamma\mu b)^T \, \delta^0 \leq e^{-\frac{T}{2}} \delta^0 \leq \frac{c}{\mu b}.$$

If $\gamma = \frac{2c}{LC}$, and if $T \geq \frac{LC}{2c\mu b} \log \frac{\mu b \delta^0}{c}$, we have

$$(1 - \gamma\mu b)^T \, \delta^0 = \left(1 - \frac{2c\mu b}{LC}\right)^T \delta^0 \leq e^{-\frac{2c\mu b T}{LC}} \delta^0 \leq \frac{c}{\mu b}.$$

Due to the choice of $\gamma$, we have $\frac{LC\gamma}{2\mu b} \leq \frac{c}{\mu b}$.

Therefore, we obtain that $\mathbb{E}\left[f(x^T) - f^*\right] \leq \frac{3c}{\mu b}$.

∎

### F.4  Proof of Corollary 10

If we substitute $c$ for $0$ in (22), then, for every $T \geq 1$, we obtain

$$\mathbb{E}\left[f(x^T) - f^*\right] \leq (1 - \gamma\mu b)^T \, \delta^0 + \frac{LC\gamma}{2\mu b}.$$

We bound every term in the right-hand side of the latter inequality by $\frac{\varepsilon}{2}$.

If $\gamma = \frac{\mu b}{L(A+\mu B)}$, and if $T \geq \frac{L(A+\mu B)}{\mu^2 b^2} \log \frac{2\delta^0}{\varepsilon}$, then we have

$$(1 - \gamma\mu b)^T \, \delta^0 = \left(1 - \frac{\mu^2 b^2}{L(A+\mu B)}\right)^T \delta^0 \leq e^{-\frac{\mu^2 b^2 T}{L(A+\mu B)}} \delta^0 \leq \frac{\varepsilon}{2}.$$

If $\gamma = \frac{1}{2\mu b}$, and if $T \geq 2 \log \frac{2\delta^0}{\varepsilon}$,

$$(1 - \gamma\mu b)^T \, \delta^0 \leq e^{-\frac{T}{2}} \delta^0 \leq \frac{\varepsilon}{2}.$$

If $\gamma = \frac{\varepsilon\mu b}{LC}$, and if $T \geq \frac{LC}{\varepsilon\mu^2 b^2} \log \frac{2\delta^0}{\varepsilon}$, then we have

$$(1 - \gamma\mu b)^T \, \delta^0 = \left(1 - \frac{\mu^2 b^2 \varepsilon}{LC}\right)^T \delta^0 \leq e^{-\frac{\mu^2 b^2 \varepsilon T}{LC}} \delta^0 \leq \frac{\varepsilon}{2}.$$

Due to the choice of $\gamma$, we have $\frac{LC\gamma}{2\mu b} \leq \frac{\varepsilon}{2}$.

Then, if

$$T \geq \max\left\{2, \frac{L(A+\mu B)}{\mu^2 b^2}, \frac{LC}{\varepsilon\mu^2 b^2}\right\} \log \frac{2\delta^0}{\varepsilon},$$

we obtain $\mathbb{E}\left[f(x^T) - f^*\right] \leq \varepsilon$.

∎

### F.5 Proof of Corollary 11

From (22), when $A = 0$, $b \leq 1$, $0 < \gamma < \min\left\{\frac{b}{LB}, \frac{1}{\mu b}\right\}$, for every $T \geq 1$, we have

$$\mathbb{E}\left[f(x^T) - f^*\right] \leq (1 - \gamma\mu b)^T \delta^0 + \frac{LC\gamma}{2\mu b} + \frac{c}{\mu b}.$$

Observe that $\frac{b}{LB} \leq \frac{1}{\mu b}$. Let $\gamma = \min\left\{\frac{b}{LB}, \frac{\varepsilon\mu b + 2c}{LC}\right\}$.

If minimum is attained when $\gamma = \frac{b}{BL}$, then we have that $\frac{C}{\mu B} - \frac{2c}{\mu b} \leq \varepsilon$. If $T \geq \frac{B}{b}\frac{\kappa}{b}\log\frac{2\delta^0}{\varepsilon}$, then

$$(1 - \gamma\mu b)^T \delta^0 + \frac{LC\gamma}{2\mu b} + \frac{c}{\mu b} \leq e^{-\frac{T\mu b^2}{BL}}\delta^0 + \frac{C}{2\mu B} + \frac{c}{\mu b} \leq \varepsilon + \frac{2c}{\mu b}.$$

If $\gamma = \frac{\varepsilon\mu b + 2c}{LC}$ and $T \geq \frac{C}{\varepsilon\mu b + 2c}\frac{\kappa}{b}\log\frac{2\delta^0}{\varepsilon}$, then

$$(1 - \gamma\mu b)^T \delta^0 + \frac{LC\gamma}{2\mu b} + \frac{c}{\mu b} \leq e^{-\frac{T\mu b(\varepsilon\mu b + 2c)}{LC}}\delta^0 + \frac{\varepsilon}{2} + \frac{c}{\mu b} + \frac{c}{\mu b} = \varepsilon + \frac{2c}{\mu b}.$$

Then, if $T \geq \max\left\{\frac{B}{b}, \frac{C}{\varepsilon\mu b + 2c}\right\}\frac{\kappa}{\varepsilon}\log\frac{2\delta^0}{\varepsilon}$, then $\mathbb{E}\left[f(x^T) - f^*\right] = \mathcal{O}\left(\varepsilon + \frac{2c}{\mu b}\right)$.

∎

## G   Strongly convex case

In Section 6.3 we have stated that Theorem 4 on convergence of BiasedSGD under Biased ABC assumption can be applied in strongly convex settings. We compared the rate obtained to the known convergence results in strongly convex scenario. Below we present recent results, derive several corollaries from Theorem 4 and make a formal comparison of our results to the known results.

### G.1   Known results for convergence in function values

The next theorem is Theorem 4.6 from [Bottou et al., 2018].

**Theorem 8** *Let Assumptions 0, 8 and 11 hold. Then, as long as $0 < \gamma \leq \frac{q}{L(U+u^2)}$, for all $T \geq 1$, we have*

$$\mathbb{E}\left[f(x^T) - f(x^*)\right] \leq (1 - \gamma\mu q)^T\left(\delta^0 - \frac{\gamma LQ}{2\mu q}\right) + \frac{\gamma LQ}{2\mu q}.$$

Let us derive the convergence rate in Theorem 8 to compare it to our result obtained in the next section.

**Corollary 12** *Choose stepsize $\gamma > 0$ as $\gamma = \min\left\{\frac{q}{L(U+u^2)}, \frac{\varepsilon\mu q}{LQ}, \frac{1}{2\mu q}\right\}$. Then, if*

$$T \geq \max\left\{2, \frac{L\left(U + u^2\right)}{q^2\mu}, \frac{LQ}{\varepsilon\mu^2 q^2}\right\}\log\frac{2\delta^0}{\varepsilon},$$

*we have*

$$\mathbb{E}\left[f(x^T) - f(x^*)\right] \leq \varepsilon.$$

Next three theorems are analogues of Theorems 12 – 14 from [Beznosikov et al., 2020] respectively.

**Theorem 9** *Let Assumptions 0 and 11 hold. Let $g \in \mathbb{B}^1(\alpha, \beta)$ (that is, let Assumption 1 be satisfied). Then as long as $0 \leq \gamma \leq \frac{2}{\beta L}$, for all $t \in \mathbb{N}$, we have*

$$\mathbb{E}\left[f(x^t) - f(x^*)\right] \leq \left(1 - \frac{\alpha}{\beta}\gamma\mu(2 - \gamma\beta L)\right)^t\left(f(x^0) - f(x^*)\right).$$

*If we choose $\gamma = \frac{1}{\beta L}$, then*

$$\mathbb{E}\left[f(x^t) - f(x^*)\right] \leq \left(1 - \frac{\alpha}{\beta^2}\frac{\mu}{L}\right)^t\left(f(x^0) - f(x^*)\right).$$

**Theorem 10** *Let Assumptions 0 and 11 hold. Let $g \in \mathbb{B}^2(\tau, \beta)$ (that is, let Assumption 2 be satisfied). Then as long as $0 \leq \gamma \leq \frac{2}{\beta L}$, for all $t \in \mathbb{N}$, we have*

$$\mathbb{E}\left[f(x^t) - f(x^*)\right] \leq (1 - \tau\gamma\mu(2 - \gamma\beta L))^t \left(f(x^0) - f(x^*)\right).$$

*If we choose $\gamma = \frac{1}{\beta L}$, then*

$$\mathbb{E}\left[f(x^t) - f(x^*)\right] \leq \left(1 - \frac{\tau}{\beta}\frac{\mu}{L}\right)^t \left(f(x^0) - f(x^*)\right).$$

**Theorem 11** *Let Assumptions 0 and 11 hold. Let $g \in \mathbb{B}^3(\delta)$ (that is, let Assumption 3 be satisfied). Then as long as $0 \leq \gamma \leq \frac{1}{L}$, for all $t \in \mathbb{N}$, we have*

$$\mathbb{E}\left[f(x^t) - f(x^*)\right] \leq \left(1 - \frac{\gamma\mu}{\delta}\right)^t \left(f(x^0) - f(x^*)\right).$$

*If we choose $\gamma = \frac{1}{L}$, then*

$$\mathbb{E}\left[f(x^t) - f(x^*)\right] \leq \left(1 - \frac{\mu}{\delta L}\right)^t \left(f(x^0) - f(x^*)\right).$$

The authors of [Beznosikov et al., 2020] make the following observation. For every gradient estimator $g \in \mathbb{B}^1(\alpha, \beta)$, there exists a unique gradient estimator $\frac{1}{\beta}g \in \mathbb{B}^3\left(\frac{\beta^2}{\alpha}\right)$. By Theorem 11, we get the bound of $\mathcal{O}\left(\frac{\beta^2}{\alpha}\frac{L}{\mu}\log\frac{1}{\varepsilon}\right)$ on $\mathcal{T}$ which coincides with the result of Theorem 9 applied to $g$. If $g \in \mathbb{B}^3(\delta)$, then $g \in \mathbb{B}^1\left(\frac{1}{4\delta^2}, 2\right)$. Applying Theorem 9, we get that $\mathcal{O}\left(16\delta^2\frac{L}{\mu}\log\frac{1}{\varepsilon}\right)$ which is worse than the result of Theorem 11 by a factor of $16\delta$. For every $g \in \mathbb{B}^2(\tau, \beta)$, there exists a unique $g \in \mathbb{B}^1\left(\tau^2, \beta\right)$. Applying Theorem 10 we obtain $\mathcal{O}\left(\frac{\beta}{\tau}\frac{L}{\mu}\log\frac{1}{\varepsilon}\right)$, whence applying Theorem 9 we obtain $\mathcal{O}\left(\frac{\beta^2}{\tau^2}\frac{L}{\mu}\log\frac{1}{\varepsilon}\right)$. Since $\beta \geq \tau$, the second result is worse by a factor of $\frac{\beta}{\tau}$.

### G.2 Convergence in function values: our results

Observe that Assumption 10 is more general than Assumption 11. Therefore, Theorem 4 can be applied to functions that satisfy Assumption 11.

**Theorem 12** *Let Assumptions 0, 9 and 11 hold. Choose a stepsize such that*

$$0 < \gamma < \min\left\{\frac{\mu b}{L(A + \mu B)}, \frac{1}{\mu b}\right\}.$$

*Then, for every $T \geq 1$, we have*

$$\mathbb{E}\left[f(x^T) - f(x^*)\right] \leq (1 - \gamma\mu b)^T \delta^0 + \frac{LC\gamma}{2\mu b} + \frac{c}{\mu b}, \tag{57}$$

*where $\delta^0 = f(x^0) - f(x^*)$.*

Clearly, all of the corollaries from Theorem 4 hold in the strongly convex setup as well. Therefore, we do not write them here again.

Observe that if $A = c = 0$, we recover the result of Theorem 8 (see Theorem 4.6 from [Bottou et al., 2018]).

**Corollary 13** *Suppose $A = c = 0$. Choose stepsize $\gamma > 0$ as $\gamma = \min\left\{\frac{b}{LB}, \frac{\varepsilon b\mu}{LC}, \frac{1}{2\mu b}\right\}$. Then, if*

$$T \geq \max\left\{2, \frac{LB}{b^2\mu}, \frac{LC}{\varepsilon b^2\mu^2}\right\}\log\frac{2\delta^0}{\varepsilon},$$

*we have*

$$\mathbb{E}\left[f(x^T) - f(x^*)\right] \leq \varepsilon.$$

To recover the result from Corollary 12, one needs to substitute $B$ for $U + u^2$, $C$ for $Q$, $b$ for $q$ in accordance with the representation of Assumption 8 in Biased ABC framework (see Theorem 13 and Table 1).

Observe that if $A = C = c = 0$, we retrieve the results similar to Theorems 9 – 11.

**Corollary 14** *Suppose $A = C = c = 0$. Choose stepsize $\gamma > 0$ as $\gamma = \frac{b}{LB}$. Then, for every $T \geq 1$, we have*

$$\mathbb{E}\left[f(x^T) - f(x^*)\right] \leq \left(1 - \frac{b^2 \mu}{BL}\right)^T \delta^0.$$

*If $T \geq \frac{BL}{b^2 \mu} \log \frac{\delta^0}{\varepsilon}$, then we have*

$$\mathbb{E}\left[f(x^T) - f(x^*)\right] \leq \varepsilon.$$

If we substitute $B$ for $\beta^2$, $b$ for $\frac{\alpha}{\beta}$ (see Theorem 13 and Table 1), Corollary 14 yields the rate of $\mathcal{O}\left(\frac{\beta^4}{\alpha^2} \frac{L}{\mu} \log \frac{\delta^0}{\varepsilon}\right)$, which is worse by a factor of $\frac{\beta^2}{\alpha}$ than the rate granted by Theorem 9 [Beznosikov et al., 2020, Theorem 12].

If we substitute $B$ for $\beta^2$, $b$ for $\tau$ (see Theorem 13 and Table 1), Corollary 14 yields the rate of $\mathcal{O}\left(\frac{\beta^2}{\tau^2} \frac{L}{\mu} \log \frac{\delta^0}{\varepsilon}\right)$, which is worse by a factor of $\frac{\beta}{\tau}$ than the rate granted by Theorem 10 [Beznosikov et al., 2020, Theorem 13].

If we substitute $B$ for $2\left(2 - \frac{1}{\delta}\right)$, $b$ for $\frac{1}{2\delta}$ (see Theorem 13 and Table 1), Corollary 14 yields the rate of $\mathcal{O}\left(\delta^2 \frac{L}{\mu} \log \frac{\delta^0}{\varepsilon}\right)$, which is worse by a factor of $\delta$ than the rate granted by Theorem 11 [Beznosikov et al., 2020, Theorem 14].

### G.3 Proof of Corollary 12

If $\gamma = \frac{q}{L(U+u^2)}$ and $T \geq \frac{L(U+u^2)}{q^2 \mu} \log \frac{2\delta^0}{\varepsilon}$, then

$$(1 - \gamma\mu q)^T \left(\delta^0 - \frac{\gamma L Q}{2\mu q}\right) \leq \left(1 - \frac{q^2 \mu}{L(U + u^2)}\right)^T \delta^0 \leq e^{-\frac{q^2 \mu T}{L(U+u^2)}} \delta^0 \leq \frac{\varepsilon}{2}.$$

If $\gamma = \frac{\varepsilon \mu q}{LQ}$ and $T \geq \frac{LQ}{\varepsilon \mu^2 q^2} \log \frac{2\delta^0}{\varepsilon}$, then

$$(1 - \gamma\mu q)^T \left(\delta^0 - \frac{\gamma L Q}{2\mu q}\right) \leq \left(1 - \frac{\varepsilon \mu^2 q^2}{LQ}\right)^T \delta^0 \leq e^{-\frac{\mu^2 q^2 T}{LQ}} \delta^0 \leq \frac{\varepsilon}{2}.$$

If $\gamma = \frac{1}{2\mu q}$ and $T \geq 2 \log \frac{2\delta^0}{\varepsilon}$, then

$$(1 - \gamma\mu q)^T \left(\delta^0 - \frac{\gamma L Q}{2\mu q}\right) \leq e^{-\frac{T}{2}} \delta^0 \leq \frac{\varepsilon}{2}.$$

Due to the choice of $\gamma$, we have $\frac{\gamma L Q}{2\mu q} \leq \frac{\varepsilon}{2}$.

Then, if

$$T \geq \max\left\{2, \frac{L(U + u^2)}{q^2 \mu}, \frac{LQ}{\varepsilon \mu^2 q^2}\right\} \log \frac{2\delta^0}{\varepsilon},$$

we obtain $\mathbb{E}\left[f(x^T) - f(x^*)\right] \leq \varepsilon$.

∎

### G.4 Proof of Theorem 12

Follow exactly the same steps as in the proof of Theorem 4.

∎

## G.5 Proof of Corollary 13

If $\gamma = \frac{b}{LB}$ and $T \geq \frac{LB}{b^2\mu} \log \frac{2\delta^0}{\varepsilon}$, then

$$(1 - \gamma\mu b)^T \delta^0 = \left(1 - \frac{b^2\mu}{LB}\right)^T \delta^0 \leq e^{-\frac{Tb^2\mu}{LB}} \delta^0 \leq \frac{\varepsilon}{2}.$$

If $\gamma = \frac{\varepsilon b\mu}{LC}$ and $T \geq \frac{LC}{\varepsilon b^2\mu^2} \log \frac{2\delta^0}{\varepsilon}$, then

$$(1 - \gamma\mu b)^T \delta^0 = \left(1 - \frac{\varepsilon b^2\mu^2}{LC}\right)^T \delta^0 \leq e^{-\frac{T\varepsilon b^2\mu^2}{LC}} \delta^0 \leq \frac{\varepsilon}{2}.$$

If $\gamma = \frac{1}{2\mu b}$ and $T \geq 2 \log \frac{2\delta^0}{\varepsilon}$, then

$$(1 - \gamma\mu b)^T \delta^0 \leq e^{-\frac{T}{2}} \delta^0 \leq \frac{\varepsilon}{2}.$$

Due to the choice of $\gamma$, we have $\frac{LC\gamma}{2\mu b} \leq \frac{\varepsilon}{2}$. Then, if

$$T \geq \max\left\{2, \frac{LB}{b^2\mu}, \frac{LC}{\varepsilon b^2\mu^2}\right\} \log \frac{2\delta^0}{\varepsilon},$$

we obtain $\mathbb{E}\left[f(x^T) - f(x^*)\right] \leq \varepsilon$.

∎

## G.6 Proof of Corollary 14

Consider (57) and recall that $A = C = c = 0$. Note that in this case $\frac{\mu b}{L(A+\mu B)} = \frac{b}{LB}$ is no greater that $\frac{1}{\mu b}$. Indeed,

$$b \|\nabla f(x)\|^2 \leq \langle \mathbb{E}[g(x)], \nabla f(x)\rangle \leq \|\mathbb{E}[g(x)]\| \cdot \|\nabla f(x)\|$$

(by Cauchy–Schwarz inequality), which (combined with Biased ABC) leads to

$$b^2 \|\nabla f(x)\|^2 \leq \|\mathbb{E}[g(x)]\|^2 \leq \mathbb{E}\left[\|g(x)\|^2\right] \leq B \|\nabla f(x)\|^2.$$

Therefore, we have that $b^2 \leq B$. Then, $b^2 \leq \frac{L}{\mu}B \iff \frac{b}{LB} \leq \frac{1}{\mu b}$.

Hence, we can choose $\gamma = \frac{b}{LB}$, which yields that

$$\mathbb{E}\left[f(x^T) - f(x^*)\right] \leq \left(1 - \frac{b^2\mu}{LB}\right)^T \delta^0.$$

If $T \geq \frac{LB}{b^2\mu} \log \frac{\delta^0}{\varepsilon}$, then

$$\left(1 - \frac{b^2\mu}{LB}\right)^T \delta^0 \leq e^{-\frac{Tb^2\mu}{LB}} \delta^0 \leq \varepsilon.$$

∎

## G.7 Iterate convergence: further discussion

In Section 6.3 we introduce strict Assumption 12 and formulate convergence Theorem 5 subject to this condition. It is reasonable to ask whether Assumption 12 is realistic. In this part of the appendix we give a useful example of a setting that meets the requirements of the assumption imposed.

It is easy to see that Assumption 12 holds only when $b$ is relatively large, and $A$ is small, which is not necessarily the case in practice. However, let us show that it can be satisfied. Consider the $\ell_2$-regularized logistic regression with $f_j = \log\left(1 + e^{-b_j\langle e_j, x\rangle}\right) + \frac{1}{2}\|x\|^2$, where $e_j$ is the $j$-th unit vector, $b_j \in \{0, 1\}$, $j \in [n]$, $n \geq 2$. It is straightforward to show that all $f_j$ and $f = \frac{1}{n}\sum_{j=1}^n f_j$ are $\frac{5}{4}$-smooth and 1-strongly-convex. Consider the estimator from Definition 2, and let $a_k = k$, $k \in \mathbb{N} \cup \{0\}$, $p_j = \frac{1}{5}$. From (27)–(31), we obtain that $A_r = \frac{2}{n}$, $B_r = \frac{2}{5}$, $C_r = \frac{4\Delta^*}{n}$, $b_r = \frac{4}{5}$, $c_r = 0$. Then Assumption 12 holds since $\frac{2}{n} - \frac{1}{4} < 1$.

### G.8 Proof of Theorem 5

Let $r^t \stackrel{\text{def}}{=} x^t - x^*$. We get

$$\left\|r^{t+1}\right\|^2 = \left\|(x^t - \gamma g^t) - x^*\right\|^2 = \left\|x^t - x^* - \gamma g^t\right\|^2 = \left\|r^t\right\|^2 - 2\gamma\langle r^t, g^t\rangle + \gamma^2 \left\|g^t\right\|^2.$$

Now we compute expectation of both sides of the inequality, conditional on $x^t$ :

$$\mathbb{E}\left[\left\|r^{t+1}\right\|^2 | x^t\right] = \left\|r^t\right\|^2 - 2\gamma\langle r^t, \mathbb{E}[g^t | x^t]\rangle + \gamma^2 \mathbb{E}\left[\left\|g^t\right\|^2 | x^t\right].$$

Notice that

$$2\langle r^t, \mathbb{E}[g^t | x^t]\rangle = 2\langle r^t, \mathbb{E}[g^t | x^t] - \nabla f(x^t)\rangle + 2\langle r^t, \nabla f(x^t)\rangle.$$

Due to $\mu$-convexity, we have

$$\langle r^t, \nabla f(x^t)\rangle \geq D_f(x^t, x^*) + \frac{\mu}{2}\left\|r^t\right\|^2. \tag{58}$$

Further, using Young's Inequality (Lemma 3, (68)), we get

$$-2\langle r^t, \mathbb{E}[g^t | x^t] - \nabla f(x^t)\rangle \leq s\left\|r^t\right\|^2 + \frac{1}{s}\left\|\mathbb{E}[g^t | x^t] - \nabla f(x^t)\right\|^2. \tag{59}$$

Notice that

$$\begin{aligned}
\left\|\mathbb{E}[g^t | x^t] - \nabla f(x^t)\right\|^2 &= \left\|\mathbb{E}[g^t | x^t]\right\|^2 - 2\langle \mathbb{E}[g^t | x^t], \nabla f(x^t)\rangle + \left\|\nabla f(x^t)\right\|^2 \\
&\leq 2AD_f(x^t, x^*) + B\left\|\nabla f(x^t)\right\|^2 + C \\
&\quad - 2\left(b\left\|\nabla f(x^t)\right\|^2 - c\right) + \left\|\nabla f(x^t)\right\|^2.
\end{aligned}$$

Below we use this fact from Lemma 1:

$$\left\|\nabla f(x^t)\right\|^2 \leq 2LD_f(x^t, x^*). \tag{60}$$

This leads to

$$\begin{aligned}
\mathbb{E}\left[\left\|r^{t+1}\right\|^2 | x^t\right] &\stackrel{(58),(59)}{\leq} (1 - \gamma(\mu - s))\left\|r^t\right\|^2 - 2\gamma D_f(x^t, x^*) \\
&\quad + \gamma^2 \mathbb{E}\left[\left\|g^t\right\|^2 | x^t\right] + \frac{\gamma}{s}\left(\left\|\mathbb{E}[g^t | x^t] - \nabla f(x^t)\right\|^2\right) \\
&\stackrel{(60)}{\leq} (1 - \gamma(\mu - s))\left\|r^t\right\|^2 - 2\gamma D_f(x^t, x^*) \\
&\quad + \gamma^2\left(2AD_f(x^t, x^*) + B\left\|\nabla f(x^t)\right\|^2 + C\right) \\
&\quad + \frac{\gamma}{s}\left[2AD_f(x^t, x^*) + B\left\|\nabla f(x^t)\right\|^2 + C\right. \\
&\qquad \left. -2\left(b\left\|\nabla f(x^t)\right\|^2 - c\right) + \left\|\nabla f(x^t)\right\|^2\right] \\
&= (1 - \gamma(\mu - s))\left\|r^t\right\|^2 \\
&\quad - 2\gamma D_f(x^t, x^*)\left[1 - A\gamma - \frac{A}{s} - L\left(\gamma B + \frac{B}{s} - \frac{2b}{s} + \frac{1}{s}\right)\right] + \\
&\quad + \gamma^2 C + \frac{\gamma(C + 2c)}{s}.
\end{aligned}$$

Due to (23), we have

$$\mathbb{E}\left[\left\|r^{t+1}\right\|^2 | x^t\right] \leq (1 - \gamma(\mu - s))\left\|r^t\right\|^2 + \gamma^2 C + \frac{\gamma(C + 2c)}{s}.$$

Take expectation again on both sides and use the tower property

$$\mathbb{E}\left[\left\|r^{t+1}\right\|^2\right] = \mathbb{E}\left[\mathbb{E}\left[\left\|r^{t+1}\right\|^2 | x^t\right]\right].$$

We arrive at

$$\mathbb{E}\left[\left\|r^{t+1}\right\|^2\right] \leq (1 - \gamma\,(\mu - s))\,\mathbb{E}\left[\left\|r^t\right\|^2\right] + \gamma^2 C + \frac{\gamma\,(C + 2c)}{s}.$$

Unrolling the recurrence and noting that $\mathbb{E}\left[\left\|r^0\right\|^2\right] = \left\|r^0\right\|^2$ gives us

$$\mathbb{E}\left[\left\|r^t\right\|^2\right] \leq (1 - \gamma\,(\mu - s))^t\,\left\|r^0\right\|^2$$
$$+ \gamma\left(\gamma C + \frac{C + 2c}{s}\right)\sum_{i=0}^{t-1}(1 - \gamma\,(\mu - s))^i$$
$$\leq (1 - \gamma\,(\mu - s))^t\,\left\|r^0\right\|^2 + \frac{\gamma C + \frac{C+2c}{s}}{\mu - s}.$$

∎

# H    Assumptions 1–8 in biased ABC framework

In Table 1 we have presented the values of control variables $A, B, C, b$ and $c$ in our Biased ABC framework for a gradient estimator that satisfies any of assumptions listed in Section 4. Here we give a formal proof of these results.

**Theorem 13** *The following relations hold.*

*i  Suppose $g(x)$ satisfies Assumption 1. Then it satisfies Assumption 9 with $A = 0, B = \beta^2, C = 0, b = \frac{\alpha}{\beta}, c = 0$.*

*ii  Suppose $g(x)$ satisfies Assumption 2. Then it satisfies Assumption 9 with $A = 0, B = \beta^2, C = 0, b = \tau, c = 0$.*

*iii  Suppose $g(x)$ satisfies Assumption 3. Then it satisfies Assumption 9 with $A = C = c = 0, B = 2\left(2 - \frac{1}{\delta}\right), b = \frac{1}{2\delta}$.*

*iv  Suppose $g(x)$ satisfies Assumption 4. Then it satisfies Assumption 9 with $A = C = c = 0, B = 2(1 + \xi + \eta), b = \frac{1-\eta}{2}$.*

*v  Suppose $g(x)$ satisfies Assumption 5. Then it satisfies Assumption 9 with $A = 0, B = \zeta, C = 0, b = \rho, c = 0$.*

*vi  Suppose $g(x)$ satisfies Assumption 6.  Then it satisfies Assumption 9 with $A = 0, B = 2(M + 1)(m + 1), C = 2(M + 1)\varphi^2 + \sigma^2, b = \frac{1-m}{2}, c = \frac{\varphi^2}{2}$.*

*vii  Suppose $g(x)$ satisfies Assumption 7. Then it satisfies Assumption 9 with $A = 0, B = 2, C = 2\Delta^2, b = \frac{1}{2}, c = \frac{\Delta^2}{2}$.*

*viii  Suppose $g(x)$ satisfies Assumption 8. Then it satisfies Assumption 9 with $A = 0, B = U + u^2, C = Q, b = q, c = 0$.*

**Proof of Theorem 13.** Let us prove all of the assertions stated in Theorem 13 one by one.
i From (3), we deirve that $\langle\nabla f(x, \mathbb{E}\left[g(x)\right])\rangle \geq \frac{\alpha}{\beta}\left\|\nabla f(x)\right\|^2$. Therefore, we can choose $b = \frac{\alpha}{\beta}$, $c = 0$. From (5), we obtain that $A = 0, B = \beta^2, C = 0$.

ii From (4), we derive that $\langle\nabla f(x, \mathbb{E}\left[g(x)\right])\rangle \geq \tau\left\|\nabla f(x)\right\|^2$. Therefore, we can choose $b = \tau$, $c = 0$. From (5), we obtain that $A = 0, B = \beta^2, C = 0$.

iii From (6), we derive that

$$\langle\mathbb{E}\left[g(x)\right], \nabla f(x)\rangle \geq \frac{1}{2}\left(\mathbb{E}\left[\left\|g(x)\right\|^2\right] + \frac{1}{\delta}\left\|\nabla f(x)\right\|^2\right) \geq \frac{1}{2\delta}\left\|\nabla f(x)\right\|^2.$$

Further,

$$\mathbb{E}\left[\|g(x)\|^2\right] = \mathbb{E}\left[\|g(x) - \nabla f(x) + \nabla f(x)\|^2\right]$$
$$\leq 2\mathbb{E}\left[\|g(x) - \nabla f(x)\|^2\right] + 2\|\nabla f(x)\|^2$$
$$\leq 2\left(2 - \frac{1}{\delta}\right)\|\nabla f(x)\|^2.$$

iv From (8), we derive that

$$\langle\mathbb{E}\left[g(x)\right], \nabla f(x)\rangle \geq \frac{1}{2}\left(\|\mathbb{E}\left[g(x)\right]\|^2 + (1 - \eta)\|\nabla f(x)\|^2\right) \geq \frac{1 - \eta}{2}\|\nabla f(x)\|^2.$$

Further, from (7), (8) and (9), we obtain that

$$\mathbb{E}\left[\|g(x)\|^2\right] = \mathbb{E}\left[\|g(x) - \nabla f(x) + \nabla f(x)\|^2\right]$$
$$\leq 2\mathbb{E}\left[\|g(x) - \nabla f(x)\|^2\right] + 2\|\nabla f(x)\|^2$$
$$\leq 2\left(1 + \xi + \eta\right)\|\nabla f(x)\|^2.$$

v From (10), we conclude that $b = \rho$, $c = 0$. From (11), we derive that $A = 0$, $B = \zeta$, $C = 0$.

vi It follows from the proof of Theorem 2–ix.

vii From (14), we have

$$\langle\mathbb{E}\left[g(x)\right], \nabla f(x)\rangle \geq \frac{1}{2}\left(\mathbb{E}\left[\|g(x)\|^2\right] + \|\nabla f(x)\|^2\right) - \frac{\Delta^2}{2} \geq \frac{1}{2}\|\nabla f(x)\|^2 - \frac{\Delta^2}{2},$$
$$\mathbb{E}\left[\|g(x)\|^2\right] = \mathbb{E}\left[\|g(x) - \nabla f(x) + \nabla f(x)\|^2\right]$$
$$\leq 2\mathbb{E}\left[\|g(x) - \nabla f(x)\|^2\right] + 2\|\nabla f(x)\|^2$$
$$\leq 2\|\nabla f(x)\|^2 + 2\Delta^2.$$

viii It follows from the proof of Theorem 2–viii.

∎

# I  New estimators in biased ABC framework: proofs for Section B

In this section we prove the results announced in Section B.

## I.1  Proof of Claim 2

First, let us find constants for (18):

$$\langle\nabla f(x), \mathbb{E}\left[g(x)\right]\rangle = \left\langle\frac{1}{n}\sum_{i=1}^{n}\nabla f_i(x), \mathbb{E}\left[\frac{1}{|S|}\sum_{i=1}^{n}v_i\nabla f_i(x)\right]\right\rangle$$
$$\geq \left\langle\frac{1}{n}\sum_{i=1}^{n}\nabla f_i(x), \frac{1}{n}\sum_{i=1}^{n}\min\{p_i\}\nabla f_i(x)\right\rangle$$
$$\geq \min_{i}\{p_i\}\|\nabla f(x)\|^2.$$

Second, let us find an upper bound on the variance of the gradient estimator $g(x)$. Notice that, since $\tilde{g}(x)$ is independent of $X$, and $\mathbb{E}\left[X\right] = 0$, we can write that

$$\mathbb{E}\left[\|g(x)\|^2\right] = \mathbb{E}\left[\|\tilde{g}(x)\|^2\right] + \mathbb{E}\left[\|X\|^2\right] = \mathbb{E}\left[\|\tilde{g}(x)\|^2\right] + \sigma^2.$$

Clearly, $\mathbb{E}[\mathbb{I}_i] = p_i$. Note, that, for $i \neq j \in [n]$, random sets $S_i$ and $S_j$ are independent, random variables $\mathbb{I}_i$ and $\mathbb{I}_j$ are also independent. Therefore,

$$\mathbb{E}\left[\mathbb{I}_i \mathbb{I}_j\right] = \mathbb{E}[\mathbb{I}_i]\mathbb{E}[\mathbb{I}_i] = p_i p_j.$$

Further, let us bound the second moment of $\tilde{g}(x)$ from above:

$$
\begin{aligned}
\mathbb{E}\left[\|\tilde{g}(x)\|^2\right] &= \mathbb{E}\left[\left\|\frac{1}{|S|}\sum_{i=1}^{n}\mathbb{I}_i \nabla f_i(x)\right\|^2\right] \\
&\leq \mathbb{E}\left[\frac{1}{|S|}\sum_{i=1}^{n}\mathbb{I}_i \|\nabla f_i(x)\|^2\right] \\
&= \sum_{i=1}^{n}\mathbb{E}\left[\frac{\mathbb{I}_i}{|S|}\right]\|\nabla f_i(x)\|^2 \\
&\leq \sum_{i=1}^{n}\mathbb{E}\left[\frac{1}{|S|}\right]\|\nabla f_i(x)\|^2 \\
&\leq \frac{1}{n\min_i\{p_i\}}\sum_{i=1}^{n}\|\nabla f_i(x)\|^2.
\end{aligned}
$$

Due to Assumption 13, we obtain that

$$
\begin{aligned}
\mathbb{E}\left[\|\tilde{g}(x)\|^2\right] &\leq \frac{2\max_i\{L_i\}}{n\min_i\{p_i\}}\sum_{i=1}^{n}D_{f_i}(x, x^*) \\
&\leq \frac{2\max_i\{L_i\}}{\min_i\{p_i\}}D_f(x, x^*) + \frac{2\max_i\{L_i\}}{\min_i\{p_i\}}\Delta^*.
\end{aligned}
$$

Therefore, we can choose $A = \frac{\max_i\{L_i\}}{\min_i p_i}$, $B = 0$, $C = 2A\Delta^* + \sigma^2$, $b = \min_i\{p_i\}$, $c = 0$.

$\blacksquare$

## I.2   Proof of Claim 3

Let us establish (18) first:

$$\langle \nabla f(x), \mathbb{E}[g(x)]\rangle = \left\langle \nabla f(x), \frac{1}{n}\sum_{i=1}^{n}c_i \nabla f_i(x)\right\rangle \geq \min_i\{c_i\}\|\nabla f(x)\|^2.$$

Further, we establish (19). We use the convexity of the $k_2$-norm and Lemma 1.

$$
\begin{aligned}
\mathbb{E}\left[\|g(x)\|^2\right] &\leq \frac{1}{n}\sum_{i=1}^{n}\mathbb{E}\left[\|v_i \nabla f_i(x)\|^2\right] \\
&= \frac{1}{n}\sum_{i=1}^{n}\mathbb{E}\left[v_i^2\right]\|\nabla f_i(x)\|^2 \\
&\leq \frac{2\max_i\left\{L_i \mathbb{E}\left[v_i^2\right]\right\}}{n}\sum_{i=1}^{n}D_{f_i}(x, x^*) \\
&= 2\max_i\left\{L_i \mathbb{E}\left[v_i^2\right]\right\}D_f(x, x^*) + 2\max_i\left\{L_i \mathbb{E}\left[v_i^2\right]\right\}\Delta^*.
\end{aligned}
$$

$\blacksquare$

## I.3 Proof of Claim 4

First, we establish that (18) holds:

$$\langle \nabla f(x), \mathbb{E}\left[g(x)\right]\rangle = \left\langle \nabla f(x), \frac{1}{n}\sum_{j=1}^{n} p_j \tilde{g}_j(x)\right\rangle + \left\langle \nabla f(x), \frac{1}{n}\sum_{j=1}^{n}(1-p_j)\nabla f_j(x)\right\rangle$$

$$\geq \max_{j}\{p_j\}\langle \nabla f(x), \tilde{g}(x)\rangle + \max_{j}\{1-p_j\}\left\|\nabla f(x)\right\|^2$$

$$\geq \left(\max_{j}\{p_j\}\cdot \inf_{k\in\mathbb{Z}} \frac{2a_k}{a_k+a_{k+1}} + \max_{j}\{1-p_j\}\right)\left\|\nabla f(x)\right\|^2.$$

Further, we need to show that (19) is also valid.

$$\mathbb{E}\left[\left\|g(x)\right\|^2\right] = \mathbb{E}\left[\left\|\frac{1}{n}\sum_{j=1}^{n}\mathbb{I}_j \tilde{g}_j(x) + \frac{1}{n}\sum_{j=1}^{n}(1-\mathbb{I}_j)\nabla f_j(x)\right\|^2\right]$$

$$\leq 2\mathbb{E}\left[\left\|\frac{1}{n}\sum_{j=1}^{n}\mathbb{I}_j \tilde{g}_j(x)\right\|^2\right] + 2\mathbb{E}\left[\left\|\frac{1}{n}\sum_{j=1}^{n}(1-\mathbb{I}_j)\nabla f_j(x)\right\|^2\right]$$

$$= \frac{2}{n^2}\mathbb{E}\left[\left\|\sum_{j=1}^{n}\mathbb{I}_j \tilde{g}_j(x)\right\|^2\right] + \frac{2}{n^2}\mathbb{E}\left[\left\|\sum_{j=1}^{n}(1-\mathbb{I}_j)\nabla f_j(x)\right\|^2\right]. \tag{61}$$

Let us deal with each term separately. For the first one we have

$$\mathbb{E}\left[\left\|\sum_{j=1}^{n}\mathbb{I}_j \tilde{g}_j(x)\right\|^2\right] = \sum_{j=1}^{n}\mathbb{E}\left[\mathbb{I}_j^2\right]\left\|\tilde{g}_j\right\|^2 + 2\sum_{j\neq h}\mathbb{E}\left[\mathbb{I}_j\right]\mathbb{E}\left[\mathbb{I}_h\right]\langle \tilde{g}_j, \tilde{g}_h\rangle$$

$$= \sum_{j=1}^{n} p_j \left\|\tilde{g}_j\right\|^2 + 2\sum_{j\neq h} p_j p_h \langle \tilde{g}_j, \tilde{g}_h\rangle$$

$$= \sum_{j=1}^{n} p_j(1-p_j)\left\|\tilde{g}_j\right\|^2 + \left\|\sum_{j=1}^{n} p_j \tilde{g}_j\right\|^2.$$

From $L_j$-smoothness of $f_j(x)$, $j\in[n]$, and from Lemma 1, we have that

$$\mathbb{E}\left[\left\|\sum_{j=1}^{n}\mathbb{I}_j \tilde{g}_j(x)\right\|^2\right] \leq \max_{j}\{p_j(1-p_j)\}\left(\sup_{k\in\mathbb{Z}}\frac{2a_{k+1}}{a_k+a_{k+1}}\right)^2 \sum_{j=1}^{n}\left\|\nabla f_j(x)\right\|^2$$

$$+ n^2 \max_{j}\{p_j^2\}\left(\sup_{k\in\mathbb{Z}}\frac{2a_{k+1}}{a_k+a_{k+1}}\right)^2 \left\|\nabla f(x)\right\|^2$$

$$\leq 2\max_{j}\{p_j(1-p_j)\}\left(\sup_{k\in\mathbb{Z}}\frac{2a_{k+1}}{a_k+a_{k+1}}\right)^2 \sum_{j=1}^{n} L_j D_{f_j}(x,x^*)$$

$$+ n^2 \max_{j}\{p_j^2\}\left(\sup_{k\in\mathbb{Z}}\frac{2a_{k+1}}{a_k+a_{k+1}}\right)^2 \left\|\nabla f(x)\right\|^2$$

$$\leq 2n \max_{j}\{L_j\}\max_{j}\{p_j(1-p_j)\}\left(\sup_{k\in\mathbb{Z}}\frac{2a_{k+1}}{a_k+a_{k+1}}\right)^2 \cdot D_f(x,x^*)$$

$$+ 2n \max_{j}\{L_j\}\max_{j}\{p_j(1-p_j)\}\left(\sup_{k\in\mathbb{Z}}\frac{2a_{k+1}}{a_k+a_{k+1}}\right)^2 \Delta^*$$

$$+ n^2 \max_{j}\{p_j^2\}\left(\sup_{k\in\mathbb{Z}}\frac{2a_{k+1}}{a_k+a_{k+1}}\right)^2 \left\|\nabla f(x)\right\|^2.$$

For the second term in (61), we have

$$\mathbb{E}\left[\left\|\sum_{j=1}^{n}\left(1-\mathbb{I}_{j}\right)\nabla f_{j}(x)\right\|^{2}\right] = \sum_{j=1}^{n}\mathbb{E}\left[\left(1-\mathbb{I}_{j}\right)^{2}\right]\|\nabla f_{j}(x)\|^{2}$$

$$+ 2\sum_{j\neq h}\mathbb{E}\left[(1-\mathbb{I}_{j})\right]\mathbb{E}\left[(1-\mathbb{I}_{h})\right]\langle\nabla f_{j}(x),\nabla f_{h}(x)\rangle$$

$$= \sum_{j=1}^{n}(1-p_{j})\|\nabla f_{j}(x)\|^{2}$$

$$+ 2\sum_{j\neq h}(1-p_{j})(1-p_{h})\langle\nabla f_{j}(x),\nabla f_{h}(x)\rangle$$

$$= \sum_{j=1}^{n}(1-p_{j})p_{j}\|\nabla f_{j}(x)\|^{2} + \left\|\sum_{j=1}^{n}\left(1-p_{j}\right)\nabla f_{j}(x)\right\|^{2}$$

$$\leq \max_{j}\{p_{j}(1-p_{j})\}\sum_{j=1}^{n}\|\nabla f_{j}(x)\|^{2}$$

$$+ n^{2}\max_{j}\{(1-p_{j})^{2}\}\|\nabla f(x)\|^{2}.$$

Further, due to $L_{j}$-smoothness of $f_{j}$, $j\in[n]$, and due to Lemma 1, we obtain

$$\mathbb{E}\left[\left\|\sum_{j=1}^{n}\left(1-\mathbb{I}_{j}\right)\nabla f_{j}(x)\right\|^{2}\right] \leq 2\max_{j}\{p_{j}(1-p_{j})\}\sum_{j=1}^{n}L_{j}D_{f_{j}}(x,x^{*})$$

$$+ n^{2}\max_{j}\{(1-p_{j})^{2}\}\|\nabla f(x)\|^{2}$$

$$\leq 2n\max_{j}\{p_{j}(1-p_{j})\}\max_{j}\{L_{j}\}D_{f}(x,x^{*})$$

$$+ 2n\max_{j}\{p_{j}(1-p_{j})\}\max_{j}\{L_{j}\}\Delta^{*}$$

$$+ n^{2}\max_{j}\{(1-p_{j})^{2}\}\|\nabla f(x)\|^{2}$$

Therefore, from (61), we have

$$\mathbb{E}\left[\|g(x)\|^{2}\right] \leq \frac{4}{n}\max_{j}\{L_{j}\}\max_{j}\{p_{j}(1-p_{j})\}\left(\left(\sup_{k\in\mathbb{Z}}\frac{2a_{k+1}}{a_{k}+a_{k+1}}\right)^{2}+1\right)D_{f}\left(x,x^{*}\right)$$

$$+ 2\max_{j}\{p_{j}^{2}\}\left(\left(\sup_{k\in\mathbb{Z}}\frac{2a_{k+1}}{a_{k}+a_{k+1}}\right)^{2}+1\right)\|\nabla f(x)\|^{2}$$

$$+ \frac{4}{n}\max_{j}\{L_{j}\}\max_{j}\{p_{j}(1-p_{j})\}\left(\left(\sup_{k\in\mathbb{Z}}\frac{2a_{k+1}}{a_{k}+a_{k+1}}\right)^{2}+1\right)\Delta^{*}.$$

∎

# J   Known estimators in biased ABC framework: proofs for Section C

## J.1   Proof of Claim 5

Observe that

$$\frac{(\nabla f(x))_{(d-k+1)}^{2}+\ldots+(\nabla f(x))_{(d)}^{2}}{k} \geq \frac{(\nabla f(x))_{1}^{2}+\ldots+(\nabla f(x))_{d}^{2}}{d},$$

and
$$\langle g(x), \nabla f(x) \rangle = \|g(x)\|^2 = (\nabla f(x))^2_{(d-k+1)} + \ldots + (\nabla f(x))^2_{(d)}.$$

Therefore,
$$\langle g(x), \nabla f(x) \rangle \geq \frac{k}{d} \|\nabla f(x)\|^2,$$

and $b$ can be set to $\frac{k}{d}$, $c$ can be set to $0$.

Clearly, $\|g(x)\|^2 \leq \|\nabla f(x)\|^2$ which implies that $A = C = 0$, $B = 1$.

$\blacksquare$

## J.2   Proof of Claim 6

Observe that
$$\langle \mathbb{E}\left[g(x)\right], \nabla f(x) \rangle = \|\nabla f(x)\|^2.$$

This implies that $b = 1, c = 0$. Also, notice that
$$\mathbb{E}\left[\|g(x)\|^2\right] = \left(\frac{d}{k}\right)^2 \mathbb{E}\left[\sum_{i \in S} (\nabla f(x))^2_i \, e_i\right] = \frac{d}{k} \|\nabla f(x)\|^2.$$

Therefore, $A = C = 0$, $B = \frac{d}{k}$.

$\blacksquare$

## J.3   Proof of Claim 7

Observe that
$$\langle \mathbb{E}\left[g(x)\right], \nabla f(x) \rangle = \frac{k}{d} \|\nabla f(x)\|^2.$$

This implies that $b = \frac{k}{d}$, $c = 0$. Also, notice that
$$\mathbb{E}\left[\|g(x)\|^2\right] = \mathbb{E}\left[\sum_{i \in S} (\nabla f(x))^2_i \, e_i\right] = \frac{k}{d} \|\nabla f(x)\|^2.$$

Therefore, $A = C = 0$, $B = \frac{k}{d}$.

$\blacksquare$

## J.4   Proof of Claim 8

Lemma 6 of [Beznosikov et al., 2020] states that adaptive random sparsification operator belongs to $\mathbb{B}^1\left(\frac{1}{d}, 1\right), \mathbb{B}^2\left(\frac{1}{d}, 1\right), \mathbb{B}^3(d)$. It follows that $A = 0, B = 1, C = 0$ (see (5)) and $b = \frac{1}{d}, c = 0$.

$\blacksquare$

## J.5   Proof of Claim 9

**Definition 19** *Let $\omega \geq 1$. An estimator $g(x)$ belongs to a set $\mathbb{U}\left(\omega\right)$, if $g(x)$ is unbiased $(\mathbb{E}\left[g(x)\right] = \nabla f(x)$, for all $x \in \mathbb{R}^d)$, and if its second moment is bounded as*
$$\mathbb{E}\left[\|g(x)\|^2\right] \leq \omega \|\nabla f(x)\|^2, \quad \forall x \in \mathbb{R}^d. \tag{62}$$

Lemma 8 of [Beznosikov et al., 2020] states that general unbiased rounding operator belongs to $\mathbb{U}(\omega)$ with
$$\omega = \frac{Z}{4} = \frac{1}{4} \sup_{k \in \mathbb{Z}} \left(\frac{a_k}{a_{k+1}} + \frac{a_{k+1}}{a_k} + 2\right),$$

where $Z$ is defined in (32).

Since $g(x)$ is unbiased, we have $b = 1, c = 0$. From (62) we have that $A = C = 0$, $B = \frac{Z}{4}$.

$\blacksquare$

### J.6 Proof of Claim 10

Lemma 9 of [Beznosikov et al., 2020] states that general biased rounding operator belongs to $\mathbb{B}^1(\alpha, \beta), \mathbb{B}^2(\gamma, \beta),$ and $\mathbb{B}^3(\delta)$, where

$$\beta = F, \quad \gamma = G, \quad \alpha = \gamma^2, \quad \delta = \sup_{k \in \mathbb{Z}} \frac{(a_k + a_{k+1})^2}{4 a_k a_{k+1}}.$$

Therefore,

$$A = C = c = 0, \quad B = F^2, \quad b = \frac{G^2}{F}.$$

with $F$ and $G$ defined in (33).

∎

### J.7 Proof of Claim 11

Since natural compression estimator is a special case of general unbiased rounding estimator with $a_k = 2^k$, we obtain that $g(x)$ belongs to a set $\mathbb{U}\left(\frac{9}{8}\right)$, and, in a similar way as in the proof of Claim 9, we obtain that $A = C = c = 0, B = \frac{9}{8}, b = 1$.

∎

### J.8 Proof of Claim 12

Lemma 10 of [Beznosikov et al., 2020] states that exponential dithering operator belongs to $\mathbb{U}(H_a)$. Since $g(x)$ is unbiased, we have that $b = 1, c = 0$. From (62) we have that $A = C = 0, B = H_a$.

∎

### J.9 Proof of Claim 13

Natural dithering estimator is a special case of exponential dithering operator in case when $a = 2$. Therefore, Claim 13 is a direct consequence of Claim 12, and we have $A = C = c = 0, B = H_2, b = 1$.

∎

### J.10 Proof of Claim 14

Lemma 11 of [Beznosikov et al., 2020] states that the composition operator of Top-$k$ sparsification and exponential dithering with base $a$ belongs to $\mathbb{B}^1\left(\frac{k}{d}, H_a\right), \mathbb{B}^2\left(\frac{k}{d}, H_a\right), \mathbb{B}^3\left(\frac{d}{k}H_a\right)$, where $H_a$ is a constant defined in (34).

Therefore, from (5), we have

$$A = 0, \ B = H_a^2, \ C = 0, \ b = \frac{k}{dH_a}, \ c = 0.$$

∎

### J.11 Proof of Claim 15

When $f$ is convex and satisfies Assumption 0 with a constant $L$, Nesterov and Spokoiny [2017] (Lemma 3 and Theorem 4) bound the bias in the following way:

$$\|\mathbb{E}\left[g_{GS}(x)\right] - \nabla f(x)\|^2 \le \frac{\tau^2}{4} L^2 (d+3)^3.$$

Therefore, due to (46) and (47), we obtain that

$$\langle \nabla f(x), \mathbb{E}\left[g(x)\right] \rangle \ge \frac{1}{2} \|\nabla f(x)\|^2 - \frac{\tau^2}{8} L^2 (d+3)^3.$$

Further, from Theorem 4 of [Nesterov and Spokoiny, 2017], we have that

$$\mathbb{E}\left[\|g_{GS}(x)\|^2\right] \leq 2(d+4)\|\nabla f(x)\|^2 + \frac{\tau^2}{2}L^2(d+6)^3.$$

We can choose

$$A = A_{GS} \overset{\text{def}}{=} 0, \ B = B_{GS} \overset{\text{def}}{=} 2(d+4), \ C = C_{GS} \overset{\text{def}}{=} \frac{\tau^2}{2}L^2(d+6)^3,$$

$$b = b_{GS} = \frac{1}{2}, \ c = c_{GS} \overset{\text{def}}{=} \frac{\tau^2}{8}L^2(d+3)^3.$$

∎

## J.12 Proof of Claim 16

It is easy to see that it satisfies Assumption 7 with $\Delta = w\sqrt{d}$. Then, it follows that $\|\mathbb{E}\left[g(x)\right] - \nabla f(x)\|^2 \leq w^2 d$. Therefore, $\langle \mathbb{E}\left[g(x)\right], \nabla f(x)\rangle \geq \|\nabla f(x)\|^2 - w^2 d + \|\mathbb{E}\left[g(x)\right]\|^2 \geq \|\nabla f(x)\|^2 - w^2 d$. We can choose $b = 1$, $c = w^2 d$.

Further, $\mathbb{E}\left[\|g(x)\|^2\right] = \|g(x)\|^2 \leq \|\nabla f(x)\|^2$. It means that we can choose $A = C = 0$, $B = 1$.

∎

## J.13 Proof of Claim 17

Observe, that $g(x)$ satisfies Assumption 7 with $\Delta = \frac{\sqrt{d}}{\chi}$. Indeed, for every $j \in [d]$, we have

$$\frac{1}{n}\sum_{i=1}^n (\nabla f_i(x))_j - \frac{1}{\chi} \leq \frac{1}{n}\sum_{i=1}^n \frac{1}{\chi}\left(R\left(\chi\nabla f_i(x)\right)\right)_j \leq \frac{1}{n}\sum_{i=1}^n (\nabla f_i(x))_j + \frac{1}{\chi}.$$

Therefore, $\|g(x) - \nabla f(x)\|^2 \leq \frac{d}{\chi^2}$. In accordance with Theorem 13 - vii, we obtain that we can choose $A = 0$, $B = 2$, $C = \frac{2d}{\chi^2}$, $b = \frac{1}{2}$, $c = \frac{d}{2\chi^2}$.

∎

## J.14 Proof of Claim 18

In accordance with Khirirat et al. [2018b, Lemma 2], $g(x)$ satisfies Assumption 5 with $\rho = 1$, $\zeta = d$. It follows from Theorem 13 that $g(x)$ satisfies $BiasedABC$ with $A = 0$, $B = d$, $C = 0$, $b = 1$ and $c = 0$.

∎

## J.15 Proof of Claim 19

In accordance with Karimireddy et al. [2019, Lemma 8] $g(x)$ satisfies Assumption 3 with $\delta(x) = \frac{d\|x\|_2^2}{\|x\|_1^2} \leq d$. It follows from Theorem 13 that $g(x)$ satisfies $BiasedABC$ with $A = 0$, $B = 2\left(2 - \frac{1}{d}\right)$, $C = 0$, $b = \frac{1}{2d}$ and $c = 0$.

∎

## J.16 Proof of Claim 20

Observe that

$$\|\mathbb{E}\left[g(x)\right] - \nabla f(x)\|^2 \leq \max_j \{\lambda_j\} \cdot c^2 h^2 + \left(1 - \min_j \{\lambda_j\}\right)\|\nabla f(x)\|^2$$

$$- \left\|\text{Diag}\left(\sqrt{\lambda_1(1-\lambda_1)}, \ldots, \sqrt{\lambda_d(1-\lambda_d)}\right) \cdot G_h(x)\right\|^2.$$

Therefore, we obtain that

$$\langle \mathbb{E}\left[g(x)\right], \nabla f(x)\rangle \geq \frac{1}{2}\left(-\max_j\{\lambda_j\} \cdot c^2 h^2 + \min_j\{\lambda_j\} \cdot \|\nabla f(x)\|^2\right)$$

$$+ \frac{1}{2}\left(\|\mathbb{E}\left[g(x)\right]\|^2 + \left\|\mathrm{Diag}\left(\sqrt{\lambda_1(1-\lambda_1)}, \ldots, \sqrt{\lambda_d(1-\lambda_d)}\right) \cdot G_h(x)\right\|^2\right)$$

$$= \frac{1}{2}\left(-\max_j\{\lambda_j\} \cdot c^2 h^2 + \min_j\{\lambda_j\} \cdot \|\nabla f(x)\|^2\right)$$

$$+ \frac{1}{2}\left\|\mathrm{Diag}\left(\sqrt{\lambda_1}, \ldots, \sqrt{\lambda_d}\right) \cdot G_h(x)\right\|^2$$

$$\geq \frac{1}{2}\left(-\max_j\{\lambda_j\} \cdot c^2 h^2 + \min_j\{\lambda_j\} \cdot \|\nabla f(x)\|^2\right).$$

Further, it is easy to see that

$$\mathbb{E}\left[\|g(x)\|^2\right] \leq \max_j\{\lambda_j\} \cdot \mathbb{E}\left[\|G_h(x)\|^2\right] \leq 2\widetilde{A}\max_j\{\lambda_j\}\left(f(x) - f^*\right) + \widetilde{C}\max_j\{\lambda_j\}.$$

∎

## K   Proofs of the results presented in Table 3

We proved in Claim 3 that Biased independent sampling estimator (see Def. 1) satisfies Biased ABC assumption. On the other hand, in Theorem 2 (parts viii and ix) we show that it Assumptions 6 and 8 do not hold for it. Therefore, it does not satisfy Assumptions $1-8$ (see Figure 1).

We proved in Claim 4 that Distributed general biased rounding estimator (see Def. 2) satisfies Biased ABC assumption. As for Biased independent sampling estimator (see Def. 1), it is straightforward to show that Distributed general biased rounding estimator does not satisfy Assumptions 6 and 8. Therefore, Assumptions $1-8$ (see Figure 1) do not hold for it.

In [Beznosikov et al., 2020, Lemma 7] it is proven that Top-$k$ (see Def. 3) estimator satisfies Assumption 3. Therefore, in accordance with Figure 1, we only need to verify that Assumption 5 holds, and Assumption 7 does not hold for Top-$k$. The argument in the proof of Claim 5 shows that Assumption 5 is satisfied for $g(x)$. Consider $f(x) = \frac{x_1^2}{2} + \frac{x_2^2}{2}$, $x \in \mathbb{R}^2$, and Top-1 estimator. For every $x_2$ in $\mathbb{R}$, consider $x = (x_1, x_2) \in \mathbb{R}^2$, such that $x_1 \geq x_2$. Clearly, $g(x) = (x_1, 0)$, $\nabla f(x) = (x_1, x_2)$. Then, $\|g(x) - \nabla f(x)\|^2 = x_2^2$. For any $\Delta \geq 0$, there exists $x_2$ such that $x_2^2 \geq \Delta^2$. Therefore, Assumption 7 does not hold for $g(x)$.

Rand-$k$ (see Def. 4) is a stochastic estimator, it does not satisfy Assumption 5. Since $\|\mathbb{E}\left[g(x)\right] - \nabla f(x)\|^2 = 0$, $\mathbb{E}\left[\|g(x) - \mathbb{E}\left[g(x)\right]\|^2\right] = \left(\frac{d}{k} - 1\right)\|\nabla f(x)\|^2$, it satisfies Assumption 4. It remains to show that it does not satisfy Assumptions 3 and 7. Consider $f(x) = \frac{x_1^2}{2} + \frac{x_2^2}{2}$, $x \in \mathbb{R}^2$, and Rand-1 estimator. For every $x_2$ in $\mathbb{R}$, consider $x = (x_1, x_2) \in \mathbb{R}^2$, such that $x_1 \geq x_2$. Clearly, $\mathbb{E}\left[\|g(x) - \nabla f(x)\|^2\right] = \|\nabla f(x)\|^2 = x_1^2 + x_2^2$, and this expression can not be bounded by any constant $\Delta^2 \geq 0$, which implies that Assumption 7 does not hold. Also, there is no $\delta > 0$, such that $\|\nabla f(x)\| \leq \left(1 - \frac{1}{\delta}\right)\|\nabla f(x)\|^2$, for all $x \in \mathbb{R}^2$, which implies that Assumption 3 does not hold.

In [Beznosikov et al., 2020, Lemma 5] it is proven that Biased Rand-$k$ estimator (see Def. 5) satisfies Assumption 3. Therefore, in accordance with Figure 1, we only need to verify that Assumptions 5 and 7 do not hold for Biased Rand-$k$. Since this estimator is stochastic, Assumption 5 does not hold. Consider $f(x) = \frac{x_1^2}{2} + \frac{x_2^2}{2}$, $x = (x_1, x_2) \in \mathbb{R}^2$, and Biased Rand-1 estimator. We have that $\mathbb{E}\left[\|g(x) - \nabla f(x)\|^2\right] = \frac{1}{2}\|\nabla f(x)\|^2 = \frac{x_1^2}{2} + \frac{x_2^2}{2}$, and this expressioin can not be bounded by any constant $\Delta^2 \geq 0$. Therefore, Assumption 7 is not satisfied.

In [Beznosikov et al., 2020, Lemma 6] it is proven that Adaptive random sparsification (see Def. 6) satisfies Assumption 3. Therefore, in accordance with Figure 1, we only need to verify that Assumptions 5 and 7 do not hold for Adaptive random sparsification estimator. Since it is stochastic,

Assumption 5 does not hold. Consider $f(x) = \frac{x_1^2}{2} + \frac{x_2^2}{2}$, $x = (x_1, x_2) \in \mathbb{R}^2$, and Adaptive random sparsification estimator. Observe that

$$\mathbb{E}\left[\|g(x) - \nabla f(x)\|^2\right] = \|\nabla f(x)\|_2^2 \left(1 - \frac{\|\nabla f(x)\|_3^3}{\|\nabla f(x)\|_1 \|\nabla f(x)\|_2^2}\right)$$

$$= \left(x_1^2 + x_2^2\right)\left(1 - \frac{x_1^3 + x_2^3}{(|x_1| + |x_2|)\left(x_1^2 + x_2^2\right)}\right).$$

Let $\lambda > 0$ be some constant. Consider $x \in \mathbb{R}^2$ such that $|x_1| = \lambda|x_2|$. Then

$$\mathbb{E}\left[\|g(x) - \nabla f(x)\|^2\right] = \lambda x_2^2,$$

and, for any $\Delta^2 \geq 0$, there exists $x_2 \in \mathbb{R}$, such that $\lambda x_2^2 \geq \Delta^2$. Therefore, Assumption 7 does not hold.

General unbiased rounding (see Def. 7) belongs to $\mathbb{U}\left(\frac{Z}{4}\right)$ (see Claim 9) with $Z$ defined in (32). Then, $\mathbb{E}\left[\|g(x) - \nabla f(x)\|^2\right] \leq \left(\frac{Z}{4} - 1\right)\|\nabla f(x)\|^2$, and $g(x)$ satisfies Assumption 4. Therefore, in accordance with Figure 1, we only need to verify that Assumptions 3, 5 and 7 do not hold. Let $a_k = 6^k$, $k \in \mathbb{Z}$. Consider $f(x) = \frac{x^2}{2}$, $x = \in \mathbb{R}$, and General unbiased rounding estimator. Then,

$$\mathbb{E}\left[\|g(x) - \nabla f(x)\|^2\right] = \mathbb{E}\left[\|g(x)\|^2\right] - \|\nabla f(x)\|^2$$

$$= -6^{2k+1} + 7 \cdot 6^k \cdot x - x^2.$$

Let $x = \frac{7}{4} \cdot 6^k$. Then $\mathbb{E}\left[\|g(x) - \nabla f(x)\|^2\right] \geq x^2$ :

$$\frac{\mathbb{E}\left[\|g(x) - \nabla f(x)\|^2\right]}{x^2} = \frac{51}{49} > 1.$$

Then, Assumption 3 does not hold. Note that, for every constant $\Delta^2 \geq 0$, there exists $k \in \mathbb{Z}$, such that $x^2 > \Delta^2$. Therefore, Assumption 7 is not satisfied. Since this estimator is stochastic, Assumption 5 does not hold as well.

In [Beznosikov et al., 2020, Lemma 9] it is proven that General biased rounding estimator (see Def. 8) satisfies Assumption 3. Therefore, it remains to prove that Assumption 5 holds and Assumption 7 does not hold. It follows from the proof of Claim 10 that General biased rounding estimator satisfies Assumption 5. Observe that it is deterministic, which implies that

$$\mathbb{E}\left[\|g(x) - \nabla f(x)\|^2\right] = \|g(x) - \nabla f(x)\|^2.$$

Let $f(x) = \frac{x^2}{2}$, $a_k = 2^{2k+1}$. For a sequence of iterations $\{x_k\} = 2^{2k} + 0.1$, $k \in \mathbb{N}$, we have $\|g(x) - \nabla f(x)\|^2 = 2^{2k} - 0.1$, and there is no constant $\Delta^2 \geq 0$, such that $2^{2k} - 0.1 \leq \Delta^2$, for every $k \in \mathbb{N}$.

Natural compression (see Def. 9) belongs to $\mathbb{U}\left(\frac{9}{8}\right)$ (see Claim 9). Then, $\mathbb{E}\left[\|g(x) - \nabla f(x)\|^2\right] \leq \frac{1}{8}\|\nabla f(x)\|^2$, and $g(x)$ satisfies Assumption 3. Therefore, in accordance with Figure 1, we only need to verify that Assumptions 5 and 7 do not hold. Since $g(x)$ is a stochastic estimator, Assumption 5 is not satisfied. Consider $f(x) = \frac{x^2}{2}$, $x \in \mathbb{R}$, and Natural compression estimator. Then

$$\mathbb{E}\left[\|g(x) - \nabla f(x)\|^2\right] = \mathbb{E}\left[\|g(x)\|^2\right] - \|\nabla f(x)\|^2$$

$$= -2^{2k+1} + 3 \cdot 2^k \cdot x - x^2.$$

Let $x = \frac{3}{2} \cdot 2^k$. Then $\mathbb{E}\left[\|g(x) - \nabla f(x)\|^2\right] = 2^{2k-2}$. For every constant $\Delta^2 \geq 0$, there exists $k \in \mathbb{Z}$, such that $2^{2k-2} > \Delta^2$. Therefore, Assumption 7 does not hold.

It is shown in [Beznosikov et al., 2020, Lemma 11] that General exponential dithering estimator (see Def. 10) satisfies Assumption 3. Therefore, it remains to show that Assumptions 5 and 7 do

not hold. Since this estimator is stochastic, Assumptions 5 is not satisfied. Further, let $f(x) = \frac{x_1^2 + x_2^2}{2}$, $p = 2$, $x_1 = 3k$, $x_2 = 4k$, $k \in \mathbb{Z}$, $s = 1$, $a = 2$. Then, $\mathbb{E}\left[g(x)\right] = (2k, k)$. Therefore, $\mathbb{E}\left[\|g(x) - \nabla f(x)\|^2\right] = 15k^2 - 5k^2 + 10k^2 = 20k^2$, and there is no $\delta > 0$, such that $20k^2 \leq \left(1 - \frac{1}{\delta}\right)15k^2$.

Since Natural dithering (see Def. 11) is a special case of General exponential dithering, and we established all of the inclusions for it regardless of the value of $a$, the same Assumptions hold or do not hold for this estimator.

It is proven in [Beznosikov et al., 2020, Lemma 11] that Composition of Top-$k$ and exponential dithering satisfies Assumption 3. Since this estimator is stochastic, it does not satisfy Assumption 5. Suppose $f(x) = \frac{x^2}{2}$, $d = 1$, and the estimator is composed of Top-1 and exponential dithering with base $a = 2$ and $s = 1$. Since $d = 1$, $k = 1$, the problem and the estimator is exactly the same as in the previous case, where we showed that General exponential dithering estimator does not satisfy Assumption 7.

Let us show that Gaussian smoothing (see Def. 13) does not satisfy Assumptions 8. Suppose $f(x) = x$, $x \in \mathbb{R}$, $d = 1$. Then $g_{GS}(x) = z^2$, $z \sim \mathcal{N}(0, 1)$, and we have $\langle \mathbb{E}\left[g_{GS}(x)\right], \nabla f(x)\rangle = 1$, which means that Assumption 8 does not hold. Let us show that Assumption 7 is not satisfied as well. Consider $f(x) = \frac{x^2}{2}$, $x \in \mathbb{R}$, $d = 1$. Then, $g_{GS}(x) = \frac{2xz^2 + \tau z^3}{2}$. Observe that $\mathbb{E}\left[g_{GS}(x)\right] = x$. We have that

$$\mathbb{E}\left[\|g_{GS}(x) - \nabla f(x)\|^2\right] = \mathbb{E}\left[\|g_{GS}(x)\|^2\right] - x^2$$
$$= \mathbb{E}\left[x^2 z^4 + \frac{\tau^2 z^6}{4} + x\tau z^5\right] - x^2$$
$$= 2x^2 + \frac{15}{4}\tau^2,$$

and it can not be bounded by $\Delta^2 \geq 0$.

In Claim 16 we show that hard-threshold sparsifier (see Def. 14) satisfies Assumption 7. It is easy to see that it satisfies Assumption 4. If we consider $f(x) = \frac{x^2}{2}$, and, for all $t \in \mathbb{N}$, $|x_t| < \omega$, then $\mathbb{E}\left[\|g(x) - \nabla f(x)\|^2\right] = \|\nabla f(x)\|^2$, and Assumption 3 is not satisfied. Further, let us show that this estimator does not satisfy Assumption 5. Consider $f(x) = \frac{x^2}{2}$, and $x \in \mathbb{R}^d$, $d > 1$, such that, for $\ell = [d/2]$, $x_1 = \ldots = x_\ell = \frac{\omega}{2}$, and $x_{\ell+1} = \ldots = x_d = \omega$. Then,

$$\langle \mathbb{E}\left[g(x)\right], \nabla f(x)\rangle = \|x\|^2 - \omega(d - [d/2] + 1),$$

and Assumption 5 does not hold.

In Claim 17 we prove that Scaled integer rounding (see Def. 15) satisfies Assumption 7. Also, it is easy to see that $\|g(x) - \nabla f(x)\|^2 \leq \|\nabla f(x)\|^2$, and equality holds for $f(x) = \frac{x^2}{2}$, $n = d = 1$, $x = 0.25$. Therefore, $g(x)$ does not satisfy Assumption 3, and satisfies Assumption 4. Since rounding preserves the sign (or rounds a number to 0), we have that $\langle \nabla f(x), \frac{1}{\chi}R\left(\chi\left(\nabla f(x)\right)_i\right)\rangle \geq 0$. Also, $\|g(x)\|^2 \leq 4\|\nabla f(x)\|^2$. This means, $g(x)$ satisfies Assumption 5. There is a misprint in Table 3, refer to Table 9.

It is proven in Claim 18 that Biased dithering estimator (see Def. 16) satisfies Assumption 5. Further, let $f(x) = \frac{x_1^2}{2} + \ldots + \frac{x_9^2}{2}$, $d = 9$, $k \in \mathbb{N}$, $x_1 = \ldots = x_9 = k$. Then we have

$$\mathbb{E}\left[\|g(x) - \nabla f(x)\|^2\right] = \|\mathbb{E}\left[g(x)\right] - \nabla f(x)\|^2$$
$$= \|(\text{sign}(x_1)\left(|x_1| - \|\nabla f(x)\|\right), \ldots, \text{sign}(x_9)\left(|x_9| - \|\nabla f(x)\|\right))\|^2$$
$$= \|\nabla f(x)\|^2 \left(\sqrt{9} - 1\right)^2$$
$$= 4\|\nabla f(x)\|^2,$$

which means that Assumption 6 is not satisfied.

In Claim 19 we prove that Sign compression estimator (see Def. 17) satisfies Assumption 3. Therefore, in accordance with Figure 1, we only need to deal with Assumptions 5 and 7. Since this estimator is deterministic, it follows from Claim 19 that Assumption 5 holds for it. Suppose $f(x) = \frac{x_1^2}{2} + \frac{x_2^2}{2}$, $d = 2$. Then

$$
\begin{aligned}
\mathbb{E}\left[\|g(x) - \nabla f(x)\|^2\right] &= \|\mathbb{E}\left[g(x)\right] - \nabla f(x)\|^2 \\
&= \left\|\left(\text{sign}(x_1)\frac{|x_1|}{2}, \text{sign}(x_2)\frac{|x_2|}{2}\right) - (x_1, x_2)\right\|^2 \\
&= \frac{x_1^2}{4} + \frac{x_2^2}{4} \\
&= \frac{1}{4}\|\nabla f(x)\|^2.
\end{aligned}
$$

It follows that Assumption 7 does not hold.

## L  Relation between assumption 3 and contractive compression

In Assumption 3, one can observe a resemblance to the contractive compression property, as shown in the following equation:

$$
\mathbb{E}\left[\|\mathcal{C}(x) - x\|^2\right] \leq \left(1 - \frac{1}{\delta}\right)\|x\|^2 \quad \forall x \in \mathbb{R}^d. \tag{63}
$$

The contractive compression property is commonly utilized in methods dealing with biased compression (e.g., TopK), as demonstrated in various studies [Stich et al., 2018; Karimireddy et al., 2019; Stich and Karimireddy, 2020; Beznosikov et al., 2020; Gorbunov et al., 2020; Cordonnier, 2018; Richtárik et al., 2021; Fatkhullin et al., 2021; Richtárik et al., 2022]. However, equations (6) and (63) are not generally equivalent since in practise one may not aim to compress exactly a gradient itself.

## M  Relation between Assumption 7 and absolute compression

Within Assumption 7, a similarity to the absolute compression property

$$
\mathbb{E}\left[\|\mathcal{C}(x) - x\|^2\right] \leq \Delta^2 \quad \forall x \in \mathbb{R}^d \tag{64}
$$

can be discerned. Nonetheless, it should be noted that the expressions in equations (14) and (64) do not typically exhibit equivalence.

Various instances of absolute compression have been extensively employed by practitioners over the years [Tang et al., 2020; Sahu et al., 2021; Danilova and Gorbunov, 2022]. A prominent example is the hard-threshold sparsifier $\mathcal{C}_{\text{HT}}(x)$ [Sahu et al., 2021; Dutta et al., 2020; Ström, 2015]. It can be demonstrated that $\mathcal{C}_{\text{HT}}(x)$ adheres to Eq. (14) with $\Delta = \lambda\sqrt{d}$. Additional examples encompass (stochastic) rounding schemes with limited error [Gupta et al., 2015; Khirirat et al., 2020] and integer rounding [Sapio et al., 2021; Mishchenko et al., 2021].

The absolute compression assumption has also been featured in several studies [Sahu et al., 2021; Danilova and Gorbunov, 2022; Khirirat et al., 2020, 2022; Chen et al., 2021a], which examine the Error Feedback mechanism [Stich et al., 2018; Karimireddy et al., 2019; Stich and Karimireddy, 2020].

Specifically, Sahu et al. [2021] established that hard-threshold sparsifiers are optimal for minimizing total error (a unique quantity that emerges in the analysis of EC-SGD) with respect to any fixed sequence of errors.

Furthermore, the authors of [Sahu et al., 2021] elucidate both the theoretical and practical advantages of absolute compressors in comparison to $\delta$-contractive ones expressed in Equation (63).

## N  Relations between the estimators from Assumptions 1–3

Below we restate Theorem 2 from [Beznosikov et al., 2020] about the relations between these sets in terms of biased gradient estimators instead of biased compressors.

**Theorem 14 (Relations between the estimators from Assumptions 1–3)** *Let $\lambda > 0$ be a scaling parameter.*

1. *If $g \in \mathbb{B}^1(\alpha, \beta)$, then*

   - $\beta^2 \geq \alpha$ *and* $\lambda g \in \mathbb{B}^1\left(\lambda^2 \alpha, \lambda \beta\right)$,
   - $g \in \mathbb{B}^2\left(\alpha, \beta^2\right)$ *and* $\frac{1}{\beta} g \in \mathbb{B}^3\left(\frac{\beta^2}{\alpha}\right)$.

2. *If $g \in \mathbb{B}^2\left(\tau, \beta\right)$, then*

   - $\beta \geq \tau$ *and* $\lambda g \in \mathbb{B}^2\left(\lambda \tau, \lambda \beta\right)$,
   - $g \in \mathbb{B}^1\left(\tau^2, \beta\right)$ *and* $\frac{1}{\beta} g \in \mathbb{B}^3\left(\frac{\beta}{\tau}\right)$

3. *If $g \in \mathbb{B}^3\left(\delta\right)$, then*

   - $\delta \geq 1$,
   - $g \in \mathbb{B}^2\left(\frac{1}{2\delta}, 2\right) \subseteq \mathbb{B}^1\left(\frac{1}{4\delta^2}, 2\right)$.

We do not prove it here and refer the reader to the original paper.

## O   Equivalence of Assumption 6 and [Ajalloeian and Stich, 2020, Def. 1]

Definition 1 in [Ajalloeian and Stich, 2020] is written in the following way.

**Definition 20** *Let $(\mathcal{D}, \mathcal{F})$ be a measurable space and $Y$ be a random element of this space. Let gradient estimator $g(x, Y)$ have a form*

$$g(x, Y) = \nabla f(x) + b(x) + \mathcal{Z}(x, Y),$$

*where $b(x) : \mathbb{R}^d \to \mathbb{R}^d$ is a bias and $\mathcal{N} : \mathbb{R}^d \times \mathcal{D} \to \mathbb{R}^d$ is a zero-mean noise, i.e. $\mathbb{E}\left[\mathcal{Z}(x, Y) | Y\right] = 0$, for all $x \in \mathbb{R}^d$.*

*There exist constants $M, \sigma^2 \geq 0$ such that*

$$\mathbb{E}\left[\|\mathcal{Z}(x, Y)\|^2\right] \leq M \|\nabla f(x) + b(x)\|^2 + \sigma^2, \quad \forall x \in \mathbb{R}^d. \tag{65}$$

*There exist constants $0 \leq m < 1$ and $\varphi^2 \geq 0$, such that*

$$\|b(x)\|^2 \leq m \|\nabla f(x)\|^2 + \varphi^2, \quad \forall x \in \mathbb{R}^d. \tag{66}$$

For the purpose of clarity, we rewrote the inequalities (65) and (66) in the notation adopted in our paper (see Section 4). Below we establish their equivalence.

**Claim 21** *Definition 20 is equivalent to Assumption 6.*

**Proof of Claim 21.** Observe that $\mathcal{Z}(x, Y) = g(x, Y) - \mathbb{E}\left[g(x, Y)\right]$, $\nabla f(x) + b(x) = \mathbb{E}\left[g(x, Y)\right]$, $b(x) = \mathbb{E}\left[g(x, Y)\right] - \nabla f(x)$. It remains to perform these substitutions in (65) and (66).

## P   Proof of Lemma 1

Let $x_+ = x - \frac{1}{L}\nabla f(x)$, then using the $L$-smoothness of $f$ we obtain

$$f(x_+) \leq f(x) + \langle \nabla f(x), x_+ - x \rangle + \frac{L}{2} \|x_+ - x\|^2.$$

Since $f^* \leq f(x_+)$ and the definition of $x_+$ we have,

$$f^* \leq f(x_+) \leq f(x) - \frac{1}{L} \|\nabla f(x)\|^2 + \frac{1}{2L} \|\nabla f(x)\|^2 = f(x) - \frac{1}{2L} \|\nabla f(x)\|^2.$$

It remains to rearrange the terms to get the claimed result.

∎

## Q   Young's inequality

Throughout the paper we use the following version of a well-known inequality:

**Lemma 3 (Young's Inequality)**  *For every $s > 0$, for any vectors $u, h \in \mathbb{R}^d$, we have*

$$||u \pm h||^2 \leq (1 + s)\,||u||^2 + \left(1 + \frac{1}{s}\right)||h||^2. \tag{67}$$

*Or, equivalent,*

$$\pm 2\langle u, h \rangle \leq s||u||^2 + \frac{1}{s}||h||^2. \tag{68}$$

**Proof of Lemma 3.** Let $u' = \sqrt{s}u$, $h' = \frac{h}{\sqrt{s}}$. Then (68) can be rewritten as

$$\pm 2\langle u', h' \rangle \leq ||u'||^2 + ||h'||^2.$$

Or, equivalent, $\|u' \pm h'\|^2 \geq 0$.

∎