# OpenReview forum: "A Guide Through the Zoo of Biased SGD"
_NeurIPS.cc/2023/Conference — NeurIPS 2023 poster_

### Official Review · Reviewer_R984 · 2023-06-16

**Soundness:** 4 excellent
**Presentation:** 4 excellent
**Contribution:** 3 good
**Rating:** 6
**Confidence:** 4

**Summary:**

The paper analyzes the assumptions for biasedSGD for various applications and proposes a more general and weaker ABC-assumption that covers the mentioned assumptions. The paper provides a detailed comparison with several existing literature trying to build up a unified framework for biasedSGD. It seems to be a further generalization of [Ahmad Ajalloeian and Sebastian U Stich, 2020].

**Strengths:**

The paper provides a clear link of different assumption used in the BiasedSGD literature and compare them in Figure 1. In additional, it proposes a set of assumption for biasedSGD that can guarantee the same iteration complexity as the unbiased SGD. I appreciate the clarity and structure of these relationships as they can be invaluable when addressing a new problem requiring BiasedSGD, thus saving time for the reader.

**Weaknesses:**

I mainly have two concerns: missing reference on other biasedSGD methods and how the proposed ABC assumption can serve as a useful assumption.

This paper overlooks a substantial body of literature on biasedSGD in contexts beyond classical stochastic optimization/finite-sum and distributed training settings, such as
1. Stochastic compositional/minimax/bilevel optimization. [1,2,3]
2. Conditional stochastic optimization. [4,5,6]
3. Distributionally robust optimization. [7,8]
4. Meta learning. [9]

These works are centered on constructing biased gradient estimators and analyzing biasedSGD. Unlike the manuscript, these studies not only consider iteration complexity but also the total sample size needed to reach the desired solution, given the potential high sampling/computational cost of creating low-bias or unbiased gradient estimators. This angle is neglected in the manuscript, causing the title "A Guide Through the Zoo of Biased SGD" to seem overstated. Consequently, Section 2, addressing the source of bias, appears narrow given the wide applicability of biasedSGD.

I suggest that the authors limit their claims, incorporate and discuss the above references, and address other bias sources. Specifically, it would be insightful if they could compare two biasedSGD construction methodologies:
1. The approach in [1] that progressively reduces the bias level of the biased gradient estimators as the algorithm evolves.
2. The approach in [5] that constructs low-bias gradient estimators at a cheap cost at each iteration.
Notably, it seems that [1,2,3] don't meet the proposed ABC condition, but the proposed biasedSGD algorithm can still achieve optimal sample complexity. For [5], the gradient estimator constructed in [5] should satisfy ABC condition or its variants but the submitted paper fails to address the potential high costs of obtaining gradient estimators that adhere to the proposed ABC assumption. The authors should discuss these limitations.

The proposed ABC assumption appears difficult to validate. It remains unclear when this condition should be employed.



[1] Chen, Tianyi, Yuejiao Sun, and Wotao Yin. "Closing the gap: Tighter analysis of alternating stochastic gradient methods for bilevel problems." Advances in Neural Information Processing Systems 34 (2021): 25294-25307.

[2] Chen, Tianyi, Yuejiao Sun, and Wotao Yin. "Solving stochastic compositional optimization is nearly as easy as solving stochastic optimization." IEEE Transactions on Signal Processing 69 (2021): 4937-4948.

[3] Chen, Tianyi, Yuejiao Sun, and Wotao Yin. "A single-timescale stochastic bilevel optimization method." arXiv preprint arXiv:2102.04671 (2021).

[4] Hu, Yifan, et al. "Biased stochastic first-order methods for conditional stochastic optimization and applications in meta learning." Advances in Neural Information Processing Systems 33 (2020): 2759-2770.

[5] Hu, Yifan, Xin Chen, and Niao He. "On the bias-variance-cost tradeoff of stochastic optimization." Advances in Neural Information Processing Systems 34 (2021): 22119-22131.

[6] Hu, Yifan, Xin Chen, and Niao He. "Sample complexity of sample average approximation for conditional stochastic optimization." SIAM Journal on Optimization 30.3 (2020): 2103-2133.

[7] Levy, Daniel, et al. "Large-scale methods for distributionally robust optimization." Advances in Neural Information Processing Systems 33 (2020): 8847-8860.

[8] Wang, Jie, Rui Gao, and Yao Xie. "Sinkhorn distributionally robust optimization." arXiv preprint arXiv:2109.11926 (2021).

[9] Ji, Kaiyi, Junjie Yang, and Yingbin Liang. "Theoretical convergence of multi-step model-agnostic meta-learning." Journal of machine learning research (2022).

**Questions:**

See weakness. Other questions:

1. Ahmad Ajalloeian and Sebastian U Stich 2020 has been published. Please update the preprint references if they are published.
2. Table 4, it says that "In most cases, we ensure the same rate", but the table 4 last column does not seem to suggest so.
3. For strongly convex case, what happens when Assumption 12 fail?

**Limitations:**

The main drawback of the work is its predominant focus on iteration complexity in specific contexts, such as finite-sum, zeroth-order, and distributed stochastic optimization settings. Other perspectives on BiasedSGD, such as sample costs/computational costs for different problem settings, are not sufficiently addressed. Numerous biasedSGD methods applicable to a variety of other contexts are overlooked.

---

> ### Author Rebuttal · Authors · 2023-08-07
>
> >**Strengths:**
>
> **Answer:**
> We genuinely appreciate the time and effort you dedicated to our work. Your comments were incredibly useful. In addition, we thank the reviewer for recognizing the fundamental concepts within our paper and highlighting the presentation of our results. In the sections that follow, we assiduously address all concerns raised by the reviewer.
>
> >**Weaknesses:**
>
> **Answer:**
>
> Thank you for the feedback and for pointing out that biased gradient estimators are widely used outside classical stochastic optimization/finite-sum and distributed training settings!
>
> >The approach in [5] that constructs low-bias gradient estimators at a cheap cost at each iteration. Notably, it seems that [1,2,3] don't meet the proposed ABC condition, but the proposed biasedSGD algorithm can still achieve optimal sample complexity. For [5], the gradient estimator constructed in [5] should satisfy ABC condition or its variants but the submitted paper fails to address the potential high costs of obtaining gradient estimators that adhere to the proposed ABC assumption. The authors should discuss these limitations.
>
>
>
> We are grateful to the reviewers for providing a collection of pertinent and valuable references. These will be duly cited in the revised version of our paper.
>
>
>
> It's observed that the results from papers [1-6] don’t appear to align with our proposed ABC assumption. However, this discrepancy doesn’t diminish the value of our contribution. The ABC assumption was formulated for a different problem setup compared to that in papers [1-4].
>
>
>
> Specifically, papers [1,3] explore the bilevel optimization problem as shown:
>
>
>
> $$
> \min_{x \in \mathbb{R}^d} F(x):=\mathbb{E}_{\xi}\left[f\left(x, y^*(x) ; \xi\right)\right]
> $$
>
> $$
> \text { s.t. } y^*(x)=\underset{y \in \mathbb{R}^{d^{\prime}}}{\arg \min } \mathbb{E}_\phi[g(x, y ; \phi)]
> $$
>
>
>
> This problem differs from ours, which exclusively addresses the unconstrained minimization problem in the form $\min _{x \in \mathbb{R}^d} f(x)$ without any explicit bilevel structure. While, under a particular choice of function $f(x,y)$, the bilevel problem can imply our non-bilevel problem, this is a special case with limited relevance.
>
>
>
> Paper [2], on the other hand, delves into a nested problem given by:
>
>
>
> $$
> \min_{\theta \in \mathbb{R}^d} F(\theta):=f_N\left(f_{N-1}\left(\cdots f_1(\theta) \cdots\right)\right)
> $$
>
> with $f_n(\cdot):=\mathbb{E}_{\xi_n}\left[f_n\left(\cdot ; \xi_n\right)\right], \quad n=1,2, \ldots, N$
>
>
>
> This is also outside the purview of our paper.
>
>
>
> Paper [6] tackles another unique problem, termed the conditional stochastic optimization problem:
>
>
>
> $$
> \min_{x \in \mathcal{X}} F(x):= \mathbb{E}\_{\xi} f_\{\xi} \left(\mathbb{E}\_{\eta \mid \xi} g\_{\eta(x, \xi)}\right)
> $$
>
> over a convex set $\mathcal{X} \subseteq \mathbb{R}^d$
>
>
>
> In contrast, the estimation method in paper [5], although orthogonal to our approach, doesn't diverge entirely since it adopts a distinct mathematical framework, specifically the MLMC.
>
>
>
> >The proposed ABC assumption appears difficult to validate. It remains unclear when this condition should be employed.
>
>  Regarding the proposed ABC assumption, we acknowledge reviewers' concerns about its applicability and will address this by directing their attention to Tables 2 and 8, illustrating specific valid scenarios.
>
> Even within classical contexts, numerous scholarly works exist. However, our research exposes a substantial gap, evident in the inadequacy of proposed assumptions in capturing common sources of bias, notably pronounced in intricate sampling schemes, a cornerstone of Federated Learning. Our work extends prior findings to encompass these settings.
>
> Contrary to concerns about validation difficulty, our Assumption 9 demands fewer verifications (two) than the classical Assumption 8 (three) by L. Bottou et al. (2018). Most Machine Learning cases have nonnegative loss functions, allowing $f^{*}$ to be set to zero. Our framework embraces instances like Huber Loss, a 1-smooth convex function, and specific sampling estimators, complying with Assumption 9 while diverging from weaker Assumptions 6 and 8. This underscores Assumption 9's effectiveness in specific contexts where established assumptions falter, especially for estimators influenced by sampling bias.
>
> >**Questions:**
>
> **Answer:**
>
> Thank you for your questions.
>
> Regarding the references, we value your attention to detail. While we will cross-reference the preprint sources, we are struggling to locate the 2020 publication by Ahmad Ajalloeian and Sebastian U Stich. Could you please share the source link for this publication? Your help will ensure accurate reference updating.
>
> In Table 4, we establish a consistent rate of change based on factors like $\kappa=L/\mu,$ $1/\varepsilon,$ $L,$ and $\delta_0$. Notably, other influencing factors can be encompassed within the O-big notation. Nonetheless, we include them explicitly to emphasize that broader applicability comes with increased complexity.
>
> If Assumption 12 proves invalid, we lack an existing analysis guaranteeing iteration-based convergence under our proposed general Assumption 9. Yet, we do guarantee function value convergence, detailed in Theorem 12 of Section G2. This assurance directly stems from Theorem 4, as Assumption 10 (Polyak–Łojasiewicz) covers a broader context than Assumption 11 (strong convexity). Consequently, Theorem 12 resides in the appendix, while Section 6.3's beginning explicitly states the applicability of Theorem 4 to Assumption 11-compliant functions.
>
> Please feel free to seek further clarification or share the source link for the aforementioned publication.
>
> We believe that our responses have effectively addressed your concerns. We respectfully ask you to reconsider your initial evaluation. We are optimistic that our explanations will provide greater satisfaction, and the additional information we have shared could potentially warrant a higher rating.

---

> > ### Comment · Reviewer_R984 · 2023-08-10
> > **Thank you.**
> >
> > Thanks for the looking into the reference in such a short time and for the response. It addresses my concerns. I am happy to raise the score to 6.
> >
> > Regarding the question, I mean [Ahmad Ajalloeian and Sebastian U Stich. Analysis of SGD with biased gradient estimators. arXiv preprint arXiv:2008.00051, 2020.] has been published at ICML workshop 2020 with name "On the Convergence of SGD with Biased Gradients".

---

> > > ### Author Response · Authors · 2023-08-14
> > > **Reply**
> > >
> > > Dear Reviewer R984,
> > >
> > > Thank you for your reply and increasing the score!
> > >
> > > We would be delighted to engage in a discussion regarding certain facets of the work and provide further elucidation.
> > >
> > > In reference to the ICML workshop, we regret to inform you that citing a workshop paper is not feasible, as this particular workshop does not possess official proceedings.

---

### Official Review · Reviewer_84Sm · 2023-06-29

**Soundness:** 3 good
**Presentation:** 3 good
**Contribution:** 4 excellent
**Rating:** 6
**Confidence:** 3

**Summary:**

This paper establishes connections among the existing assumptions found in the literature on Stochastic Gradient Descent (SGD) with biased estimators. The authors provide a comprehensive map that illustrates the relationships between these assumptions, addressing the lack of coherence found in previous works. Furthermore, they introduce a new set of assumptions, namely Biased ABC that have been proven to be weaker than the previous ones. It enables a thorough analysis of Biased SGD in both convex and non-convex settings. The paper also showcases examples where biased estimators outperform their unbiased counterparts or where unbiased versions are not available. Finally, the authors validate their theoretical findings through experimental results, which further supports the effectiveness of their framework.

**Strengths:**

This paper provides a comprehensive overview of the existing literature elaborating on connections between the existing assumptions and the proposed assumption used in the analysis for biased SGD.

Often, when comparing two theoretical results, one may demonstrate slightly tighter result by relying on stricter assumptions, while the other may provide a slightly worse dependence but arguing they have milder assumption. Assessing which approach is superior can be a challenging task. In this regard, the authors have tackled a crucial and significant subject. They have made a good effort towards unifying the various assumptions by proposing a new set of assumption which believed to be weaker than all previous ones. These proposed assumptions cover the existing assumptions, and the convergence results can be suitably adapted accordingly.



**Weaknesses:**

The newly proposed assumption Biased ABC is an incremental work based on existing work on biased SGD and the work done by Khaled and Richtárik [2023] for unbiased SGD. I felt there is not much novelty exists in this newly proposed assumption.

Also, in my opinion,  the balance between the discussion on the background and introduction of existing assumptions and the presentation of the authors' own theoretical results could be further improved.



**Questions:**

1. Is there an example to support Claim 1?


**Limitations:**

1. Figure 1 summarized one of the main contribution of this paper. However, the way it presented right now is not ideal. Especially after including the blue dashed lines. With so many lines that are flying around, the main message that this figure tries to deliver is not well received, in my opinion. I would recommend the author find a more concise way to present the connections in Figure 1.

---

> ### Author Rebuttal · Authors · 2023-08-07
>
> >**Strengths:**
>
> **Answer:**
>
> We sincerely appreciate the time and effort you dedicated to offering your valuable feedback and insights on our work. Your comments have proven immensely helpful. Our gratitude extends to the reviewer for identifying the foundational concepts within our paper. In the sections that follow, we diligently address all concerns raised by the reviewer.
>
> >**Weaknesses:**
>
> **Answer:**
>
> Initially, we independently formulated the scalar product condition outlined in Assumption 9, unaware of its prior introduction by L. Bottou. The absence of explicit mention of the consideration of the biased version of SGD led to our independent derivation. While we recognized that Khaled and Richtarik presented a highly generalized assumption in the unbiased context, we extended it to encompass the biased scenario in the most comprehensive manner we were cognizant of. It's worth noting that Table 4 underscores our analysis's equivalence, highlighting our work's broader scope.
>
> Acknowledging the wealth of ideas already in existence, we concur that our work is founded on established concepts. Yet, the multitude of publications on the subject suggests that the amalgamation of these ideas into Assumption 9 is no straightforward feat.
>
> Indeed, our current paper may not extensively delve into existing assumptions or optimally present our findings. However, should our submission be accepted, we are hopeful that an additional page allocation for the main content will offer us the opportunity to comprehensively discuss prior results, provide a robust motivation for the introduction of our assumption, and succinctly summarize our contributions.
>
> >**Questions:**
>
> **Answer:**
>
> The example substantiating Claim 1 is located on page 32 within the section dedicated to proving the implications depicted in Figure 1. We appreciate your observation regarding its accessibility, and we are considering subdividing the relevant section into subsections, with a distinct subsection dedicated to the proof of Claim 1. This subdivision should enhance the clarity and facilitate easier navigation to the pertinent content.
>
> Note that we corrected the proof of Theorem 2 viii and ix. You can check the updated proof in the general response above.
>
> >**Limitations:**
>
> **Answer:**
>
> We sincerely appreciate your feedback regarding the diagram. Our work delves into a multitude of assumptions, and while the visualization of connections is challenging, it remains an essential aspect. The diagrammatic representation offers a concise and convenient presentation compared to a table format. Its effectiveness lies in not needing to illustrate every single relationship due to the transitivity principle. For instance, no lines are drawn between ABS and SG1, as ABS doesn't imply FSML (which encompasses SG1), and conversely, BREQ (a subset of SG1) doesn't imply BND (a subset of ABS).
>
> This diagram serves to highlight why Theorems 1 and 2 solely focus on the connections indicated by the blue lines and black arrows. The consideration of removing the blue dashed lines is an option under contemplation, as it would potentially necessitate supplementary explanatory text placed outside the diagram, which might not be as reader-friendly. We will weigh this possibility. Importantly, the current format enables readers to easily locate desired connections and corresponding theorem sections with a simple click.
>
> We are open to exploring the possibility of removing the blue dashed lines from the diagram and replacing them with explanatory footnotes. As we evaluate this approach, it remains uncertain how effectively the new diagram will convey information. Therefore, our intention is to experiment with different alternatives to ascertain the most illustrative and reader-friendly presentation. Your suggestion is valued, and we are committed to refining the diagram in a manner that best enhances clarity and understanding.
>
> Thank you once again for your valuable input, which we highly regard in our continuous efforts to enhance the clarity and accessibility of our work.
>
> We hold a strong belief that our comments have effectively addressed your concerns. With utmost respect, we kindly request you to consider revisiting your initial evaluation. We remain hopeful that our responses will prove more satisfactory, and the additional explanations we've provided could potentially justify an elevated score. Your acknowledgment of our response would be greatly appreciated, particularly if you deem further discussion to be necessary.

---

> > ### Comment · Reviewer_84Sm · 2023-08-14
> >
> > I would like to thank the authors for their rebuttal. After carefully considering the rebuttal and the other reviews, I have chosen to keep my score.

---

> > > ### Author Response · Authors · 2023-08-14
> > > **Reply**
> > >
> > > Dear Reviewer,
> > >
> > > Thank you for your reply! We are pleased to engage in a discussion regarding certain aspects of the work and provide clarification where needed.

---

### Official Review · Reviewer_PDsK · 2023-07-06

**Soundness:** 3 good
**Presentation:** 3 good
**Contribution:** 3 good
**Rating:** 5
**Confidence:** 2

**Summary:**

This paper summarizes various recent assumptions about the convergence of biased SGD and exposes their implication relations. Furthermore, it proposes a weaker assumption, Biased ABC,  in terms of implication relations and analyzes the results under this assumption.

**Strengths:**

This paper summarizes various recent assumptions about the convergence of biased SGD and exposes their implication relations.
It is well written and in particular, Figure 1 is informative.

**Weaknesses:**

It seems that submitting a paper, which has an Appendix of 50 pages including proofs, to an international conference where the peer review period is short and there is not enough time to adequately check proofs, is a negative impression in itself.
In addition, the fact that the paper ends with a section on experiments and does not include a summary gives a half-hearted impression of the paper's presentation, which I suppose cannot be helped given the insufficient number of pages, but conversely, it seems to mean that this paper is not suitable for an international conference with a tight page limit.

**Questions:**

Although the implication shows that Biased ABC is the weaker assumption, at first glance it seems that this assumption is more difficult to check because it requires more variables to be shown to exist and f* is needed than the other assumptions.
To explain more intuitively why this assumption is weaker, please explain a situation that can be checked with this assumption but not with the other assumptions, or a situation that applies to this assumption but not to the other assumptions.
Also, I know that if Assumption 8 holds, then Assumption 9 holds, but I could not catch the proof that the converse does not hold, so please tell me where in the Appendix it is written.
This is important for showing the weakness of Assumption 9, which leads directly to raising the score.
Since the experiment section is halfway through, wouldn't it be better to put it into the experiment part of the Appendix and write a discussion and conclusion?
 The experiment itself does not seem to make much sense in this paper.

**Limitations:**

Too many pages in Appendix make it difficult to check the correctness of proofs in the limited peer review period of an international conference.
Currently, it is difficult to see that the assumption is weaker than Assumption 8.

---

> ### Author Rebuttal · Authors · 2023-08-08
>
> >**Strengths:**
>
> **Answer:**
>
> We extend our sincere appreciation for the dedication of your time and effort in sharing your valuable feedback and insights on our work. Your comments have proven to be immensely valuable. Additionally, we would like to convey our gratitude to the reviewer for recognizing the fundamental concepts within our paper and for highlighting the ease of readability. We have taken into consideration all the concerns raised by the reviewer, and our responses can be found below.
>
> >**Weaknesses:**
>
> **Answer:**
>
> We politely disagree that paper with 50 pages is not appropriate for international conference. There are many examples of large papers accepted to top international conferences such as ICML, NeurIPS, etc. Let us provide such examples:
>
> Secure Distributed Training at Scale ICML 2022 (https://proceedings.mlr.press/v162/gorbunov22a.html) has **61 pages**
>
> A Statistical Analysis of Polyak-Ruppert Averaged Q-Learning AISTATS 2023 (https://proceedings.mlr.press/v206/li23b.html) has **55 pages**
>
> Variational inference via Wasserstein gradient flows NeurIPS 2022 (https://openreview.net/pdf?id=K2PTuvVTF1L) has **44 pages**
>
> Linearly Converging Error Compensated SGD NeurIPS 2020 (https://proceedings.neurips.cc/paper/2020/file/ef9280fbc5317f17d480e4d4f61b3751-Paper.pdf)** has **99 pages**
>
> Can Reinforcement Learning Efficiently Find Stackelberg-nash Equilibria In General-sum Markov Games With Myopic Followers? ICLR 2022 (https://openreview.net/pdf?id=rqfwi51Tgc) has **40 pages**
>
> Local SGD: Unified Theory and New Efficient Methods AISTATS 2021(http://proceedings.mlr.press/v130/gorbunov21a/gorbunov21a-supp.pdf) has **80 pages**
>
> Recognizing the limitations imposed by the compressed time frame allocated for proof validation, we wish to emphasize our strong conviction that the paper we have submitted is indeed worthy of being showcased at this conference. The crux of our belief rests upon the inherent value embedded within our exploration of the biased SGD method, which we assert holds the potential to deliver substantial benefits to numerous researchers within the machine learning community.
>
> The significance of our study lies in its potential to contribute to a deeper understanding of the nuanced aspects of biased SGD, an area of growing relevance within the realm of machine learning. By unraveling the intricacies and implications of this method, our work has the capacity to provide fresh insights, innovative techniques, and potentially transformative perspectives that could resonate with a wide spectrum of researchers, spanning from novices to seasoned experts.
>
> Moreover, we envision that the dissemination of our findings at this conference could catalyze valuable discussions, foster collaboration, and prompt further exploration within the ML community. The opportunity to share our research in this context holds the promise of stimulating intellectual exchange and collective growth, ultimately advancing the field's collective knowledge and pushing the boundaries of what is currently understood.
>
> In conclusion, we firmly maintain that the substantial potential impact of our paper, coupled with its alignment with the overarching goals and interests of the conference, render it a fitting candidate for inclusion in the conference proceedings. We eagerly anticipate the chance to present our work and engage in the rich discourse that such a platform affords.
>
> >**Questions:**
>
> **Answer:**
>
> We appreciate your inquiry. In many real-world applications of Machine Learning, loss functions exhibit a nonnegative nature, thereby resulting in f* being simply equivalent to 0. The illustration in Figure 1 suggests that Assumption 8 (FSML) serves as a foundation for Assumption 9 (BiasedABC), and this relationship is substantiated by the proof provided in Theorem 2, specifically in part viii. Similarly, the diagram implies that Assumption 6 (BND) implies Assumption 9 (BiasedABC), a deduction which is corroborated by the evidence presented in Theorem 2, part ix. Within these segments of Theorem 2, we also establish the non-validity of the converse implications.
>
> In alignment with the diagram's transitivity principle, the aforementioned connections constitute the entirety of the verifications required. To demonstrate the non-validity of the reverse implications from Assumption 9 to both Assumption 8 and Assumption 6, we employ the same counterexample. It is essential to note that an error exists in the proofs provided for parts viii and ix (pages 30-32), as astutely pointed out by Reviewer 8vPv. Consequently, we present an alternative counterexample, akin to the example put forth in Proposition 1 by Khaled and Richtarik (2013), which effectively substantiates both segments of Theorem 2. Please check the general response.
>
> It is important to highlight that the proof of Claim 1 (page 32) also serves to demonstrate that Assumption 9 does not imply Assumption 6. This is owing to the fact that our scalar product condition pertaining to the bias is considerably less restrictive than its counterpart (13) as delineated in Assumption 6. Overall, we fixed the Theorem 2 viii and ix in the general response, so we showed an example when Assumptions 6 and 8 do not hold, but new Assumption 9 still holds.
>
> Anticipating an additional page allocation for the camera-ready version, we plan to integrate specific sections for discussions and clarifications. However, should the extra page not be approved, we will exclude the incorporation of plots within the main content.
>
> We value your attention to detail and your discerning insights, which significantly contribute to the refinement of our work.
>
> >**Limitations**
>
> **Answer**:
> Please check replies above.
>
> With confidence in our comments addressing your concerns, we respectfully request a reconsideration of your evaluation. We hope our responses prove more satisfactory and justify a higher score. Please acknowledge if more discussion is needed.

---

> > ### Comment · Reviewer_PDsK · 2023-08-19
> > **After rebuttle**
> >
> > Thank you for showing an example when Assumptions 6 and 8 do not hold, but new Assumption 9 still holds.
> > My concern has been adressed.
> > I raised score.
> > However, I still consider that this paper is unsuitable for an international conference where short peer review is forced.
> > We know that papers have already been published at international conferences that devote pages to many proofs, which in itself is considered problematic.
> > We understand the opinion that there is a precedent and that it should be accepted this time as well, but that does not change anything.
> > Considering that papers with incorrect proofs may also pass peer review, papers that need to be carefully checked for proofs should still take such a step.

---

### Official Review · Reviewer_8vPv · 2023-07-12

**Soundness:** 3 good
**Presentation:** 2 fair
**Contribution:** 3 good
**Rating:** 7
**Confidence:** 4

**Summary:**

This paper focuses on the relationships among various assumptions used in the analysis of biased SGD. The authors establish connections among the large number of existing assumptions present in the literature, and propose an assumption (Biased ABC) which captures all prior assumptions. The authors also show that this assumption captures stochastic gradient oracles that prior assumptions did not allow. Using this assumption, they give a simple proof of SGD convergence in a number of settings (both convex and non-convex). While in some cases, the rates are slightly suboptimal (due in part to the more general setting they consider), in many cases, these proofs recover the same (or nearly the same) bounds of prior work under weaker assumptions.

**Strengths:**

This paper provides a very nice overview of the assumptions made in various works on analysis of biased SGD. They establish relationships among these assumptions, and suggest an assumption (Biased ABC) which captures all of these assumptions. This characterization will surely be of great utility to researchers interested or working in this area. By taking a more general assumption, they are able to recover a number of existing results in the literature, sometimes slightly generalizing the results by weakening the assumptions. Sometimes, however, the results obtained are slightly cruder (due to their assumption being weaker). Although there are no significant new proof techniques (at least that I saw) in this paper, I think there is a lot of value in simplifying and unifying results of prior works.

**Weaknesses:**

[Theorem 2 viii and ix]

If I am not mistaken, it appears there is a bug in the proof of “The reverse implication does not hold” part of Theorem 2 viii and ix. You consider $f_1(x) = (x_1^2 - x_2^2)/2$ and $f_2(x) = -f_1(x)$, and claim “Due to Claim 2, it satisfies Assumption 9 (functions $f_1$ and $f_2$ are $1$-smooth)”. To apply Claim 2, however, Assumption 13 must hold, which not only requires the functions $f_i$ to be smooth, but also to be bounded from below. $f_1(x)$ is clearly not bounded from below (take $x_1=0$ and $x_2 \rightarrow +\infty$). Thus, it seems that Claim 2 cannot be applied to your example.

[Organization of the paper]

I personally found the paper a bit hard to navigate. I think this is largely due to the sheer number of claims, definitions, and parts of proofs that are in the paper. I think some reorganization could greatly improve the readability of the paper.

For instance, when reading section B of the appendix, you mention the Biased independent sampling without replacement (Definition 1). I was curious why Assumption 8 can break for this setting (i.e., why you need $A>0$ in Assumption 9), but actually understanding this required finding the proof of Claim 2 and the proof of Theorem 2 viii and ix. It didn’t feel like a natural flow to have to dig so much for what seemed to be the punchline of introducing Definition 1. Is there a way to bring this all in one place?

I understand that organizing a paper with so many short results is difficult. However, I think that some results are less interesting than others, and perhaps can be moved to the end of the paper to reduce clutter. For instance, I think the main interesting parts of Theorem 2 concern Assumption 9. Perhaps you can separate these parts from the others, so that these are easier to find?

Another note is that some parts of the appendix feel like they do not have enough bridge text. For instance, section C is just a long list of Definitions and Claims. Could you add some text describing these settings, or why they might be interesting to the reader?

Also, in the main body, you mention lots of assumptions, but there is no mention of actual examples where the stochastic gradients satisfy these examples. All of this is saved for the appendix. I think it would help readability of the paper if some of the less interesting assumptions were moved to the appendix (perhaps only keep a couple of the main assumptions in the main body), and you moved some of the examples to the main body (e.g., the Biased independent sampling w/o replacement, or another of your choosing).

**Questions:**

[Regarding Assumption ABC]

I found the term $A(f(x) - f^*)$ term in eq. (19) to be somewhat mysterious, yet there is not much discussion in the main body of your paper as to why you consider this term in the variance bound. In the appendix, it becomes somewhat clear that you introduce $A$ to handle “Biased independent sampling”, but this this isn’t discussed much in the paper. Are there other reasons why you consider this term in the variance bound?

It is worth noting that a similar assumption (in the unbiased setting) was considered by Polyak and Tsypkin (1973) “Pseudogradient Adaptation and Training Algorithms” (see the first equation in their Section 3). They also don’t really discuss why they consider this term in the variance bound. I don’t think I’ve seen this type of assumption anywhere else in the literature (except in Khaled and Richtárik (2023) “Better theory for SGD in the nonconvex world”, as you mention). Perhaps you have some insights into this? Also, it is probably worth giving the Polyak and Tsypkin paper a citation, given the similarity of your assumptions.

[Regarding the discussions in weaknesses section above]

Could you please address if the error in the proof of Theorem 2 viii and ix is easy to overcome? (I will set my score now to weak accept due to this error).

Could you please describe how you plan to incorporate the feedback on paper organization?

[Regarding Theorem 3]

Theorem 3 gives a guarantee on $\min_t E[||\nabla f(x_t)||^2]$, while, at least for the unbiased setting, the results of Ghadimi and Lan (2013) give convergence for the average iterate. Looking at your proof, it appears that you actually get convergence for a weighted average of the gradients, but where the *smaller* iterates are weighted higher. That seemed quite counter-intuitive to me, as I would have expected a higher weighting of later iterates. Do you have any intuition on why the weighting is this way? Is there any hope in getting convergence of the average iterate in your framework?

---

> ### Author Rebuttal · Authors · 2023-08-07
>
> >**Strengths:**
>
> **Answer:**
>
> We appreciate for the time and dedication you devoted to sharing your valuable perspectives on our paper. Your comments were tremendously beneficial. We also extend our thanks for recognizing the fundamental concepts within our paper. We have taken into account and addressed all the concerns raised in the text below.
>
> >**Weaknesses:**
>
> >**[Theorem 2 viii and ix]:**
>
> **Answer:**
>
> Thank you for the checking this proof! We appreciate that you find a mistake. We agree that these two parts of the theorem are not correct. Please check the fixed proof of Theorem 2 viii and ix in the general response above.
>
> >**[Organization of the paper] **
>
> >Is there a way to bring this all in one place?
>
> **Answer:**
>
> Thank you for reviewing Appendix B. We agree that introducing Definition 1 closer to the proof of Theorem 2 makes sense. One option is to place it in Appendix C alongside other gradient estimators. Alternatively, it could be introduced near the proof of Theorem 2. Our preference is to include Definition 1 in Appendix C with the other estimators, as we find this to be the best approach. Your suggestions are welcome and appreciated.
>
> >Perhaps you can separate...
> **Answer:**
>
> We appreciate your suggestion. We are considering dividing the theorem into two distinct sections. The initial section will encompass statements from i to vii, while the subsequent section will incorporate statements viii and ix. Additionally, we plan to allocate these two segments into separate subsections. This arrangement will not only facilitate convenient access but will also enhance their visibility within the table of contents.
>
> >Could you add some text...
>
> **Answer:**
>
> We appreciate your valuable comment. We agree with your suggestion to add explanatory comments between definitions and claims for clarity. We're committed to providing practical examples to enhance understanding and adding intuitive explanations after theoretical claims. Your feedback has refined our work, and we're excited to implement these improvements.
>
> > I think it would help readability...
>
> **Answer:**
>
> Thank you for your idea! With an additional page expected in the camera-ready version, we're excited about the potential to add more examples to the main paper. However, moving some definitions to the appendix might not be possible due to their role in Figure 1. Balancing visual coherence and conceptual integrity is a priority as we consider this. Your input is invaluable in shaping our approach, and we're committed to enhancing the paper's quality and effectiveness.
>
> >Questions:
> >[Regarding Assumption ABC]
>
> **Answer:**
>
> We can shed light on the reasoning behind the utilization of the term $A(f(x)−f∗)$ as follows. When delving into the analysis of the 2nd moment of a gradient estimator, a common scenario emerges where we find ourselves needing to constrain the expression $E := \sum_{i=1}^n q_i\Vert \nabla f_i(x) \Vert^2$, where $f(x) = \frac{1}{n}\sum_{i=1}^n f_i(x)$. While $E$ cannot be confined solely by the norm of the overall gradient $B\Vert\nabla f(x)\Vert^2$, nor by a constant $C$, smoothness suffices to bound this by $A(f(x)−f∗)$. Further, there exist quadratic stochastic opt problems where the 2nd moment of an (unbiased) stochastic gradient is precisely equal to $2(f(x)-f_*)$ (take expectation on both sides of (3.11) in Richtarik, Takac, Stochastic Reformulations of Linear Systems: Algorithms and Convergence Theory, SIMAX 2020).
>
> When functions show varying gradient magnitudes and estimators align closely with those magnitudes, constraining the 2nd moment of the estimator becomes tough. In such cases, bounding the 2nd moment becomes impossible. We'll delve into this in our paper, especially in Sec 5, providing clarifying explanations for better understanding.
>
> >Also, it is probably worth giving the Polyak and Tsypkin paper a citation...
>
> **Answer:**
>
> Much appreciated, we will reference the work of Polyak and Tsypkin (1973). As highlighted by Khaled and Richtárik (2023) in their publication "Better theory for SGD in the nonconvex world," Polyak and Tsypkin explored a related assumption during their analysis of pseudogradient algorithms. They succeeded in establishing an asymptotic convergence bound for a variant of gradient descent in the unbiased scenario. In contrast, our study focuses on non-asymptotic convergence rates in the biased setting.
>
> >[Regarding the discussions in weaknesses section above]
>
> **Answer:**
>
> Please answers regarding weaknesses above. Overall, we are going to move definition 1 to appendix C or even closer to Theorem 2.
>
> >[Regarding Theorem 3]
>
> **Answer:** We need to obtain telescoping sum in equation (57) from equation (56). Equation (56) states that
> $$  \frac{\gamma b}{2} r^t \leq\left(1+L A \gamma^2\right) \delta^t-\delta^{t+1}+\frac{L C \gamma^2}{2}+c \gamma.$$
> In order to have  telescoping sum we need to have coordinated coefficients with consistent indices. To make it possible we use $w_t=\frac{w_{t-1}}{1+L A \gamma^2}$. Finally, we obtain
> $$\frac{b w_t r^t}{2} \leq \frac{w_{t-1}}{\gamma} \delta^t-\frac{w_t}{\gamma} \delta^{t+1}+\frac{L C \gamma w_t}{2}+\frac{c w_t}{2},$$
> For every $ 0 \leq t \leq T-1$, we get
> $$\frac{b}{2} \sum_{t=0}^{T-1} w_t r^t \leq \frac{w_{-1}}{\gamma} \delta^0-\frac{w_{T-1}}{\gamma} \delta^T+\frac{L C \gamma+c}{2} \sum_{t=0}^{T-1} w_t.$$
> The underlying idea is that the term ${1+L A \gamma^2}$ surpasses the value of $1$, necessitating a reduction by the same factor to align coefficients. However, obtaining guarantees for the average iterate at this level of generality remains a challenge. It is possible that by considering more specific assumptions, we can derive a rate for the average iterate.
>
> We believe our comments have effectively addressed your concerns. We kindly request a reconsideration of your initial evaluation. The added explanations may justify a higher score. Please acknowledge if further discussion is necessary.

---

> > ### Comment · Reviewer_8vPv · 2023-08-18
> > **Thanks**
> >
> > I would like to thank the authors for their detailed responses to my questions. I've reviewed their fixes to the proof of Theorem 2 parts viii-ix, and I agree that, for this choice of objective and stochastic gradients, the "reverse implication" does not hold. Thanks for fixing this. In the next version of the paper, I would still suggest that the authors incorporate some of the reorganization as discussed above. Additionally, I think it would benefit the paper to include the author's response to "Regarding Assumption ABC" above in the next revision. With these changes to the paper, I am happy to raise my score from 6 -> 7.

---

### Author Rebuttal · Authors · 2023-08-07

Dear Reviewers, it was noted by several Reviewers that there is a mistake in the proof of Theorem 2 parts viii-ix, where the absence of the reverse implications from Assumption 9 towards Assumptions 6 and 8 is established. The counterexample we provided was incorrect. We present the corrected version with the proof below. Please, kindly check it! We are intending to replace the current part of the proof with this piece.

Consider function $f$ which is $1$-smooth and lower bounded by $0:$
	$$
	f(x)= \begin{cases}\frac{x^2}{2}, & \text { if }|x|<1, \\\\ |x|-\frac{1}{2}, & \text { otherwise}\end{cases}
	$$
	(Huber Loss). Consider a biased estimator
	$$
	g(x)= \begin{cases}\nabla f(x)+\sqrt{|x|}+1 & \text { with probability } 1 / 2, \\\\ \nabla f(x)-\sqrt{|x|}+1 & \text { with probability } 1 / 2.\end{cases}
	$$
	Observe that $\mathbb{E}g(x)=\nabla f(x) + 1.$ Suppose condition (17) of Assumption 8 holds. Then
	$$
	\exists U, Q\geq 0:\quad \mathbb{E}\left[\Vert g(x)\Vert^2\right] - \Vert \mathbb{E}\left[g(x)\right]\Vert^2 \leq U\Vert \nabla f(x)\Vert ^2+Q.
	$$
Consider the point $x=U+Q+4$, then $|x|>1$ and hence $\nabla f(x)=1$ by the definition of $f.$ Then
	$$
	\mathbb{E}\left[\Vert g(x)\Vert ^2\right] \leq U+Q+4.
	$$
	On the other hand, we fall into contradiction since
	$$
			\mathbb{E}\left[\Vert g(x)\Vert ^2\right] = \frac{1}{2}\left((2+\sqrt{x})^2+(2-\sqrt{x})^2\right) = x + 4 = U+Q+8.
	$$
	It follows that condition (17) of Assumption 8 does not hold. Suppose that condition (12) of Assumption 6 holds. Then
	$$
	\exists M,\sigma^2\geq 0: \quad \mathbb{E}\left[\Vert g(x)\Vert ^2\right] - \Vert \mathbb{E}\left[g(x)\right]\Vert^2 \leq M\Vert\mathbb{E}\left[g(x)\right]\Vert^2+\sigma^2.
	$$
	Consider the point $x=4(M+1) + \sigma^2,$ then $|x|>1$ and hence $\nabla f(x) = 1$ by the definition of $f.$ Then we obtain that
	$$
	\mathbb{E}\left[\Vert g(x)\Vert^2\right] \leq 4(M+1) + \sigma^2.
	$$
	On the other hand, we fall into contradiction since
	$$
			\mathbb{E}\left[\Vert g(x)\Vert^2\right] = \frac{1}{2}\left((2+\sqrt{x})^2+(2-\sqrt{x})^2\right) = x + 4 = 4(M+1) + \sigma^2 + 4.
	$$
	It follows that condition (12) of Assumption 6 does not hold.  We now show that Assumption 9 holds: first, suppose that $x\geq 1$, then
	$$
		\mathbb{E}\left[\Vert g(x)\Vert^2\right] =\frac{1}{2}\left((2+\sqrt{|x|})^2+(2-\sqrt{|x|})^2\right)
		=\frac{1}{2}(8+2|x|)=4+|x|
$$
$$
		=\frac{9}{2}+\left(f(x)-f^{\text {inf }}\right),
	$$
	since for $|x| \geq 1$ we have $f(x)-f^{\inf }=|x|-1 / 2.$  In turn,
	$$
	\langle \nabla f(x), \mathbb{E}\left[g(x)\right]\rangle = \langle \nabla f(x), \nabla f(x) + 1\rangle \geq \Vert \nabla f(x)\Vert^2.
	$$
	Suppose that $x\leq -1,$ then $\nabla f(x) = -1,$ and
	$$
		\mathbb{E}\left[\Vert g(x)\Vert^2\right] =\frac{1}{2}\left((\sqrt{|x|})^2+(-\sqrt{|x|})^2\right) \\
		=|x| =\frac{1}{2}+\left(f(x)-f^{\text {inf }}\right).\\
	$$
	In turn,
	$$
		\langle \nabla f(x), \mathbb{E}\left[g(x)\right]\rangle = \langle \nabla f(x), \nabla f(x) + 1\rangle = \Vert \nabla f(x)\Vert^2 - 1.
	$$
	Now suppose that $|x| \leq 1$, then
	$$
		\mathbb{E}\left[\Vert g(x)\Vert^2\right] =\frac{1}{2}\left((x+\sqrt{|x|})^2+(x-\sqrt{|x|})^2\right) \\
		=x^2+|x| \leq 1+1=2.
	$$
	In turn,
	$$
	\langle \nabla f(x), \mathbb{E}\left[g(x)\right]\rangle = \langle \nabla f(x), \nabla f(x) + 1\rangle =\Vert \nabla f(x)\Vert^2+x\geq \Vert \nabla f(x)\Vert^2 - 1.
	$$
	It means that, for all $x \in \mathbb{R}$,
	$$
	\mathbb{E}\left[\Vert g(x)\Vert^2\right] \leq f(x)-f^{\mathrm{inf}}+\frac{9}{2}
	$$
	and
	$$
	\langle \nabla f(x), \mathbb{E}\left[g(x)\right]\rangle \geq \Vert \nabla f(x)\Vert^2 - 1.
	$$
	It follows that Assumption 9 is satisfied with $A=1 / 2, B=0$, and $C=\frac{9}{2},$ $b = 1,$ $c=1.$

Thank you!

Several reviewers have pointed out that our explanation for the motivation behind the newly proposed Assumption 9 is not sufficiently detailed. Specifically, the introduction of the term 2A(f(x) - f^{*}) in equation (19) lacks clarity and is not adequately addressed in the main section. When bounding the second moment of the sampling gradient estimator, it becomes necessary to bound the weighted sum of the squared norms of individual gradients. However, it is generally not feasible to bound this by the squared norm of the gradient alone (as the gradient is essentially a sum of individual gradients, and the sum of squared functions cannot always be bounded by the square of their sum) or by a constant. This rationale underscores the introduction of the A-term in equation (19), as it aids in bounding the second moment of the sampling gradient estimators.

Concerning the challenges in verifying the BiasedABC assumption, it's worth mentioning that in Machine Learning, loss functions are commonly bounded from below by $f^{*}=0$. Additionally, it's important to note that Assumption 8, introduced in the classical work by L. Bottou, involves checking three conditions as opposed to the two conditions in Assumption 9. In Tables 2 and 8, we present the constants with which numerous practical estimators meet our assumption. Furthermore, Claims 2 – 4 can aid in determining these constants for various sampling schemes.

---

### Decision · Program_Chairs · 2023-09-21

**Decision:**

Accept (poster)

**Comment:**

The paper unifies several previous assumptions and analysis for biased gradient problems.

It is agreed upon reviewers that the paper does a good job in unifying the various existing assumptions on Biased SGD, and will serve as a good reference for future work. Nevertheless, the contribution of the paper on the algorithmic and analytic aspects feels a bit incremental.

The paper is borderline, but all in all most reviewers believe it should be accepted and hence the decision to accept.